## Replications

psychology

cognitive control, conflict adaptation, domain-generality, replication

**Author for correspondence:**
Balazs Aczel
e-mail: aczel.balazs@ppk.elte.hu

# Is there evidence for cross-domain congruency sequence effect? A replication of Kan et al. (2013)

Balazs Aczel[1], Marton Kovacs[1,2], Miklos Bognar[1,2], Bence Palfi[3], Andree Hartanto[4], Sandersan Onie[5], Lucas E. Tiong[4] and Thomas Rhys Evans[6]

[1]Institute of Psychology, and [2]Doctoral School of Psychology, ELTE, Eotvos Lorand University, Budapest, Hungary
[3]School of Psychology, University of Sussex, Brighton, UK
[4]School of Social Science, Singapore Management University, Singapore, Singapore
[5]School of Psychology, University of New South Wales, Sydney, New South Wales, Australia
[6]School of Human Sciences, University of Greenwich, London, UK

BA, 0000-0001-9364-4988; BP, 0000-0002-6739-8792; TRE, 0000-0002-6670-0718

Exploring the mechanisms of cognitive control is central to understanding how we control our behaviour. These mechanisms can be studied in conflict paradigms, which require the inhibition of irrelevant responses to perform the task. It has been suggested that in these tasks, the detection of conflict enhances cognitive control resulting in improved conflict resolution of subsequent trials. If this is the case, then this so-called congruency sequence effect can be expected to occur in cross-domain tasks. Previous research on the domain-generality of the effect presented inconsistent results. In this study, we provide a multi-site replication of three previous experiments of Kan et al. (Kan IP, Teubner-Rhodes S, Drummey AB, Nutile L, Krupa L, Novick JM 2013 *Cognition* **129**, 637–651) which test congruency sequence effect between very different domains: from a syntactic to a non-syntactic domain (Experiment 1), and from a perceptual to a verbal domain (Experiments 2 and 3). Despite all our efforts, we found only partial support for the claims of the original study. With a single exception, we could not replicate the original findings; the data remained inconclusive or went against the theoretical hypothesis. We discuss the compatibility of the results with alternative theoretical frameworks.

# 1. Introduction

In day-to-day life, we often need to override automatic behaviours or habitual responses in order to reach certain goals. Most of us can relate to having to tear ourselves away from appealing foods at the store in order to make a timely supermarket run or to meet certain nutrition goals. One key factor in overriding pre-potent responses is *cognitive control*, which is broadly defined as the collection of processes which contribute to the generation of a goal-relevant response [1]. Cognitive control is typically studied using conflict paradigms, which requires the inhibition of irrelevant stimuli, features or a habitual response to perform the task. One such paradigm is the Stroop task [2].

In a typical Stroop trial, participants are presented with colour words in various print colours but are only tasked with indicating the print colour. *Congruent* trials are those in which the word and ink colour are the same (e.g. 'BLUE' in the colour blue); on *incongruent* trials, the word and the ink colour are different (e.g. 'BLUE' in the colour red). Typical findings of the Stroop task are that responses on incongruent trials are slower and less accurate compared to congruent trials due to the conflict between the word and its ink colour. Since the *Conflict Monitoring Theory* [3] posits that the detection of conflict triggers cognitive control mechanisms,[1] these mechanisms can be studied by comparing performance on the incongruent trials with responses on congruent trials. Greater cognitive control is thought to be reflected in smaller differences between congruent and incongruent trials in terms of accuracy and speed.

Previous studies have also found that congruency effects are smaller following incongruent trials rather than congruent trials, called the *congruency sequence effect* (also known as conflict adaptation or Gratton effect, [4]). However, Egner [5] proposed that the recruitment of control occurs only following the detection of the same type of conflict (e.g. Stroop conflict). Therefore, this hypothesis suggests that the overlap of stimulus dimensions between the conflict tasks is a central determinant of the transfer of control. If the congruency sequence effect is domain-general then it could be expected that cognitive control can be sustained across seemingly different tasks. For testing this hypothesis, researchers use cross-task adaptation designs in which one type of conflict task (e.g. a verbal Stroop task) is followed by a different type of conflict task (e.g. a non-verbal Flanker task). In the case of domain-generality, we should expect the effect on cross-tasks as well. Conversely, if the congruency sequence effect is domain-specific, cross-domain adaptation should not be observable.

Some early studies gave support for domain-generality ([6–8], Condition 1), while others have not found the effect ([8], Condition 2, [9–11]), for a summary, see Braem *et al*. [12]. One proposed possibility for these mixed findings was that two different conflict sources (e.g. stimulus location and stimulus feature) were simultaneously present in the designs [13]. In these cases, it is possible that the cross-task adaptation effect became masked when incongruity in one feature was presented with congruency in the other feature creating weaker conflict signals compared to the cases that induce incongruity in both of the features.

To address these challenges, Kan *et al*. [13] used a novel design in which 'conflict adaptation' was tested between entirely different domains. In their first experiment, participants were given sentences with or without syntactic conflict, followed by a Stroop trial. Syntactic conflict was manipulated using garden path sentences, which are sentences that lure the reader into one interpretation, while a clause at the end forces the reader to reanalyse the sentence, revealing its true meaning. For example, the sentence: 'The basketball player accepted the contract would have to be negotiated', is ambiguous without reinterpreting the sentence after it is read in its entirety. To eliminate this conflict, inserting 'that' after 'accepted' would disambiguate the sentence. The results showed that the detection of syntactic conflict on the sentence trials enhanced conflict resolution on the subsequent, non-syntactic Stroop tasks. In their second and third experiment, the Stroop task was preceded by a non-verbal, purely perceptual task. These designs were meant to test whether cross-task adaptation can be generalized beyond the verbal domain. The Necker cube [14] was also adopted for this study, as it can create ambiguity by inducing two different perceptual percepts. If this bistable stimulus leads to conflicting experience, then the congruency sequence effect can be expected in the subsequent Stroop trial. Compared to Stroop trials after stable, unambiguous versions of the cube, accuracy measures reflected enhanced cognitive control after the experience of conflict, providing support for cross-task adaptation. These results provide a strong argument that the detection of conflict in one domain can enhance control mechanisms in a different domain.

Hsu & Novick [15] found evidence for the domain-generality of the congruency sequence effect in reversed setting, when following incongruent Stroop trials, the listeners' ability improved in revising

---

[1]While there are other approaches in which conflict detection/experience is not the direct trigger of control engagement, we deliberate about them only in the Discussion as they are not central to our investigation.

temporarily ambiguous spoken instructions that induce brief misinterpretation. Other research also claimed the cross-task adaptation of conflict and showed that the effect is the largest when the tasks depend on the same cognitive-control mechanism [16].

However, a number of studies seem to support the domain or dimensional specificity of the congruency sequence effect. For example, Feldman et al. [17] analysed event-related potential in a Go/NoGo task using various NoGo decision criteria (i.e. when (not) to go) across the trials. The results of response time, response accuracy and event-related potential analyses indicated the presence of the congruency sequence effect only when the same NoGo decision criterion was applied across the consecutive trials. Conflict-specificity (or lack of domain-generality) of cognitive control was further supported in several studies [18–23].

The exploration of the neural background of conflict-control also shows an inconsistent set of results. While some studies supported complete domain-specificity [24–26], others found evidence for domain-generality [27] or a hybrid architecture of the two [28].

In summary, although research on the domain-generality of the congruency sequence effect has the potential to reveal important fundamental aspects of cognitive control mechanisms, the empirical data collected provide mixed evidence for cross-domain adaptation effect. Notably, most arguments against domain-generality are based on the absence of evidence, leaving the possibility open that those designs lacked sensitivity or sample size to detect the effect of interest. For these reasons, it is an advisable strategy to investigate the question by replicating a study which had provided empirical support for domain-generality. We decided to conduct a direct replication of Kan et al.'s [13] all three experiments for the following reasons:

(1) these three experiments tested the congruency sequence effect between very different domains: from a syntactic to a non-syntactic domain (Experiment 1), and from a perceptual to a verbal domain (Experiments 2 and 3);
(2) they provided strong and influential evidence on cross-task adaptation;
(3) we judged that the methodological parameters of the design allow for the conduct of a direct replication.

# 2. Methods

## 2.1. Materials

As a direct replication, this experiment closely followed the methods and procedures of the original study. The few minor deviations from the original protocol are highlighted.

All participants performed two tasks parallel in a task switching setting: either a sentence-processing task (Experiment 1) or a perceptual processing task (Experiments 2 and 3), intermixed with Stroop trials. All the tasks included congruent (unambiguous) and incongruent (ambiguous) stimuli. The four conditions were determined by the conflict state of the preceding and current stimulus: congruent–congruent (cC); congruent–incongruent (cI); incongruent–congruent (iC); and incongruent–incongruent (iI) sequences. All tests ended with a question asking whether the participant experienced any technical problems or whether they have any comment on the experiment. All the materials are openly available on the project's OSF page: https://osf.io/6bd43/.

## 2.2. Participants

According to our registration, we did not analyse the data until we collected at least 2.5 times the sample size of the original experiment [29], which were 103, 70 and 38 participants for Experiments 1, 2 and 3, respectively. Our first analyses did not provide good enough evidence for either H0 or H1; therefore, we kept collecting data while regularly conducting the analyses.[2] Since even with our greatly extended sample, we did not reach our preset evidential thresholds for the crucial tests (for more details, see the Results section of each experiment), we terminated data collection after reaching the end of our timeframe. The first experiment has been conducted as a multi-site project. Together, 153 native speakers of English from Australia, Singapore and the UK participated in the first experiment.

[2]Note that the Bayes factor retains its meaning irrespective of the used stopping rule, hence using optional stopping did not bias our conclusion [30,31].

In Experiments 2 and 3, we tested Hungarian native speakers as the stimuli were not specific to the English language. We collected data from 178 and 94 participants, respectively.

Eligibility for participation in each experiment was based on age (greater than or equal to 18 years), being a native speaker of the language of the test and being right handed. We did not collect any identifiable private data during the project. Each laboratory ascertained that the local institutional ethical review board agreed with the proposed data collection, which was conducted in accordance with the Declaration of Helsinki.

## 2.3. Statistical analyses

Our data were nested (i.e. level of trials embedded into the level of participants); therefore, we calculated aggregate scores within conditions for each participant and conducted all of the analyses on the level of the individuals. Hence, our analyses were identical to those of the original paper. We performed and reported all of our hypothesis tests as ANOVAs or paired-sample $t$-tests.[3] The steps of the analysis of the second and third experiments were similar to those of the first experiment. Therefore, we do not provide a detailed description of the analyses of the second and third experiments, except where they differ from the first experiment.

To test our hypotheses, we calculated Bayes factors ($B$), which is a measure of relative evidence provided by the data for one model over another one (in our case, for H1 over H0). We used the Bayes factor instead of frequentist statistics as the former one can distinguish between insensitive evidence and evidence for the null, which makes it more appropriate for the purpose of this replication project. We applied the conventional cut-off of 3 and ⅓ of $B$ to differentiate good enough evidence for H1 and H0, respectively [30]. We reported $p$-values along the $B$s for each test. To calculate the $B$s, we applied the R script of Dienes & Mclatchie [33] that models the predictions of H1 in raw units rather than in standardized effect sizes. We modelled the predictions of all H1s with half-normal distributions with a mode of zero as all of the alternative hypotheses have directional predictions and they presume that smaller effects are more probable than large ones [34]. Considering comparability of the $B$s across experiments, we used the parameters of the first experiments to model the predictions of the H1s of the second and third experiments (i.e. we used identical parameters across all experiments for models testing the same hypothesis). The s.d. of the H1 models are specified and justified in the Results section of Experiment 1.

We notated the $B$s as $B_{H(0, x)}$ in which $H$ implies that the distribution is half-normal, 0 indicates the mode of the distribution and $x$ stands for the s.d. of the distribution that can vary among models. We implemented all of our analyses with the R statistical software. For the NHST analyses, we used $\alpha = 0.05$ across all analyses.

As a sensitivity test, we reported analyses on the arcsin-transformed proportions, following the original procedure (electronic supplementary material, table S2).

## 2.4. General data pre-processing

For the reaction time (RT) analyses, we excluded the erroneous trials. After this exclusion, some participants had no correct trials left in at least one of the four conditions (iI, iC, cC, cI). We excluded these participant's responses from both the RT and the accuracy analyses. After these exclusions, we were left with the data of 152 (Experiment 1), 163 (Experiment 2) and 88 (Experiment 3) participants for further analysis. We replaced those Stroop trial RTs with their cut-off value where the RT was 2.5 s.d. slower or faster than the participants' overall mean RT. We conducted the accuracy data analyses on the raw proportions.

# 3. Experiment 1

## 3.1. Materials and procedure

Participants completed an intermixed sequence of Stroop and sentence trials. They read the sentences word-by-word, advancing to the next word by pressing the keyboard spacebar. We used the same sentences as the original research which were designed by Garnsey et al. [35], half of which are

---

[3]Note that this differs from our Stage 1 registration where we planned to perform only $t$-tests. We performed ANOVAs for practical reasons, but it has no effect on our results as $2 \times 2$ ANOVAs to test the interaction and the main effects lead to the same results as conducting paired $t$-tests on the difference and aggregated scores, respectively (e.g. [32, pp. 63–65]).

congruent and the other half incongruent (available on our OSF project site: https://osf.io/zkm2a/). Incongruent sentences temporarily mislead the interpretation during the reading process, and participants have to revise the content as they get to the end of the sentence. Congruent sentences are not misinterpretable and they do not need cognitive-control supported reanalysis.

Example sentences from Kan *et al.* [13]:

— 'The basketball player accepted the contract would have to be negotiated'. (Temporally incongruent/ ambiguous)
— 'The basketball player accepted that the contract would have to be negotiated'. (Congruent/ unambiguous)

In the incongruent sentence, the verb 'accept' can be interpreted as if its direct object is the noun 'contract'; however, as the participants encounter the 'would have' part, they have to reanalyse the meaning of the sentence. This feature is expected to increase the reading time due to processing difficulty. According to the original paper, this is the central source of conflict in the sentence task.

By inserting 'that' in the sentence, we diminished the possibility of temporal misinterpretation.

We used a pseudo-randomized sequence of 162 experimental trials (60 congruent and 60 incongruent Stroop trials, as well as 21 congruent and 21 incongruent sentences). A total of 29 filler sentences were included to avoid expectation-driven strategies and for the easy adaptation to the same type of experimental sentences. All filler sentences were unambiguous. Non-filler trials had four conditions that described the conflict state of the preceding and the current trial (cC, cI, iC, iI).

In Stroop-tests, participants used their dominant hand to submit their answers via button presses. Three adjacent keyboard buttons were marked with blue, green and yellow patches. Colour names matched with the ink colour in congruent trials, and mismatched in incongruent trials, but mismatching colour names were response-ineligible. We used only colour names that are not in the response set (e.g. brown, orange). Incongruencies such as this generate conflict in the representational level, and not the response level [26]. We measured RT in milliseconds and accuracy of the Stroop trials. In every experiment, the participants performed a practice sequence with all the possible stimulus types before each trial.

Every trial started with a 500 ms fixation period which was followed by either a Stroop or a sentence stimulus. The Stroop stimuli stayed on screen for 1000 ms. Sentence stimuli started with a full mask (dashes replacing every letter of the sentence) and words appeared by the participants' spacebar presses on the keyboard. Only one word was visible, the recent parts of the sentence were also masked. Word reading duration was recorded. After every trial, an empty inter-trial interval was shown for 1000 ms. After the inter-trial interval of certain filler sentences, a comprehension probe appeared and remained on the screen until the participant submitted a true or false answer. After the response, a 1500 ms blank screen preceded the next trial.

Before the experimental sequence, there were 10 Stroop trials which helped participants learn the keyboard mappings for the colour responses. It was followed by a baseline trial with 145 intermixed congruent, incongruent and filler (not colour word) Stroop stimuli. The sentence tasks were preceded by comprehension probe filler sentences to demonstrate the step-by-step nature of sentence stimuli. Subsequently, an extra practice sequence was presented with 10 Stroop and 10 filler sentence stimuli pseudo-randomly intermixed. Practice sequences contained only filler stimuli without any incongruency.

## 3.2. Data pre-processing

The data of those participants who experienced any technical problems during the experiment and of those who performed under 70% level of comprehension probe trials were not included in any of the analyses.[4] Therefore, we excluded 20 participants from Experiment 1. To mitigate the influence of outlier RTs, we replaced the raw reading times of those trials (words) that are 2.5 s.d.s above the participant's mean word reading time across all the sentences with the 2.5 s.d. cut-off value of the participant. In order to account for the biasing effect of the length of the word on the reading time, we calculated the residual reading time for each word. For every sentence region (see in the OSF repository of the project: https://osf.io/2v39r/) of each participant, we applied a simple linear regression with the actual reading time as the outcome variable and the length of the region in the number of characters as the predictor variable. We included only the congruent ($N = 21$) and the filler

---

[4]This threshold is based on the overall comprehension probe performance of the original experiment.

sentences ($N = 29$) in the linear regression, as measures of normal reading time.[5] From this calculation, we excluded the reading times of the incongruent sentences as the ambiguity manipulation is expected to make these times longer than the normal reading time. To compute the residual reading time of the sentence regions of every sentence for each participant, we subtracted the predicted reading time of the given sentence region from the actual reading time. If a sentence region consisted of more than one word, we averaged the residual reading times of the words. For a more detailed description, see the electronic supplementary materials. For both of the RT and accuracy analyses, we only included Stroop trials that were preceded by a sentence reading trial.

## 3.3. Results

### 3.3.1. Outcome neutral tests

Of the 132 participants who scored above or equal to 70% on the sentence comprehension probe test, the mean comprehension score was 0.85 (s.d. = 0.1). Moreover, 145 participants scored above chance level on the comprehension probe test with a mean comprehension score of 0.82 (s.d. = 0.12).

We tested the presence of the ambiguity effect by comparing the mean residual reading times of congruent and incongruent sentences in each sentence region separately. For the comparison, we used a one-tailed paired $t$-test where for the alternative hypothesis, we expected the mean residual reading time to be greater for the incongruent than for the congruent sentences. To model the predictions of the H1s, we used (as for all following analyses) a half-normal distribution, centred at 0 with s.d.s taken from the effect sizes of the original study. These values are 40 and 18 ms for the temporarily ambiguous and disambiguating regions, respectively. For the other sentence regions (sentence region 1, 2, 5, 6 and 7), we used the more conservative 18 ms raw effect size as our scale as it is less lenient towards the H0. We used the data of 132 participants for each sentence region comparison. We found strong evidence for the difference between the congruent and incongruent sentences in residual reading time, with the incongruent one being longer, in the temporarily ambiguous sentence region: $M = 17.26$ ms, $B_{H(0,40)} = 7.87 \times 10^6$, RR [0.6, $6.13 \times 10^3$] ($t_{131} = 6.40$, $p < 0.01$), and we found weak evidence for the difference for the disambiguating region: $M = 7.56$ ms, $B_{H(0,18)} = 2.88$, RR [17.20, $1.73 \times 10^2$] ($t_{131} = 2.05$, $p = 0.02$).[6] We also found strong evidence for a difference in the mean residual reading time in the fifth sentence region but not in the other sentence regions. Electronic supplementary material, figure S1A and table S1A show the difference in the mean residual reading time for all sentence regions.

We tested the presence of the Stroop effect in the RT data by comparing the RTs of congruent and incongruent trials. The s.d. of the distribution modelling the predictions of H1 were 28 ms, which was the size of the Stroop effect in the original study. We found a Stroop effect with $M = 38.75$ ms, $B_{H(0,28)} = 4.61 \times 10^{13}$, RR [0.8, $1.45 \times 10^4$] ($F_{1,131} = 87.03$, $p < 0.01$).

### 3.3.2. Crucial tests

The two key analyses (i.e. tests of the presence of congruency sequence effect) are the tests of the interaction between the congruency of the current Stroop trial and the congruency of the preceding sentence reading trial on the RT and accuracy data. We modelled a 30 ms s.d. for the RT analysis and 0.03 s.d. for the accuracy analysis for H1. Both of the values are equal to the raw effect sizes found in the original study. The congruency sequence effect for the RT analysis was inconclusive $M = 3.88$ ms, $B_{H(0,30)} = 0.37$, RR [0, 33] ($F_{1,131} = 0.35$, $p = 0.56$), while we found good enough evidence for an effect for the accuracy analysis with $M = 0.05$, $B_{H(0,0.03)} = 10.60$, RR [0.0095, 0.28] ($F_{1,131} = 6.50$, $p = 0.01$). Figure 1 shows the mean RT results and figure 2 shows the mean accuracy results, both broken down by the congruency type of the preceding sentence and current Stroop trials.

---

[5]As a sensitivity analysis, we recalculated the residual reading times with the inclusion of the incongruent and congruent sentences. The detailed description of the sensitivity analysis can be found in the electronic supplementary material.

[6]The results of the sensitivity analysis showed a similar pattern. We found a difference between the congruent and incongruent sentences in the temporarily ambiguous sentence region $B_{H(0,40)} = 4.55 \times 10^6$, RR [0.59, $5.89 \times 10^3$] ($t_{131} = 6.29$, $p < 0.01$). For the outcome neutral test, we found good enough evidence for the difference in the disambiguating region with $B_{H(0,18)} = 3.33$, RR [2.6, 20] ($t_{131} = 2.12$, $p = 0.02$). For the sensitivity test, electronic supplementary material, figure S1B and table S1B show the difference in the mean residual reading time of all the sentence regions.

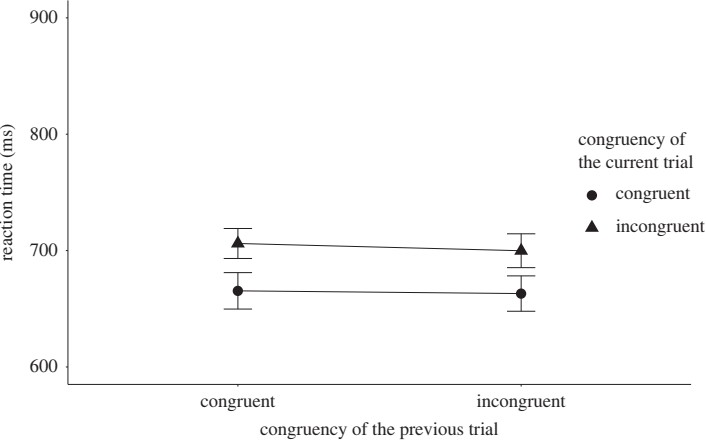

**Figure 1.** The figure shows the mean RT broken down by the congruency type of the preceding and current trials for Experiment 1. The X-axis shows the congruency type of the previous trial, while the Y-axis shows the mean RTs. The legend shows the congruency type of the current sentence trials. The error bars represent the 95% confidence interval.

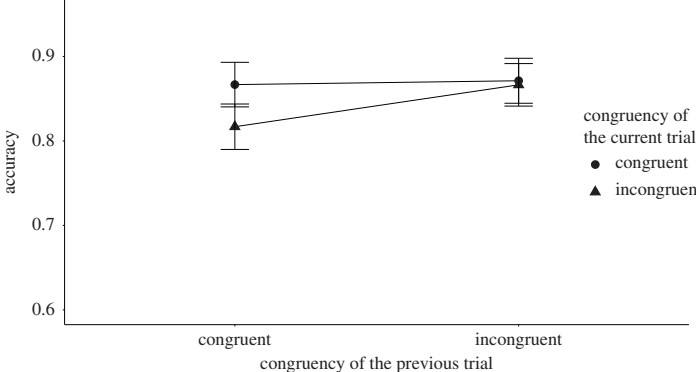

**Figure 2.** The figure shows the mean accuracy results broken down by the congruency type of the preceding and current trials for Experiment 1. The X-axis shows the congruency type of the previous trial, while the Y-axis shows the accuracy. The legend shows the congruency type of the current sentence trials. The error bars represent the 95% confidence interval.

### 3.3.3. Supporting tests of interest

We found evidence for the presence of the interaction between the congruency of the current and preceding trials in the accuracy analysis; therefore, we conducted further comparisons to test whether congruency sequence effect modulated the performance on congruent, incongruent or all trials. To explore this, we tested whether the type of the preceding trial modulates the accuracy rates of current congruent and incongruent trials. To model the H1s, we used the parameters of the test of the interaction. For the comparisons, we used a one-tailed paired-sample $t$-test. In the iI and cI trial comparison, we expected the iI trials to have a higher mean accuracy rate, whereas in the cC and iC trial comparison, we expected the cC trials to have a higher mean accuracy rate. The results show that the mean accuracy of the iI trials was higher than cI trials $M = 0.05$, $B_{H(0,0.03)} = 960.63$, RR [0.0036, 14.30] ($t_{131} = 4.21$, $p < 0.01$), while we found evidence for no difference between the cC and iC trials $M = -0.0045$, $B_{H(0,0.03)} = 0.30$, RR [0.027, Inf] ($t_{131} = -0.37$, $p = 0.64$).

We analysed the mean differences in RT broken down by the congruency type (congruent, incongruent) of the preceding sentence trial. To model the H1 in the Bayes factor analysis, we used the same prior as for the Stroop main effect of the current trial (s.d. = 28). The result of the test was inconclusive with $M = 4.28$ ms, $B_{H(0,28)} = 0.53$, RR [0, 44] ($F_{1,131} = 1.97$, $p = 0.16$).

## 4. Experiment 2

In accordance with the original paper, we also tested domain-general congruency sequence effect with the non-verbal Necker cube task. In this task, participants passively watched congruent and

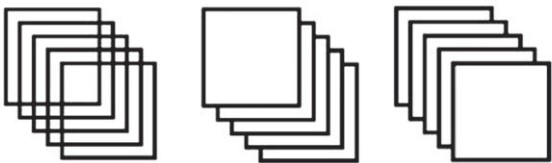

**Figure 3.** The images of the three Necker stimuli (from left to right: incongruent, congruent upward and congruent downward).

incongruent Necker cube stimuli. According to Kan *et al.* [13] and the referenced supporting literature [32,34,36], the passive observation of the bistable Necker cube induces several alterations of the cube's perceived direction. These alterations are experiences of a visual conflict only and the participants are not instructed to solve this conflict, only to indicate every alteration of the cube's mental direction. The number of direction reversals varied between individuals; therefore, the authors of the original paper hypothesized that the scale of congruency sequence effect is dependent on the size of experienced ambiguity shown by the number of reversals.

## 4.1. Materials and procedure

Participants viewed congruent and incongruent Necker stimuli pseudo-randomly intermixed with Stroop stimuli. Participants used their non-dominant hands to respond to Necker trials and their dominant hand to respond to Stroop trials. This allowed them to respond to trials without any movement across the keyboard. On incongruent Necker trials, participants indicated with button press how they perceive the direction of the cube (right-downward or left-upward) and they pressed the according button every time they experienced a direction change. On congruent Necker trials, participants were asked to indicate the direction of the stimulus as quickly as possible. The two corresponding keyboard buttons were labelled with stickers depicting the two directions of the Necker cube.

Before the experimental part, 10 Stroop and seven Necker stimuli were shown to familiarize participants with the two tasks. A baseline block was presented with 54 intermixed congruent and incongruent Stroop trials.

In the experimental block, 200 total trials were included. The above-mentioned four experimental conditions were generated by a pseudo-randomized sequence of 48 congruent Necker stimuli (24 downward, 24 upward), 28 incongruent Necker stimuli (figure 3), 62 congruent Stroop stimuli and 62 incongruent Stroop stimuli. The original study used a pseudo-randomized order of Necker and Stroop Stimuli in the experimental block. As we did not know the order of the trials in the original study, we created six blocks of pseudo-randomized sequences. Each participant was randomly assigned to one of the sequences. Each trial began with a 500 ms fixation period and it was followed by either a Stroop or a Necker stimulus. Stroop trials stayed on the screen for 1000 ms as in Experiment 1. Incongruent Necker stimuli were shown for 90 000 ms. During this period, participants were instructed to press the matching button whenever their directional interpretation changes. Congruent Necker stimuli remained on screen for 1000 ms and every trial was followed by a 1000 ms inter-trial interval.

## 4.2. Data pre-processing

We assigned participants to high and low reversal groups based on a median-split of the mean number of reversals they experienced on incongruent Necker trials. The mean number of reversals for the high reversal group was 18.61 (s.d. = 15.92; $N = 81$), while in the low reversal group, it was 3.10 (s.d. = 2.25; $N = 82$). We found a difference of 15.51 between low and high reversal groups, in comparison to the expected difference of 16 based on what was reported in the original study. We conducted the analyses only on those Stroop trials that directly followed Necker trials.

## 4.3. Results

### 4.3.1. Outcome neutral tests

We found a Stroop effect in the RT analysis with $M = 44.60$ ms, $B_{H(0,28)} = 8.08 \times 10^{18}$, RR [0.69, $1.70 \times 10^4$] ($F_{1,162} = 122.98$, $p < 0.01$).

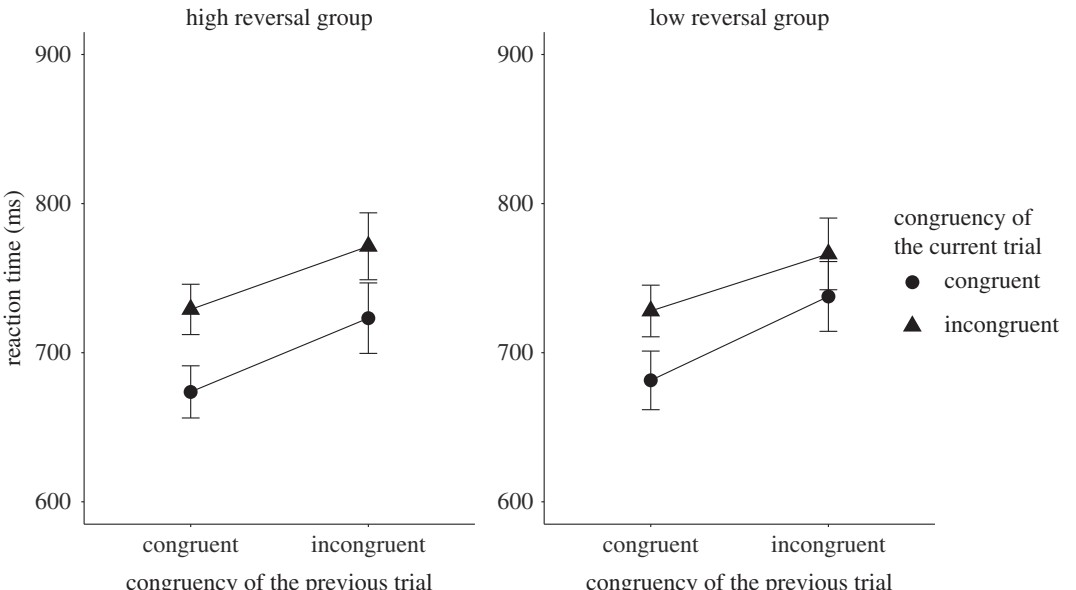

**Figure 4.** The figure shows the mean RT broken down by the congruency type of the preceding and current trials in the high and the low reversal groups for Experiment 2. The X-axis shows the congruency type of the previous trial, whereas the Y-axis shows the mean reaction times. The legend shows the congruency type of the current sentence trials. The error bars represent the 95% confidence interval.

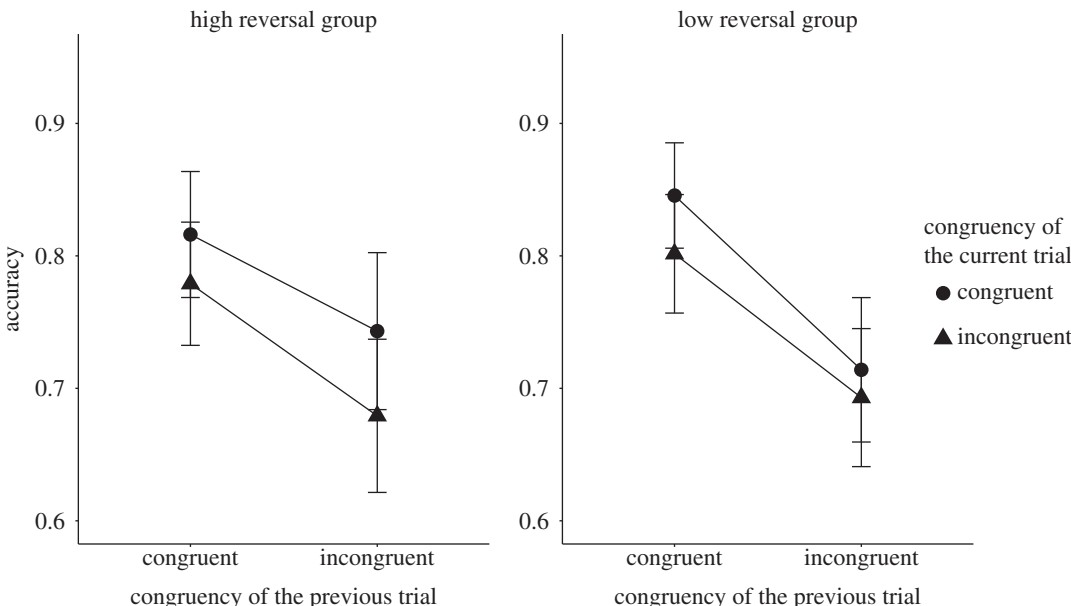

**Figure 5.** The figure shows the mean accuracy results broken down by the congruency type of the preceding and current trials in the high and the low reversal groups for Experiment 2. The X-axis shows the congruency type of the previous trial, while the Y-axis shows the accuracy. The legend shows the congruency type of the current sentence trials. The error bars represent the 95% confidence interval.

### 4.3.2. Crucial tests

The results of the two-way interactions (preceding and current trial type) were inconclusive with $M = 12.6$ ms, $B_{H(0,30)} = 1.81$, RR [0,180] ($F_{1,162} = 2.95$, $p = 0.09$) for the RT analysis and with $M = -0.0019$, $B_{H(0,0.03)} = 0.50$, RR [0, 0.05] ($F_{1,162} = 0.01$, $p = 0.92$) for the accuracy analysis. Electronic supplementary material, figures S2 and S3 show the analysis of the two-way interactions for the RT and accuracy, respectively.

We tested three-way interactions including the reversal group variable on the accuracy and RT data. The results of the RT analysis showed evidence for no effect $M = -10.90$ ms, $B_{H(0,30)} = 0.28$, RR [24.3, Inf] ($F_{1,161} = 0.55$, $p = 0.46$), while the accuracy analysis was inconclusive with $M = -0.05$, $B_{H(0,0.03)} = 0.45$, RR [0, 0.05] ($F_{1,161} = 1.82$, $p = 0.18$). Figure 4 shows the mean RT results and figure 5 shows the mean accuracy results, both broken down by the congruency type of the preceding sentence and current Stroop trials for the high and low reversal groups.

### 4.3.3. Supporting tests of interest

We tested the extent of the Stroop effect broken down by the congruency type (congruent, incongruent) of the preceding sentence trial for the RT analysis. We found the effect for the congruency of the previous trial: $M = 46.59$ ms, $B_{H(0,28)} = 1.26 \times 10^{11}$, RR [1.08, $1.73 \times 10^4$] ($F_{1,162} = 65.84$, $p < 0.01$. RTs were slower when the preceding trial type was incongruent regardless of the current trial congruency type.

Finally, as a supporting analysis, we estimated the extent to which perceived ambiguity (measured as the mean number of experienced reversals) and congruency sequence effect (iI–cI of accuracy rates and RTs) correlated. To estimate the effect size, we calculated a Pearson correlation with the 95% Bayesian credibility interval assuming a uniform prior distribution. The plausible effect sizes are small for both the RT: $r = -0.02$, 95% CI [−0.17, 0.13] and accuracy analysis: $r = 0.06$, 95% CI [−0.10, 0.21]. The correlation between the mean experienced reversals and the congruency sequence effect is shown in electronic supplementary material, figure S4 for the RT analysis and electronic supplementary material, figure S5 for the accuracy analysis.

# 5. Experiment 3

Originally, Experiment 3 was meant to show that the congruency sequence effect in the Necker cube task would not occur without the experience of internal conflict. Here, the design is identical to Experiment 2 regarding the timings and the order of trials; however, the 90 s incongruent Necker stimulus was replaced by 90 s of periodically switching congruent Necker stimuli (upward and downward). This design aimed to make the number of reversals experimentally controlled for the incongruent Necker cube trials. The authors of the original paper used the results of Experiment 2 to define the frequency of congruent Necker stimuli. In the original research, high reversal participants' average frequency of reversals was 27.6 so we showed 28 changes in direction for each 90 s period. We divided the 90 s time period into 28 time intervals with random length (min = 0.289 s; max = 7.445 s) and changed the direction of the unambiguous Necker cube in each time interval. We used the same time intervals throughout the study for all participants. Each participant saw the left facing unambiguous Necker cube first. Participants completed the same practice and the baseline sequences as in Experiment 2.

## 5.1. Data pre-processing

We assessed the level of attention to the Necker cube trials and, using the cutting point of the original study, we excluded the data of participants whose performance was below 70% on accurately identifying whether or not a stimulus change occurred on the current trial. Therefore, we excluded eight participants from further analysis, and we were left with 80 participants after all the exclusions. We included only those Stroop trials in the analyses that were preceded by a Necker trial.

## 5.2. Results

### 5.2.1. Outcome neutral tests

As an outcome neutral test, we calculated the Stroop effect of the current trial for the RT analysis. We found an effect with good enough evidence $M = 44.29$, $B_{H(0,28)} = 7.40 \times 10^7$, RR [1.5, $1.58 \times 10^4$] ($F_{1,79} = 51.74$, $p < 0.01$).

### 5.2.2. Crucial tests

We conducted two-way interactions to test the congruency sequence effect as part of the crucial tests. Both the RT analysis $M = 1.67$, $B_{H(0,30)} = 0.36$, RR [0, 32] ($F_{1,79} = 0.03$, $p = 0.87$), and the accuracy analysis $M = 0.03$, $B_{H(0,0.03)} = 1.63$, RR [0, 0.28] ($F_{1,79} = 1.47$, $p = 0.23$) yielded inconclusive results.

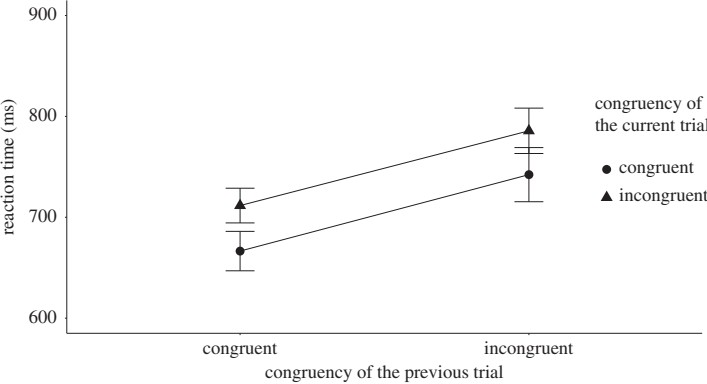

**Figure 6.** The figure shows the mean RT broken down by the congruency type of the preceding and current trials for Experiment 3. The X-axis shows the congruency type of the previous trial, while the Y-axis shows the mean reaction times. The legend shows the congruency type of the current sentence trials. The error bars represent the 95% confidence interval.

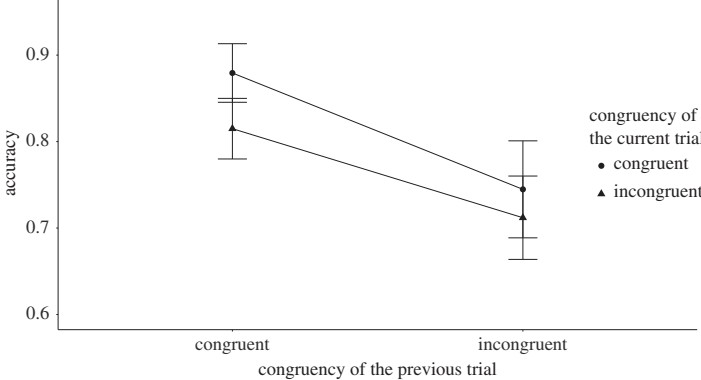

**Figure 7.** The figure shows the mean accuracy results broken down by the congruency type of the preceding and current trials for Experiment 3. The X-axis shows the congruency type of the previous trial, while the Y-axis shows the accuracy. The legend shows the congruency type of the current sentence trials. The error bars represent the 95% confidence interval.

Figure 6 shows the mean RT results and figure 7 shows the mean accuracy results of the test of the congruency sequence effect.

### 5.2.3. Supporting tests of interest

As a supporting test of interest, we tested the main effect of the congruency of the previous Necker cube trial on the mean differences in RT. We found an effect with $M = 74.93$ ms, $B_{H(0,28)} = 1.99 \times 10^{12}$, RR [1.8, 27740] ($F_{1,79} = 98.12$, $p < 0.01$)

## 6. Discussion

The present replication study investigated the domain-generality of the congruency sequence effect. By replicating the design of Kan *et al.* [13], we expected that the results would reveal the strength of evidence behind a prominent claim of domain-generality.

We collected data from three countries with nearly four times the sample size of the original study for Experiment 1, where Stroop task trials followed garden-path sentences. The data showed that the participants complied with the instructions, reflected by the presence of the Stroop effect and the longer RTs in the ambiguous regions of the incongruent sentences compared to the same region of the congruent sentences. The crucial tests brought partial support for the cross-task congruency sequence effect. While the RT results were inconclusive, nearly reaching our threshold for evidence for no effect, the accuracy results were in line with the original finding as the Bayesian analyses showed evidential support for the effect. Follow-up analyses showed that this finding was the result of

**Table 1.** Summary of our findings. Note: ✓, support for the presence of the congruency sequence effect (CSE); ✗, support against the presence of the CSE; —, inconclusive findings.

| experiment | CSE present for | |
| --- | --- | --- |
| | RT | accuracy |
| Exp. 1 | — | ✓ |
| Exp. 2 | — | — |
| Exp. 2 (reversals) | ✗ | — |
| Exp. 3 | — | — |

increased accuracy on incongruent trials when they came after incongruent sentences compared to when they came after congruent ones.

In Experiment 2, where the sentence task was replaced by Necker cubes in the design, we collected data from more than six times the original sample size. Still, the results remained inconclusive for both the RT and the accuracy tests of the congruency sequence effect. Furthermore, we found good enough evidence for no effect in the RT results when we tested whether perceiving more Necker cube reversals led to a greater congruency sequence effect.

In Experiment 3, where the orientation of the Necker cube was periodically switched by the testing programme, the analyses drove inconclusive results. The RT and accuracy data did not support the presence of the cross-task congruency sequence effect. In fact, the RT results nearly passed our preset evidential threshold for no effect. Originally, this experiment was intended to demonstrate the lack of effect since there would be no internal representational conflict experienced by the participants. Since in this replication, we found no evidence for congruency sequence effect in Experiment 2, this experiment adds relatively little to our primary aims.

Summarizing our findings, first we can conclude that our replication was successfully implemented in accordance with our plan. Our outcome neutral tests provide strong evidence to suggest that our participants understood and followed the instructions and they produced the expected default Stroop effect. Second, we found that our crucial tests brought only partial support for the claims of the original study. However, as table 1 shows, with only a single exception, we could not replicate the original findings. The data remained inconclusive or went against the hypothesis.

The results are surprising from several aspects. Should the original results not be chance findings, one would expect more supportive evidence across three experiments, particularly given our increased sample size. Bayes factor analyses indicate that the data, with one exception, were far from supporting the alternative hypothesis. Results frequently provided support for the null hypothesis.

We consider the evidence for congruency sequence effect in the accuracy results of Experiment 1 of particular interest. The theoretical interest in this supportive finding is whether the observed effect in the garden-path task can indicate the cross-domain nature of the congruency sequence effect. That is, whether readers' detection of syntactic conflict enhances cognitive control in another domain. As we discussed it in Introduction, the domain-specificity account assumes that the overlap of stimulus dimensions between the conflict tasks is a central determinant of the transfer of control [5,8,12]. Should we regard the sentence-processing task and the Stroop task to differ in their domain (syntactic versus non-syntactic) then our results of Experiment 1 provide partial evidence that control adaptation can work across domains. The interpretation is weaker if we regard the two to be more similar in nature. The original authors themselves pointed out that despite the difference in stimulus characteristics, both tasks are verbal. In fact, this motivated them to test the effect between perceptual and verbal domains in Experiments 2 and 3. If the effect is domain-general, then it remains a question regarding why the incongruent Necker cube did not decrease the congruency effect. One possibility is that the ambiguity in the Necker cube does not lead to cognitive conflict; therefore, it is not a good test of the effect. Either way, we consider that the positive result of Experiment 1 is noteworthy for the theory and it, together with the lack of support for the null in other analyses, prevents us from completely rejecting the domain-general hypothesis of the congruency sequence effect. In fact, the results are also somewhat in-line with the finding of a new study that used Flanker-trials and code-switch manipulation on sentences [37]. They have found evidence of an RT interaction between previous sentence type and current Flanker congruency consistent with control adaptation.

One limitation of this study is that it replicated the experiments on a different population and due to the anonymous data collection, we could not compare the demographics of the original sample to the replication sample.

The results brought up some new questions for further investigations. First of all, it remains unexplained why RT data of Experiment 1 were in sharp contrast with the original findings, being very far from detecting the congruency sequence effect between the two tasks. One could speculate that enhanced control after incongruent sentence trials manifested in participants' higher motivation for accuracy which could increase the time needed to spend on the incongruent Stroop trials, counteracting any speeding effect. However, our pattern of results does not support this speculation, as we measured a lower accuracy rate than the original study and after the incongruent trials, the mean RTs are not slower than after the congruent trials. Alternatively, if the effect is domain-specific, then the positive accuracy results seek an explanation of a different scope.

It is worth noting that investigating the evidence for the cross-task congruency sequence effect is interesting not just for the domain-generality debate, but it is also relevant to the research on the mechanisms of cognitive control. According to the *Conflict Monitoring Theory* [3], the detection of conflict is what triggers subsequent cognitive control. By contrast, the *Affective Signaling Hypothesis* [38] suggests that it is not conflict *per se* that is responsible for control adaptation, but the negative affect that this conflict triggers. In this view, the experience of conflict evokes a negative affective reaction and this affect is what elicits the performance-monitoring system to upgrade control resources leading to the attenuation of further conflict. From this alternative framework, it is plausible that reading incongruent garden-path sentences causes phasic negative affect, so we would expect control adaptation on subsequent trials. It is less clear, however, whether watching incongruent Necker cubes is a source of negative affect. It is possible that the lack of elicited negative affect in that design is behind the lack of evidence for the congruency sequence effect in Experiment 2. Here, it is relevant to note that our results should not be generalized to other experimental arrangements. It is possible that different designs can be more sensitive to detect the cross-task congruency sequence effect [37,39–41]. Future research should consider applying cross-task designs to test the predictions of the *Affective Signaling Hypothesis*.

In sum, this study provides a registered replication of three experiments testing the congruency sequence effect between two tasks. We could not replicate the original positive results, except in one measure in Experiment 1. In our interpretation, this pattern of findings weakens, but does not fully reject the hypothesis that the effect can occur across domains. We suggest that failure to detect control adaptation after the Necker cube might reflect that the two designs differ not just in the domain of the trigger trial but also in other characteristics, such as the affect the trigger trials induce, drawing attention to alternative frameworks of control adaptation.

Ethics. Each laboratory ascertains that the local institutional ethical review board agrees with the proposed data collection. This replication was conducted in accordance with the Declaration of Helsinki. We did not collect any identifiable private data during the project.

Data accessibility. All of our analyses were publicly preregistered on the OSF site after Stage 1 'in principle' acceptance. Collected raw and processed data are publicly shared on the OSF page of the project. Code for data management and statistical analyses were written in R and are open access. All materials of the three experiments are available through OSF: https://osf.io/6bd43/.

Authors' contributions. B.A., M.K. and B.P.: conceptualization. M.K. and M.B.: data curation. M.K. and B.P.: formal analysis. B.A., M.K., M.B., A.H., S.O., L.E.T. and T.R.E.: investigation. B.A., M.K., B.P. and M.B.: methodology. B.A. and M.B.: project administration. B.A., M.K. and M.B.: resources. M.K. and M.B.: software. B.A.: supervision. M.K. and B.P.: validation. M.K.: visualization. B.A., M.K., B.P. and M.B.: writing—original draft preparation. B.A., M.K., B.P., M.B., A.H., S.O., L.E.T. and T.R.E.: writing—review and editing.

Competing interests. We declare we have no competing interests.

Funding. B.A. was supported by the János Bolyai Research Fellowship from the Hungarian Academy of Sciences.

Acknowledgements. We are grateful to our four reviewers for their constructive comments throughout the review of our manuscript.

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
