## [Peer Review File · Royal Society Open Science]

Review History

RSOS-190749.R0 (Original submission)

Review form: Reviewer 1 (Friederike Schlaghecken)

Do you have any ethical concerns with this paper?

No

Have you any concerns about statistical analyses in this paper?

No

Recommendation?

Accept with minor revision

Comments to the Author(s)

Review RSOS-190749

The authors propose to carry out an exact replication of the three experiments reported in Kan et al. (2013), increasing participant numbers by a factor of 2.5. The proposed study seems sound from a technical perspective.

From a theoretical perspective, however, I am less convinced. The original study finds that Stroop effects are smaller following ambiguous material (garden path sentences, Necker cubes) than following unambiguous material. This is interpreted as a 'conflict adaptation' effect. Alternatively, however, this reduction might be an after-effect of the greater effort needed to process ambiguous material, regardless of any 'conflict' it presents.

The original study contains a control experiment (Exp 3) to distinguish between these two interpretations (conflict adaptation effect versus effort after-effect) for the Necker cube task (Exp 2) only. The success of this is limited (Exp 3 produces a pattern of result similar to that produced by the 'low reversal' group in Exp 2). Importantly, the study does not contain a control experiment for the sentence task (Exp 1).

I strongly recommend that the present authors close this gap by adding the missing sentence-task control experiment. An obvious way of doing so would be to use non-ambiguous sentences in normal and in disfluent font (e.g., sans forvetica or some other distorted font that reduces reading speed to the same extent as garden path sentences do). In my opinion, investigating the influence of pure reading effort on the Stroop effect might provide a more clear-cut way to distinguish between the 'conflict adaptation' and the 'effort after-effect' hypotheses.

Signed

Friederike Schlaghecken

Review form: Reviewer 2

Do you have any ethical concerns with this paper?

No

Have you any concerns about statistical analyses in this paper?

No

Recommendation?

Accept with minor revision

Comments to the Author(s)

This looks to be a valuable replication of a key paper demonstrating causal relationships between cognitive control engagement processes across syntactic, non-syntactic, verbal, and perceptual domains.

One aspect that I would like to see included (and that I don't think I saw in this version) is some additional description/clarification of the nature of the ambiguities across the multiple languages, as well as some comparison (if at all possible) or their similarity to the ones used in the Kan et al., study.

Review form: Reviewer 3 (Craig Hedge)

Do you have any ethical concerns with this paper?

No

Have you any concerns about statistical analyses in this paper?

I do not feel qualified to assess the statistics

Recommendation?

Accept with minor revision

Comments to the Author(s)

Summary

The manuscript is a stage 1 replication of Kan et al's (2013) paper "To adapt or not to adapt: The question of domain-general cognitive control". The original paper and proposed replication consist of three studies designed to test whether cognitive control adaptations are domain-general. The first study consists of participants performing intermixed trials of a sentence comprehension and Stroop tasks. In both tasks, stimuli could either be unambiguous/congruent or ambiguous/incongruent. The outcome of interest is whether there is an interaction between (current) Stroop trial congruency and (preceding) sentence comprehension trial congruency. Experiments 2 and 3 follow a similar logic, with the sentence comprehension task being replaced by a perceptual task (Necker cube viewing). An unambiguous Necker Cube stimulus has a clear directionality, whereas ambiguous Necker Cube stimulus can be perceived to go in one of two directions, and views may flip between them during passive viewing. Experiment 3 in the original study replaced the ambiguous Necker stimuli with alternating unambiguous stimuli to assess the importance of "internal representational conflict". The original study observed conflict adaptation (reduced Stroop effect following ambiguous preceding trials) in Experiments 1 and 2, but not 3. This replication study closely follows the design of the original, while targeting larger sample sizes.

Review

My opinion on this manuscript is generally positive, and I think the criteria for a Stage 1 can be met with minor revisions. The extent to which cognitive control processes are domain-general is an ongoing question in the literature. The proposed replication is potentially informative both from a reproducibility perspective, as well in terms of the broader theoretical questions. As a replication, the proposal seems strong. It closely follows the method of the original study, and benefits from an increased sample size and clear stopping rule.

I think that the manuscript would benefit from more detail on a few points, as the rationale/method were not entirely clear without reading the original paper. I also have a question regarding how the numerous critical tests will be interpreted with respect to both the original findings and the overarching theoretical question (point 3 below).

1) The authors describe and reference the statistical approach for the critical tests, but it may be useful to go through the steps in more detail in the manuscript. The authors state on page 5: *"We performed and reported all of our hypothesis tests as either paired or independent t tests for the sake of simplicity."*

The data analyses sections state that the critical analyses are two-way (Exps. 1 & 3) and three-way (Exp 2) interactions. I am not an expert on Dienes' approach, though my understanding is that these interactions can be reduced to something like a t-test per his 2014 paper. It may not be intuitive to a naïve reader how this is done, though, particularly with the three-way interaction. In order to aid comparisons with the frequentist ANOVAs in the original paper, perhaps the working and Bayes factors could be shown for the original results if sufficient information is given?

2) The performance cut-offs may warrant consideration. My reading of the original paper is that they report the performance of the participants that were excluded, but I couldn't see the threshold that they applied. If the cut-offs used in the original are not known, the authors may want to consider whether those selected are optimal from a data quality perspective. In particular:

-Page 7 of the replication states:

"using the cutting point of the original study, we excluded the data of participants whose performance was below 62% on accurately identifying whether a stimulus change occurred on the current trial."

Page 646 of the original paper states:

"One subject's data were removed from analyses because of poor performance on this straightforward task (61%)"

It is possible that the original study applied a more conservative cut-off (e.g. >70%), and no other participants fell below it.

-My understanding is that there are no exclusion criteria biased on Stroop task performance? My reading of the original manuscript is that they did not exclude anyone on this basis, but again this may be because performance did not cross an unstated threshold. As performance cut-offs are applied to other tasks, it may be appropriate to apply them here too.

-The authors state that they will exclude participants who are at chance (50%) in the sentence comprehension task in Study 1. Is <50% the threshold to be used (i.e. will participants be retained if they are at 51%)? The original paper states that the remaining participants scored *"at or above 70%"* (pg 640).

3) As in the original paper, the analysis plan for the replication consists of multiple experiments, each of which has multiple critical and supporting tests. This complexity makes it difficult to specify a single condition by which to say that the original findings were replicated. However, there is a range of possible outcomes between replicating all or none of the effects observed in the original paper.

In terms of the theoretical question of whether control adaptations are domain-general, I wondered if the authors could be more explicit in advance about what outcomes would be compatible or incompatible with this account. For example, I can imagine some combination of findings that do not have a straightforward interpretation:

- If H1 is supported in experiments 2 & 3, but not experiment 1
- If H1 is supported in the critical tests on reaction time but not in accuracy (or vice versa, which could also be inconsistent across the experiments)
- If H1 is supported in the critical test on (e.g.) reaction time, but the supporting tests show that it comes from an increase in congruent RTs and no change in incongruent RTs.

As above, I appreciate that mapping all the possible combinations of outcomes on to explanations is infeasible. However, it wasn't clear to me then whether the authors were going to consider every critical test individually, or whether a broader conclusion was going to be drawn based on a minimum set of conditions.

4) The authors could give more details regarding the calculation of residual reading time (e.g. if it involves a per-subject regression).

5) The authors give three reasons why they chose to conduct a direct replication of Kan et al. (pg 3). While these seemed like a reasonable justification, I was curious whether any other papers were also evaluated.

6) The authors state that they will apply a threshold of 3 (/0.33) as a stopping rule for their Bayes factors. I note that several journals require more conservative thresholds for registered reports (e.g. Cortex & BMJ Open science require >6). I don't object to the threshold of 3 chosen here, but it is perhaps worth further qualification or commenting that there isn't universal agreement on a 'good enough' threshold.

7) It would be useful to include a diagram of the Necker Cube stimuli.

8) A minor point - on page 5 the authors state:

"Hence, our analyses were identical to those of the original paper".

I understand this to mean that the level of aggregation they describe previously is identical, but it may also be interpreted that the original paper used the same Bayesian approach that the authors go on to discuss. Perhaps this discrepancy could be more explicit here.

9) On page 3:

"All participants performed two tasks: first a Stroop-test, then either a sentence processing task (Experiment

1), or a perceptual processing task (Experiment 2, 3)."

It may be more appropriate to say the sentence/perceptual tasks first, as they preceded the Stroop trials?

10) On page 3, I was unsure what the authors meant when they refer to "*various NoGo decision criteria*" used by Feldman et al. (2015).

Signed,

Craig Hedge

Review form: Reviewer 4 (Irene Kan)

Do you have any ethical concerns with this paper?

No

Have you any concerns about statistical analyses in this paper?

Yes

Recommendation?

Accept with minor revision

Comments to the Author(s)

Introduction:

Page 1, line 41: The paper will be strengthened if a more thorough discussion of the proposed mechanisms of conflict adaptation. The authors wrote, "It is assumed that in these tasks the detection of conflict enhances cognitive control resulting in improved conflict resolution of subsequent trials. If the presence of conflict is sufficient for this conflict adaptation effect, then conflict adaptation can be expected to occur in cross-domain tasks."

What is it that is getting “passed” between consecutive trials? What is adaptation thought to reflect? Even for studies that do find adaptation, not all agree that conflict is a critical ingredient (see work by Schmidt, Weissman, etc.), even if they do believe that cognitive control is necessary for the effect to emerge. For example, are people biasing attention to task-relevant over task-irrelevant cues in the cross-task paradigm? Such a state in one trial of one task could sustain such that one is ready to bias attention on the next trial too, even if it is another task entirely (one won’t know how to bias attention accordingly until one knows what task she’s in). Does some central conflict monitor/resolver have to have access to the representations for each task, so it “knows” how to resolve them? Or does this theoretically general mechanism not have to have such access to the representations? If not, how is reactive control achieved?

The second sentence in the above quotation is opaque. I think the authors are trying to make contact with the different theories behind what triggers such behavioral adjustments (sometimes referred to as the congruence sequence effect), however, it did not come across as such.

Page 3, line 13: The authors wrote, “Since the Conflict Monitoring Theory (3), posits that the detection of conflict triggers cognitive control mechanisms, these mechanisms can be studied by comparing performance on the incongruent trials with responses on congruent trials. Greater cognitive control is thought to reflect less observed differences in congruent and incongruent trials.”

It is unclear what is meant by “less observed differences in congruent and incongruent trials”. Again, a more detailed description of the connection to how behavior is regulated in such paradigms should be unpacked a bit more. (Relatedly, it is unclear what specific mechanisms the authors are referring to when they wrote “cognitive control mechanisms”).

Page 3, line 44: The authors wrote, “For example, the sentence: “The basketball player accepted the contract would have to be negotiated”, is ambiguous without reinterpreting the sentence after it is read in its entirety.” This description is a bit confusing. It is not ambiguous without reinterpreting (that is not the source of the syntactic ambiguity). The sentence is ambiguous because “the contract” could temporarily be the verb’s direct object or the subject of an ensuing clause. English readers are biased to commit to the direct object interpretation, because “accepted” frequently appears in such syntactic constructions; but the sentence doesn’t turn out that way. In other words, the temporary conflict arose due to such syntactic biases. Readers realize this upon encountering “would have”, which is incompatible with the direct object analysis, which they then have to abandon through some conflict resolution mechanism (theoretically).

Page 3: The rationale of the present paper would be clearer if the authors had elaborated on how the replication attempt would add to our understanding of the domain-generalty of cognitive control. It will be important and instructive to identify the theoretical contribution of this replication attempt.

Relatedly, the authors identified limitations of the studies that reported null effects (i.e., absence of evidence is not evidence for absence) and suggested that those prior studies may have lacked sensitivity. But a replication attempt of previous positive findings does not address the issue of sensitivity in prior studies that reported null findings.

Methods & Procedures:

Page 4: College students participated in the original studies. What is the demographic of the participants for this project?

Pages 4, 5 & 7 (related to number of subjects and data analysis approach): The authors will be approaching the analyses with a Bayesian perspective, while the original studies utilized the more traditional approach. Given recent interests in the inferences associated with the different analysis approaches, this difference is reasonable. However, the rationale for the authors' choice should be justified.

Furthermore, please provide an explanation of the optional stopping rule. The rationale is not obvious, and it would be instructive for the authors to explain why the optional stopping approach is acceptable/unbiased. Relatedly, why 2.5 times the original sample sizes?

Page 5, lines 7 and 8: The authors wrote, "For Experiment 2 and 3, we tested Hungarian native speakers as the stimuli were not language specific." To clarify, were the color names presented in the Hungarian speakers' native language? While I agree with the authors that the perceptual task (Necker cube) is less likely to be language-sensitive, the magnitude of the Stroop task is likely to be influenced by the subjects' language fluency. In sum, if English color names were used, the authors should provide details about subjects' fluency in English.

Page 5: Please cite Garnsey et al. (1997) when referencing the ambiguous/unambiguous sentence materials, as they were the original source of these sentences.

Garnsey, S. M., Pearlmutter, N. J., Myers, E., & Lotocky, M. A. (1997). The contributions of verb bias and plausibility to the comprehension of temporarily ambiguous sentences. *Journal of Memory and Language*, 37, 58-93. <http://dx.doi.org/10.1006/jmla.1997.2512>.

Minor methodological differences:

Some minor methodological differences were noted. Although we do not believe these differences to contribute meaningfully to the overall results, their choices should be noted or explained.

Page 4, line 52: If purple was one of the incongruent colors, this differs from the original experiment, where incongruent colors were brown, orange and red.

Page 5, line 9: Participants in the original study received 145 baseline Stroop trials rather than 165.

Page 6, line 3: The original effect size for the interaction between current and preceding Stroop congruency on the RT data was 29 ms. The authors report it as 30 ms.

Clarifications needed:

Page 5, line 11: How many practice trials were included?

Page 6, line 50: Please explain the task requirement for the congruent Necker trials.

Page 6, line 60: Please describe how subject group assignment was made. Median split was used in the original study.

Page 7, lines 29 - 34: In the original study, all but one subject achieved near ceiling performance (mean accuracy = 96%, SD = 3%) performed at ceiling. The outlier was only 61% correct on the task, and we excluded that subject's data from analyses, with the assumption that the unusually low task performance suggests a lack of engagement. Thus, to describe 62% as a "cut-off" score is not wholly accurate.

Minor comments:

Page 3, line 47: "The four conditions were determined by the conflict state of the preceding stimulus..." was not an entirely accurate description. The four conditions were determined by the status of preceding and current stimuli.

Page 4, line 21: "They read the sentences word-by-word as they stepped to the next word by pressing the keyboard spacebar." The wording is slightly awkward; we suggest "...word-by-word, advancing to the next word..."

Decision letter (RSOS-190749.R0)

03-Jun-2019

Dear Dr Aczel,

The Editors assigned to your Stage 1 Replication submission ("Is there evidence for cross-domain control adaptation? A replication of Kan et al. (2013)") have now received comments from reviewers. We would like you to revise your paper in accordance with the referee and editors suggestions which can be found below (not including confidential reports to the Editor). Please note this decision does not guarantee eventual acceptance.

Please submit a copy of your revised paper within three weeks (i.e. by the 25-Jun-2019). If deemed necessary by the Editors, your manuscript will be sent back to one or more of the original reviewers for assessment.

When submitting your revised manuscript, you must respond to the comments made by the referees and upload a file "Response to Referees" in the "File Upload" step. Please use this to document how you have responded to the comments, and the adjustments you have made. In order to expedite the processing of the revised manuscript, please be as specific as possible in your response.

Once again, thank you for submitting your manuscript to Royal Society Open Science and I look forward to receiving your revision. If you have any questions at all, please do not hesitate to get in touch. Full author guidelines may be found at <http://rsos.royalsocietypublishing.org/page/replication-studies#AuthorsGuidance>.

Kind regards,
Professor Chris Chambers
Registered Reports Editor
Royal Society Open Science
openscience@royalsociety.org

Editor Comments to Author (Professor Chris Chambers):

Four expert reviewers have now assessed the manuscript. The reviews are very high quality and guardedly positive, with three reviewers judging that the Stage 1 primary criteria are currently

met. All reviewers, however, point out a range of issues to be addressed in revision. Reviewer 1 notes that the study would benefit from an additional control experiment. Where this goes beyond the experiments reported in the original study, it is not necessary to conduct such an experiment to achieve Stage 1 acceptance. However if the authors agree, then an experiment that goes beyond the replication could be added to the design under special circumstances, provided it is clearly flagged as in addition to the target study. Reviewers 3 and 4 note a range of areas that would benefit from expansion and clarification, including the clarity of the introduction, use of terminology, and analysis plans. In relation to Reviewer 4's comment: "The rationale of the present paper would be clearer if the authors had elaborated on how the replication attempt would add to our understanding of the domain-generalty of cognitive control. It will be important and instructive to identify the theoretical contribution of this replication attempt." -- note that under the Replications policy, a theoretical justification for replication is not necessary to achieve Stage 1 acceptance, although the authors are welcome to respond positively to this suggestion if they wish. Reviewers 3 and 4 also note some methodological deviations (or potential deviations) between the replication and the original experiments. These should be carefully addressed.

Comments to Author:

Reviewer: 1

Comments to the Author(s)

Review RSOS-190749

The authors propose to carry out an exact replication of the three experiments reported in Kan et al. (2013), increasing participant numbers by a factor of 2.5. The proposed study seems sound from a technical perspective.

From a theoretical perspective, however, I am less convinced. The original study finds that Stroop effects are smaller following ambiguous material (garden path sentences, Necker cubes) than following unambiguous material. This is interpreted as a 'conflict adaptation' effect. Alternatively, however, this reduction might be an after-effect of the greater effort needed to process ambiguous material, regardless of any 'conflict' it presents.

The original study contains a control experiment (Exp 3) to distinguish between these two interpretations (conflict adaptation effect versus effort after-effect) for the Necker cube task (Exp 2) only. The success of this is limited (Exp 3 produces a pattern of result similar to that produced by the 'low reversal' group in Exp 2). Importantly, the study does not contain a control experiment for the sentence task (Exp 1).

I strongly recommend that the present authors close this gap by adding the missing sentence-task control experiment. An obvious way of doing so would be to use non-ambiguous sentences in normal and in disfluent font (e.g., sans forvetica or some other distorted font that reduces reading speed to the same extent as garden path sentences do). In my opinion, investigating the influence of pure reading effort on the Stroop effect might provide a more clear-cut way to distinguish between the 'conflict adaptation' and the 'effort after-effect' hypotheses.

Signed

Friederike Schlaghecken

Reviewer: 2

Comments to the Author(s)

This looks to be a valuable replication of a key paper demonstrating causal relationships between cognitive control engagement processes across syntactic, non-syntactic, verbal, and perceptual domains.

One aspect that I would like to see included (and that I don't think I saw in this version) is some additional description/clarification of the nature of the ambiguities across the multiple

languages, as well as some comparison (if at all possible) or their similarity to the ones used in the Kan et al., study.

Reviewer: 3

Comments to the Author(s)

Summary

The manuscript is a stage 1 replication of Kan et al's (2013) paper "To adapt or not to adapt: The question of domain-general cognitive control". The original paper and proposed replication consist of three studies designed to test whether cognitive control adaptations are domain-general. The first study consists of participants performing intermixed trials of a sentence comprehension and Stroop tasks. In both tasks, stimuli could either be unambiguous/congruent or ambiguous/incongruent. The outcome of interest is whether there is an interaction between (current) Stroop trial congruency and (preceding) sentence comprehension trial congruency. Experiments 2 and 3 follow a similar logic, with the sentence comprehension task being replaced by a perceptual task (Necker cube viewing). An unambiguous Necker Cube stimulus has a clear directionality, whereas ambiguous Necker Cube stimulus can be perceived to go in one of two directions, and views may flip between them during passive viewing. Experiment 3 in the original study replaced the ambiguous Necker stimuli with alternating unambiguous stimuli to assess the importance of "internal representational conflict". The original study observed conflict adaptation (reduced Stroop effect following ambiguous preceding trials) in Experiments 1 and 2, but not 3. This replication study closely follows the design of the original, while targeting larger sample sizes.

Review

My opinion on this manuscript is generally positive, and I think the criteria for a Stage 1 can be met with minor revisions. The extent to which cognitive control processes are domain-general is an ongoing question in the literature. The proposed replication is potentially informative both from a reproducibility perspective, as well in terms of the broader theoretical questions. As a replication, the proposal seems strong. It closely follows the method of the original study, and benefits from an increased sample size and clear stopping rule.

I think that the manuscript would benefit from more detail on a few points, as the rationale/method were not entirely clear without reading the original paper. I also have a question regarding how the numerous critical tests will be interpreted with respect to both the original findings and the overarching theoretical question (point 3 below).

1) The authors describe and reference the statistical approach for the critical tests, but it may be useful to go through the steps in more detail in the manuscript. The authors state on page 5: *"We performed and reported all of our hypothesis tests as either paired or independent t tests for the sake of simplicity."*

The data analyses sections state that the critical analyses are two-way (Exps. 1 & 3) and three-way (Exp 2) interactions. I am not an expert on Dienes' approach, though my understanding is that these interactions can be reduced to something like a t-test per his 2014 paper. It may not be intuitive to a naïve reader how this is done, though, particularly with the three-way interaction. In order to aid comparisons with the frequentist ANOVAs in the original paper, perhaps the working and Bayes factors could be shown for the original results if sufficient information is given?

2) The performance cut-offs may warrant consideration. My reading of the original paper is that they report the performance of the participants that were excluded, but I couldn't see the

threshold that they applied. If the cut-offs used in the original are not known, the authors may want to consider whether those selected are optimal from a data quality perspective. In particular:

-Page 7 of the replication states:

"using the cutting point of the original study, we excluded the data of participants whose performance was below 62% on accurately identifying whether a stimulus change occurred on the current trial."

Page 646 of the original paper states:

"One subject's data were removed from analyses because of poor performance on this straightforward task (61%)"

It is possible that the original study applied a more conservative cut-off (e.g.>70%), and no other participants fell below it.

-My understanding is that there are no exclusion criteria biased on Stroop task performance? My reading of the original manuscript is that they did not exclude anyone on this basis, but again this may be because performance did not cross an unstated threshold. As performance cut-offs are applied to other tasks, it may be appropriate to apply them here too.

-The authors state that they will exclude participants who are at chance (50%) in the sentence comprehension task in Study 1. Is <50% the threshold to be used (i.e. will participants be retained if they are at 51%)? The original paper states that the remaining participants scored *"at or above 70%"* (pg 640).

3) As in the original paper, the analysis plan for the replication consists of multiple experiments, each of which has multiple critical and supporting tests. This complexity makes it difficult to specify a single condition by which to say that the original findings were replicated. However, there is a range of possible outcomes between replicating all or none of the effects observed in the original paper.

In terms of the theoretical question of whether control adaptations are domain-general, I wondered if the authors could be more explicit in advance about what outcomes would be compatible or incompatible with this account. For example, I can imagine some combination of findings that do not have a straightforward interpretation:

- If H1 is supported in experiments 2 & 3, but not experiment 1
- If H1 is supported in the critical tests on reaction time but not in accuracy (or vice versa, which could also be inconsistent across the experiments)
- If H1 is supported in the critical test on (e.g.) reaction time, but the supporting tests show that it comes from an increase in congruent RTs and no change in incongruent RTs.

As above, I appreciate that mapping all the possible combinations of outcomes on to explanations is infeasible. However, it wasn't clear to me then whether the authors were going to consider every critical test individually, or whether a broader conclusion was going to be drawn based on a minimum set of conditions.

4) The authors could give more details regarding the calculation of residual reading time (e.g. if it involves a per-subject regression).

5) The authors give three reasons why they chose to conduct a direct replication of Kan et al. (pg 3). While these seemed like a reasonable justification, I was curious whether any other papers were also evaluated.

6) The authors state that they will apply a threshold of 3 (/0.33) as a stopping rule for their Bayes factors. I note that several journals require more conservative thresholds for registered reports (e.g. Cortex & BMJ Open science require >6). I don't object to the threshold of 3 chosen here, but it is perhaps worth further qualification or commenting that there isn't universal agreement on a 'good enough' threshold.

7) It would be useful to include a diagram of the Necker Cube stimuli.

8) A minor point - on page 5 the authors state:

"Hence, our analyses were identical to those of the original paper".

I understand this to mean that the level of aggregation they describe previously is identical, but it may also be interpreted that the original paper used the same Bayesian approach that the authors go on to discuss. Perhaps this discrepancy could be more explicit here.

9) On page 3:

"All participants performed two tasks: first a Stroop-test, then either a sentence processing task (Experiment

1), or a perceptual processing task (Experiment 2, 3)."

It may be more appropriate to say the sentence/perceptual tasks first, as they preceded the Stroop trials?

10) On page 3, I was unsure what the authors meant when they refer to "*various NoGo decision criteria*" used by Feldman et al. (2015).

Signed,
Craig Hedge

Reviewer: 4
Comments to the Author(s)

Introduction:

Page 1, line 41: The paper will be strengthened if a more thorough discussion of the proposed mechanisms of conflict adaptation. The authors wrote, "It is assumed that in these tasks the detection of conflict enhances cognitive control resulting in improved conflict resolution of subsequent trials. If the presence of conflict is sufficient for this conflict adaptation effect, then conflict adaptation can be expected to occur in cross-domain tasks."

What is it that is getting "passed" between consecutive trials? What is adaptation thought to reflect? Even for studies that do find adaptation, not all agree that conflict is a critical ingredient (see work by Schmidt, Weissman, etc.), even if they do believe that cognitive control is necessary for the effect to emerge. For example, are people biasing attention to task-relevant over task-irrelevant cues in the cross-task paradigm? Such a state in one trial of one task could sustain such that one is ready to bias attention on the next trial too, even if it is another task entirely (one won't know how to bias attention accordingly until one knows what task she's in). Does some central conflict monitor/resolver have to have access to the representations for each task, so it "knows" how to resolve them? Or does this theoretically general mechanism not have to have such access to the representations? If not, how is reactive control achieved?

The second sentence in the above quotation is opaque. I think the authors are trying to make

contact with the different theories behind what triggers such behavioral adjustments (sometimes referred to as the congruence sequence effect), however, it did not come across as such.

Page 3, line 13: The authors wrote, “Since the Conflict Monitoring Theory (3), posits that the detection of conflict triggers cognitive control mechanisms, these mechanisms can be studied by comparing performance on the incongruent trials with responses on congruent trials. Greater cognitive control is thought to reflect less observed differences in congruent and incongruent trials.”

It is unclear what is meant by “less observed differences in congruent and incongruent trials”. Again, a more detailed description of the connection to how behavior is regulated in such paradigms should be unpacked a bit more. (Relatedly, it is unclear what specific mechanisms the authors are referring to when they wrote “cognitive control mechanisms”).

Page 3, line 44: The authors wrote, “For example, the sentence: “The basketball player accepted the contract would have to be negotiated”, is ambiguous without reinterpreting the sentence after it is read in its entirety.” This description is a bit confusing. It is not ambiguous without reinterpreting (that is not the source of the syntactic ambiguity). The sentence is ambiguous because “the contract” could temporarily be the verb’s direct object or the subject of an ensuing clause. English readers are biased to commit to the direct object interpretation, because “accepted” frequently appears in such syntactic constructions; but the sentence doesn’t turn out that way. In other words, the temporary conflict arose due to such syntactic biases. Readers realize this upon encountering “would have”, which is incompatible with the direct object analysis, which they then have to abandon through some conflict resolution mechanism (theoretically).

Page 3: The rationale of the present paper would be clearer if the authors had elaborated on how the replication attempt would add to our understanding of the domain-generalty of cognitive control. It will be important and instructive to identify the theoretical contribution of this replication attempt.

Relatedly, the authors identified limitations of the studies that reported null effects (i.e., absence of evidence is not evidence for absence) and suggested that those prior studies may have lacked sensitivity. But a replication attempt of previous positive findings does not address the issue of sensitivity in prior studies that reported null findings.

Methods & Procedures:

Page 4: College students participated in the original studies. What is the demographic of the participants for this project?

Pages 4, 5 & 7 (related to number of subjects and data analysis approach): The authors will be approaching the analyses with a Bayesian perspective, while the original studies utilized the more traditional approach. Given recent interests in the inferences associated with the different analysis approaches, this difference is reasonable. However, the rationale for the authors’ choice should be justified.

Furthermore, please provide an explanation of the optional stopping rule. The rationale is not obvious, and it would be instructive for the authors to explain why the optional stopping approach is acceptable/unbiased. Relatedly, why 2.5 times the original sample sizes?

Page 5, lines 7 and 8: The authors wrote, “For Experiment 2 and 3, we tested Hungarian native speakers as the stimuli were not language specific.” To clarify, were the color names presented in the Hungarian speakers’ native language? While I agree with the authors that the perceptual task

(Necker cube) is less likely to be language-sensitive, the magnitude of the Stroop task is likely to be influenced by the subjects' language fluency. In sum, if English color names were used, the authors should provide details about subjects' fluency in English.

Page 5: Please cite Garnsey et al. (1997) when referencing the ambiguous/unambiguous sentence materials, as they were the original source of these sentences.

Garnsey, S. M., Pearlmutter, N. J., Myers, E., & Lotocky, M. A. (1997). The contributions of verb bias and plausibility to the comprehension of temporarily ambiguous sentences. *Journal of Memory and Language*, 37, 58–93. <http://dx.doi.org/10.1006/jmla.1997.2512>.

Minor methodological differences:

Some minor methodological differences were noted. Although we do not believe these differences to contribute meaningfully to the overall results, their choices should be noted or explained.

Page 4, line 52: If purple was one of the incongruent colors, this differs from the original experiment, where incongruent colors were brown, orange and red.
Page 5, line 9: Participants in the original study received 145 baseline Stroop trials rather than 165.

Page 6, line 3: The original effect size for the interaction between current and preceding Stroop congruency on the RT data was 29 ms. The authors report it as 30 ms.

Clarifications needed:

Page 5, line 11: How many practice trials were included?

Page 6, line 50: Please explain the task requirement for the congruent Necker trials.

Page 6, line 60: Please describe how subject group assignment was made. Median split was used in the original study.

Page 7, lines 29 - 34: In the original study, all but one subject achieved near ceiling performance (mean accuracy = 96%, SD = 3%) performed at ceiling. The outlier was only 61% correct on the task, and we excluded that subject's data from analyses, with the assumption that the unusually low task performance suggests a lack of engagement. Thus, to describe 62% as a "cut-off" score is not wholly accurate.

Minor comments:

Page 3, line 47: "The four conditions were determined by the conflict state of the preceding stimulus..." was not an entirely accurate description. The four conditions were determined by the status of preceding and current stimuli.

Page 4, line 21: "They read the sentences word-by-word as they stepped to the next word by pressing the keyboard spacebar." The wording is slightly awkward; we suggest "...word-by-word, advancing to the next word..."

Author's Response to Decision Letter for (RSOS-190749.R0)

See Appendix A.

RSOS-190749.R1 (Revision)

Review form: Reviewer 1 (Friederike Schlaghecken)

Do you have any ethical concerns with this paper?

No

Have you any concerns about statistical analyses in this paper?

No

Recommendation?

Accept in principle

Comments to the Author(s)

In my original review, I strongly recommended to add an experiment testing a 'mere effort' hypothesis for sentence processing. The authors argue that testing alternative explanations would go beyond a replication study. While this is, of course, correct, it doesn't provide a convincing argument in the present context, because the 'mere effort' hypothesis is not just one out of many alternative explanations. Rather, testing this hypothesis was part of the original study, but one that was only partially excluded (i.e., for Necker cubes, not for sentences), and only partially successful.

However, as the editor left the decision whether to include such an experiment to the authors, it would seem rather disingenuous to keep insisting that they do. I have no further concerns about this study.

Review form: Reviewer 3 (Craig Hedge)

Do you have any ethical concerns with this paper?

No

Have you any concerns about statistical analyses in this paper?

No

Recommendation?

Accept in principle

Comments to the Author(s)

The authors have addressed my previous comments and I think the replication plan is solid. I look forward to seeing the results of these studies.

Signed,
Craig Hedge

Review form: Reviewer 4 (Irene Kan)

Do you have any ethical concerns with this paper?

No

Have you any concerns about statistical analyses in this paper?

No

Recommendation?

Accept with minor revision

Comments to the Author(s)

The authors have addressed my initial comments in a satisfactory manner. Two additional points have arisen in the current manuscript as a result of the revision:

1) At Reviewer 3's request, the authors expanded the section on residual reading times calculation. On page 6, lines 10-12, the authors wrote, "We included only the congruent (N = 21) and the filler sentences (N = 29) in the linear regression, as measures of normal reading time." It is unclear to me why the authors only intend to include the congruent and filler sentences, when the important comparisons are those between incongruent and congruent sentences.

However, a few lines later (lines 25-28), the authors wrote, "We tested the presence of the ambiguity effect by comparing the mean residual reading times of congruent and incongruent sentences in each region separately. We expect to find a difference between the conditions in the temporarily ambiguous and in the disambiguating regions." From these sentences, it seems that the authors will be calculating residual reading times for all sentences.

In sum, I find the description of this analysis to be confusing. Please revise.

2) Footnote 4. This is a minor clarification point. The authors wrote, "This is a more conservative threshold than the original authors' 50%, nevertheless, it is based on the original experiment's overall comprehension probe performance." It should be noted that we did not use a strict threshold approach in the original study. As described in section 3.2.1 of the original paper, "One subject's data were removed because of chance-level performance (50%) in the comprehension task. The remaining participants (n = 39) scored at or above 70% (M = .90, SD = .09)." In other words, we discarded one subject's data because he/she performed at chance level. The second sentence was a descriptive of the remaining data.

Decision letter (RSOS-190749.R1)

26-Jul-2019

Dear Dr Aczel

On behalf of the Editors, I am pleased to inform you that your Manuscript RSOS-190749.R1 entitled "Is there evidence for cross-domain congruency sequence effect? A replication of Kan et al. (2013)" deemed suitable for in-principle acceptance in Royal Society Open Science subject to minor revision in accordance with the referee and editor suggestions. Please find their comments at the end of this email.

The reviewers and handling editors have recommended publication, but also suggest some minor revisions to your manuscript. Therefore, I invite you to respond to the comments and revise your manuscript.

Please you submit the revised version of your manuscript within 7 days (i.e. by the 03-Aug-2019). If you do not think you will be able to meet this date please let me know immediately.

Full author guidelines can be found here
<http://rsos.royalsocietypublishing.org/page/replication-studies#AuthorsGuidance>

Kind regards
Alice Power
Editorial Coordinator
Royal Society Open Science
openscience@royalsociety.org

on behalf of Professor Chris Chambers (Subject Editor, Royal Society Open Science)
openscience@royalsociety.org

Editor Comments to Author (Professor Chris Chambers):

Three of the four expert reviewers who assessed the first submission have reappraised the revised manuscript. All are now positive and recommend in principle acceptance (IPA). Please attend to the remaining clarifications requested by Reviewer 4. Provided the authors respond thoroughly to this reviewer's comments in a revision, IPA should be forthcoming without requiring further in-depth review.

Reviewer comments to Author:
Reviewer: 1

Comments to the Author(s)

In my original review, I strongly recommended to add an experiment testing a 'mere effort' hypothesis for sentence processing. The authors argue that testing alternative explanations would go beyond a replication study. While this is, of course, correct, it doesn't provide a convincing argument in the present context, because the 'mere effort' hypothesis is not just one out of many alternative explanations. Rather, testing this hypothesis was part of the original study, but one that was only partially executed (i.e., for Necker cubes, not for sentences), and only partially successful.

However, as the editor left the decision whether to include such an experiment to the authors, it would seem rather disingenuous to keep insisting that they do. I have no further concerns about this study.

Reviewer: 3

Comments to the Author(s)

The authors have addressed my previous comments and I think the replication plan is solid. I look forward to seeing the results of these studies.

Signed,
Craig Hedge

Reviewer: 4

Comments to the Author(s)

The authors have addressed my initial comments in a satisfactory manner. Two additional points have arisen in the current manuscript as a result of the revision:

1) At Reviewer 3's request, the authors expanded the section on residual reading times calculation. On page 6, lines 10-12, the authors wrote, "We included only the congruent (N = 21) and the filler sentences (N = 29) in the linear regression, as measures of normal reading time." It is unclear to me why the authors only intend to include the congruent and filler sentences, when the important comparisons are those between incongruent and congruent sentences.

However, a few lines later (lines 25-28), the authors wrote, "We tested the presence of the ambiguity effect by comparing the mean residual reading times of congruent and incongruent sentences in each region separately. We expect to find a difference between the conditions in the temporarily ambiguous and in the disambiguating regions." From these sentences, it seems that the authors will be calculating residual reading times for all sentences.

In sum, I find the description of this analysis to be confusing. Please revise.

2) Footnote 4. This is a minor clarification point. The authors wrote, "This is a more conservative threshold than the original authors' 50%, nevertheless, it is based on the original experiment's overall comprehension probe performance." It should be noted that we did not use a strict threshold approach in the original study. As described in section 3.2.1 of the original paper, "One subject's data were removed because of chance-level performance (50%) in the comprehension task. The remaining participants (n = 39) scored at or above 70% (M = .90, SD = .09)." In other words, we discarded one subject's data because he/she performed at chance level. The second sentence was a descriptive of the remaining data.

Author's Response to Decision Letter for (RSOS-190749.R1)

See Appendix B.

Decision letter (RSOS-191353.R0)

12-Aug-2019

Dear Dr Aczel

On behalf of the Editor, I am pleased to inform you that your Stage 1 Replication, RSOS-191353 entitled "Is there evidence for cross-domain congruency sequence effect? A replication of Kan et al. (2013)" has been accepted in principle for publication in Royal Society Open Science. Please note that you must now register your approved protocol on the Open Science Framework (<https://osf.io/rr>), using the 'Submit your approved Registered Report' option and then the 'Registered Report Protocol Preregistration' option. Please use the Registered Report option even though your article is being accepted as a Stage 1 Replication.

Please note that a time-stamped, independent registration of the protocol is mandatory under journal policy, and manuscripts that do not conform to this requirement cannot be considered at Stage 2. The protocol should be registered unchanged from its current approved state. Please include a direct URL to the registered protocol in your Stage 2 manuscript, noting that date the IPA was awarded.

Following completion of your study, we invite you to resubmit your paper for peer review as a Stage 2 Replication. Please note that your manuscript can still be rejected for publication at Stage 2 if the Editors consider any of the following conditions to be met:

- The Introduction and methods deviated from the approved Stage 1 submission (required).
- The authors' conclusions were not considered justified given the data.

We encourage you to read the complete guidelines for authors concerning Stage 2 submissions at : <http://rsos.royalsocietypublishing.org/page/replication-studies#AuthorsGuidance>. Please especially note the requirements for data sharing and that withdrawing your manuscript will result in publication of a Withdrawn Registration.

Once again, thank you for submitting your manuscript to Royal Society Open Science and I look forward to receiving your Stage 2 submission. If you have any questions at all, please do not hesitate to get in touch. We look forward to hearing from you shortly with the anticipated submission date for your stage two manuscript.

on behalf of Professor Chris Chambers (Registered Reports Editor, Royal Society Open Science)
openscience@royalsociety.org

Author's Response to Decision Letter for (RSOS-191353.R0)

See Appendix C.

RSOS-191353.R1 (Revision)

Review form: Reviewer 1 (Friederike Schlaghecken)

Is the manuscript scientifically sound in its present form?

Yes

Is the language acceptable?

No

Do you have any ethical concerns with this paper?

No

Have you any concerns about statistical analyses in this paper?

No

Recommendation?

Accept with minor revision

Comments to the Author(s)

Registered Kan et al Replication Stage 2 Review

I have no issues with the study itself (methods, data, analyses, or conclusions). However, the presentation of both the report and the raw data are in need of improvement. Specifically,

1. The text is difficult to read due to grammatical issues, e.g.,
 - a. Misplaced commas (e.g., "was followed by either a Stroop, or a sentence stimulus")
 - b. Misplaced ellipses ("on congruent, on incongruent trials or on both")
 - c. Missing words ("As the original paper, we also", "if we regard the two more similar in nature")
 - d. Unusual phrases ("To this aim", "We found the effect of the congruency")

None of these instances is particularly dramatic, however, the cumulative effect is that the text is rather hard-going, to an extent that I suspect will negatively affect the study's impact.

2. Some information is incorrect or missing:
 - a. In the figure captions, "The X axis shows the congruency type of the current trial," this should probably read "previous trial" (if the figure legends and axes titles are to be trusted)
 - b. I could not find information on what the error bars represent
 - c. In the "Supporting tests of interest", the authors seem report large difference in *Stroop effects* ("tested the extent of the Stroop effect") as a function of the preceding trial's congruency, which stands in clear contrast to the preceding analyses and the data as shown in the figures. I suspect they mean differences in mean RTs (not in Stroop effects) as a function of the preceding trial's congruency.

3. The figures are generally of low quality. In addition, however, I would strongly recommend to use the same type of chart as in the Kan et al paper (i.e., bar graphs instead of line graphs) in order to facilitate comparison between the two papers

4. The raw data on OSF is of such low quality as to be almost useless to anyone other than the authors themselves:

- a. Lacking documentation – each data file should have a corresponding file / worksheet explaining the data in each column
- b. Variable names are not sufficiently informative
- c. Large proportions of each data file appear to be merely tags inserted by the program that hold no relevant information (e.g., dozens of columns filled with “N/A”), making the data files slow to download and difficult to handle.

Signed
Friederike Schlaghecken

Review form: Reviewer 2

Is the manuscript scientifically sound in its present form?

Yes

Is the language acceptable?

No

Do you have any ethical concerns with this paper?

No

Have you any concerns about statistical analyses in this paper?

No

Recommendation?

Accept with minor revision

Comments to the Author(s)

Summary:

In this manuscript, the authors have set out to replicate the results of Kan et al., (2013), which demonstrated support for domain-generalty of cognitive control processes through a cross-task adaptation manipulation in syntactic/non-syntactic and perceptual/verbal domains. Unlike the Kan study, the authors did not find broad support for cross-domain cognitive control. Instead, they found partial support in the syntactic/non-syntactic domains and inconclusive support for experiments testing the perceptual/verbal domains.

Evaluation:

I think this is an interesting paper that will be of value to the scientific community. As the authors themselves indicate in the Introduction, the evidence for conflict adaptation across domains is somewhat mixed, and one of the biggest strengths of the paper is the large sample sizes by which they tested the cross-task cognitive control manipulations. The methods and statistical analyses seem sound.

My one general comment is for the authors to consider somewhat tempering the conclusions of the paper. Yes, it is true that the authors were not able to replicate the perceptual/verbal experimental manipulations with more subjects than in the original Kan et al. study. But at the same time, it is also true that with 108 subjects, the authors did find support for syntactic/non-syntactic cross-task adaptation as measured by accuracy. While this is a "single exception" I wonder if the authors could speculate as to why they found the support in this manipulation but not the perceptual/verbal manipulations.

Relatedly, the concept of affect impacting conflict adaptation processes is interesting, and the paper could benefit from further speculation from the authors on how they might include that within a cognitive control framework, as well as intersect with different cognitive control domains.

The authors might also be interested in recent findings demonstrating that experiencing perceptual or attentional conflict can result in adaptation processes that impact sentence processing:

<https://www.tandfonline.com/doi/full/10.1080/23273798.2020.1836379>

Review form: Reviewer 3 (Craig Hedge)

Is the manuscript scientifically sound in its present form?

Yes

Is the language acceptable?

Yes

Is it clear how to access all supporting data?

Yes

Do you have any ethical concerns with this paper?

No

Have you any concerns about statistical analyses in this paper?

No

Recommendation?

Accept as is

Comments to the Author(s)

Summary

I am reviewing a Stage 2 registered report of Kan et al. (2013). I previously reviewed the manuscript at Stage 1. The replication attempt found limited evidence for the effects in the original paper. Though the original findings did replicate for accuracy in Experiment 1, in most other cases the data were inconclusive.

Review

My opinion of the manuscript is positive and I have no suggestions for improvement. I thought that the authors handled the somewhat difficult pattern of results well in the discussion.

I responded 'no' to Stage 2 Primary Criterion #1 ("Are the introduction and methods the same as the approved Stage 1 submission?"), as the authors note some minor deviations in their supplementary material. However, I thought that these deviations were reasonable, and do not compromise the interpretation of the main results.

Signed,

Craig Hedge

Review form: Reviewer 4

Is the manuscript scientifically sound in its present form?

No

Is the language acceptable?

Yes

Do you have any ethical concerns with this paper?

No

Have you any concerns about statistical analyses in this paper?

No

Recommendation?

Reject

Comments to the Author(s)

See attached file (Appendix D).

Decision letter (RSOS-191353.R1)

Dear Dr Aczel

On behalf of the Editor, I am pleased to inform you that your Stage 2 Replication submission RSOS-191353.R1 entitled "Is there evidence for cross-domain congruency sequence effect? A replication of Kan et al. (2013)" has been accepted for publication in Royal Society Open Science subject to minor revision in accordance with the referee suggestions. Please find the referees' comments at the end of this email.

The reviewers and Subject Editor have recommended publication, but also suggest some minor revisions to your manuscript. Therefore, I invite you to respond to the comments and revise your manuscript.

Please also ensure that all the below editorial sections are included where appropriate (a non-exhaustive example is included in an attachment):

- Ethics statement

- Data accessibility

If you wish to submit your supporting data or code to Dryad (<http://datadryad.org/>), or modify your current submission to dryad, please use the following link:
<http://datadryad.org/submit?journalID=RSOS&manu=RSOS-191353.R1>

- **Competing interests**

- **Authors' contributions**

- **Acknowledgements**

- **Funding statement**

Because the schedule for publication is very tight, it is a condition of publication that you submit the revised version of your manuscript within 7 days (i.e. by the 26-Jan-2021). If you do not think you will be able to meet this date please let me know immediately.

- 1) A text file of the manuscript (tex, txt, rtf, docx or doc), references, tables (including captions) and figure captions. Do not upload a PDF as your "Main Document".
- 2) A separate electronic file of each figure (EPS or print-quality PDF preferred (either format should be produced directly from original creation package), or original software format)
- 3) Included a 100 word media summary of your paper when requested at submission. Please ensure you have entered correct contact details (email, institution and telephone) in your user account
- 4) Included the raw data to support the claims made in your paper. You can either include your data as electronic supplementary material or upload to a repository and include the relevant DOI within your manuscript
- 5) Included your supplementary files in a format you are happy with (no line numbers, Vancouver referencing, track changes removed etc) as these files will NOT be edited in production

Kind regards,
 Professor Chris Chambers
 Royal Society Open Science
 openscience@royalsociety.org

on behalf of Professor Chris Chambers (Registered Reports Editor, Royal Society Open Science)
 openscience@royalsociety.org

Associate Editor Comments to Author (Professor Chris Chambers):

Associate Editor: 1

Comments to the Author:

All four reviewers from Stage 1 very kindly returned to assess the Stage 2 manuscript. Reviewers 1, 2 and 3 are broadly positive while noting areas that would benefit from some improvement, including the balance of the conclusions, general presentation and usability of the OSF repository.

Reviewer 4 is more critical, in fact recommending rejection, largely on the basis of various deviations from the approved protocol and disagreement about the interpretation of results (linking to Stage 2 primary criterion #2 -- whether the conclusions are based on the evidence). The reviewer's assessment is very detailed and helpful and I would like to provide some guidance on certain points to address in revision. First, concerning deviations from the Stage 1 protocol, it is vital that these are as transparent as possible and clearly justified. Second, please do not make any changes to the rationale or Introduction to address additional literature. Any points raised by the reviewer in this regard (e.g. the reviewer's comment about insufficient contextualisation) should instead be addressed in the Discussion. Third, like Reviewer 4, the reviewer also finds the OSF archive difficult to navigate (with potentially missing data) so please

address this carefully. And of course, please address all other remaining points raised by this reviewer and the other reviewers in a point-by-point response and revision.

Reviewers' comments to Author:

Reviewer: 1

Comments to the Author(s)

Registered Kan et al Replication Stage 2 Review

I have no issues with the study itself (methods, data, analyses, or conclusions). However, the presentation of both the report and the raw data are in need of improvement. Specifically,

1. The text is difficult to read due to grammatical issues, e.g.,
 - a. Misplaced commas (e.g., "was followed by either a Stroop, or a sentence stimulus")
 - b. Misplaced ellipses ("on congruent, on incongruent trials or on both")
 - c. Missing words ("As the original paper, we also", "if we regard the two more similar in nature")
 - d. Unusual phrases ("To this aim", "We found the effect of the congruency")

None of these instances is particularly dramatic, however, the cumulative effect is that the text is rather hard-going, to an extent that I suspect will negatively affect the study's impact.

2. Some information is incorrect or missing:

- a. In the figure captions, "The X axis shows the congruency type of the current trial," this should probably read "previous trial" (if the figure legends and axes titles are to be trusted)
- b. I could not find information on what the error bars represent
- c. In the "Supporting tests of interest", the authors seem report large difference in *Stroop effects* ("tested the extent of the Stroop effect") as a function of the preceding trial's congruency, which stands in clear contrast to the preceding analyses and the data as shown in the figures. I suspect they mean differences in mean RTs (not in Stroop effects) as a function of the preceding trial's congruency.

3. The figures are generally of low quality. In addition, however, I would strongly recommend to use the same type of chart as in the Kan et al paper (i.e., bar graphs instead of line graphs) in order to facilitate comparison between the two papers

4. The raw data on OSF is of such low quality as to be almost useless to anyone other than the authors themselves:

- a. Lacking documentation - each data file should have a corresponding file / worksheet explaining the data in each column
- b. Variable names are not sufficiently informative
- c. Large proportions of each data file appear to be merely tags inserted by the program that hold no relevant information (e.g., dozens of columns filled with "N/A"), making the data files slow to download and difficult to handle.

Signed

Friederike Schlaghecken

Reviewer: 2

Comments to the Author(s)

Summary:

In this manuscript, the authors have set out to replicate the results of Kan et al., (2013), which demonstrated support for domain-generalty of cognitive control processes through a cross-task

adaptation manipulation in syntactic/non-syntactic and perceptual/verbal domains. Unlike the Kan study, the authors did not find broad support for cross-domain cognitive control. Instead, they found partial support in the syntactic/non-syntactic domains and inconclusive support for experiments testing the perceptual/verbal domains.

Evaluation:

I think this is an interesting paper that will be of value to the scientific community. As the authors themselves indicate in the Introduction, the evidence for conflict adaptation across domains is somewhat mixed, and one of the biggest strengths of the paper is the large sample sizes by which they tested the cross-task cognitive control manipulations. The methods and statistical analyses seem sound.

My one general comment is for the authors to consider somewhat tempering the conclusions of the paper. Yes, it is true that the authors were not able to replicate the perceptual/verbal experimental manipulations with more subjects than in the original Kan et al. study. But at the same time, it is also true that with 108 subjects, the authors did find support for syntactic/non-syntactic cross-task adaptation as measured by accuracy. While this is a "single exception" I wonder if the authors could speculate as to why they found the support in this manipulation but not the perceptual/verbal manipulations.

Relatedly, the concept of affect impacting conflict adaptation processes is interesting, and the paper could benefit from further speculation from the authors on how they might include that within a cognitive control framework, as well as intersect with different cognitive control domains.

The authors might also be interested in recent findings demonstrating that experiencing perceptual or attentional conflict can result in adaptation processes that impact sentence processing:

<https://www.tandfonline.com/doi/full/10.1080/23273798.2020.1836379>

Reviewer: 3

Comments to the Author(s)

Summary

I am reviewing a Stage 2 registered report of Kan et al. (2013). I previously reviewed the manuscript at Stage 1. The replication attempt found limited evidence for the effects in the original paper. Though the original findings did replicate for accuracy in Experiment 1, in most other cases the data were inconclusive.

Review

My opinion of the manuscript is positive and I have no suggestions for improvement. I thought that the authors handled the somewhat difficult pattern of results well in the discussion.

I responded 'no' to Stage 2 Primary Criterion #1 ("Are the introduction and methods the same as the approved Stage 1 submission?"), as the authors note some minor deviations in their supplementary material. However, I thought that these deviations were reasonable, and do not compromise the interpretation of the main results.

Signed,
Craig Hedge

Reviewer: 4
Comments to the Author(s)
See attached file.

Author's Response to Decision Letter for (RSOS-191353.R1)

See Appendix E.

Decision letter (RSOS-191353.R2)

Dear Dr Aczel:

It is a pleasure to accept your manuscript entitled "Is there evidence for cross-domain congruency sequence effect?

A replication of Kan et al. (2013)" in its current form for publication in Royal Society Open Science.

In editing the final version for copyediting, please ensure that the Data Accessibility section states the direct URL to the approved Stage 1 manuscript. In the tracked changes version of the revised manuscript, it appears that this link (<https://osf.io/ek5rz>) may have been inadvertently removed. Please ensure that it is included in a statement consisting of: "This study was preregistered prior to data collection and analysis on 20 September, 2019. The accepted Stage 1 manuscript, unchanged from the point of in-principle acceptance, may be viewed at <https://osf.io/ek5rz>"

Please see the Royal Society Publishing guidance on how you may share your accepted author manuscript at <https://royalsociety.org/journals/ethics-policies/media-embargo/>. After publication, some additional ways to effectively promote your article can also be found here

<https://royalsociety.org/blog/2020/07/promoting-your-latest-paper-and-tracking-your-results/>.

on behalf of Professor Chris Chambers (Subject Editor)
openscience@royalsociety.org

Dear prof Chambers,

We are happy to submit a revised version of our RRR manuscript to the *Royal Society Open Science*.

We would like to thank you, and the reviewers for their comments and suggestions. Below, in bold, you can find the detailed responses to all comments.

Kind regards,
Balazs Aczel, on behalf of all co-authors

Editor's Comments

Four expert reviewers have now assessed the manuscript. The reviews are very high quality and guardedly positive, with three reviewers judging that the Stage 1 primary criteria are currently met. All reviewers, however, point out a range of issues to be addressed in revision. Reviewer 1 notes that the study would benefit from an additional control experiment. Where this goes beyond the experiments reported in the original study, it is not necessary to conduct such an experiment to achieve Stage 1 acceptance. However if the authors agree, then an experiment that goes beyond the replication could be added to the design under special circumstances, provided it is clearly flagged as in addition to the target study. Reviewers 3 and 4 note a range of areas that would benefit from expansion and clarification, including the clarity of the introduction, use of terminology, and analysis plans. In relation to Reviewer 4's comment: "The rationale of the present paper would be clearer if the authors had elaborated on how the replication attempt would add to our understanding of the domain-generalty of cognitive control. It will be important and instructive to identify the theoretical contribution of this replication attempt." -- note that under the Replications policy, a theoretical justification for replication is not necessary to achieve Stage 1 acceptance, although the authors are welcome to respond positively to this suggestion if they wish. Reviewers 3 and 4 also note some methodological deviations (or potential deviations) between the replication and the original experiments. These should be carefully addressed.

Reply to the Editor

The main aim of the present replication is to ascertain that the effect is valid. There are a good number of contesting theories trying to explain the empirical effect, therefore, we would not add a new experiment to this replication, nevertheless, we intend to discuss the alternative explanations in the Discussion section to motivate the readers to build on our findings to further advance the theoretical questions of the topic.

We accepted the argument of Reviewer 4, 'conflict adaptation' is a theoretically loaded label and while this is what the original authors used, we changed our wording to the synonymous 'congruency sequence effect' everywhere in the text as well as in the title.

All other comments of the reviewers are answered below.

Comments to Author:

Reviewer: 1

Comments to the Author(s)

Review RSOS-190749

Comment 1

The authors propose to carry out an exact replication of the three experiments reported in Kan et al. (2013), increasing participant numbers by a factor of 2.5. The proposed study seems sound from a technical perspective.

From a theoretical perspective, however, I am less convinced. The original study finds that Stroop effects are smaller following ambiguous material (garden path sentences, Necker cubes) than following unambiguous material. This is interpreted as a ‘conflict adaptation’ effect. Alternatively, however, this reduction might be an after-effect of the greater effort needed to process ambiguous material, regardless of any ‘conflict’ it presents.

The original study contains a control experiment (Exp 3) to distinguish between these two interpretations (conflict adaptation effect versus effort after-effect) for the Necker cube task (Exp 2) only. The success of this is limited (Exp 3 produces a pattern of result similar to that produced by the ‘low reversal’ group in Exp 2). Importantly, the study does not contain a control experiment for the sentence task (Exp 1).

I strongly recommend that the present authors close this gap by adding the missing sentence-task control experiment. An obvious way of doing so would be to use non-ambiguous sentences in normal and in disfluent font (e.g., sans forvetica or some other distorted font that reduces reading speed to the same extent as garden path sentences do). In my opinion, investigating the influence of pure reading effort on the Stroop effect might provide a more clear-cut way to distinguish between the ‘conflict adaptation’ and the ‘effort after-effect’ hypotheses.

Signed

Friederike Schlaghecken

Reply 1

We agree with the reviewer that the effect, if exists, can have alternative explanations. In fact, there are several candidate accounts. Besides Botvinick et al.’s conflict-based explanation, others argue for working memory triggered dynamic adjustment (e.g., Jha & Kivonage, 2010); the affective signal hypothesis (Dreisbach & Fisher, 2015); or a dynamic goal state explanation (Scherbaum et al., 2012), just to mention a few.

We believe that before finding the most convincing explanation to the effect, we have to ascertain that the effect is valid. This is the main aim of the present study. While we would not add a new experiment to this replication, we intend to discuss the alternative explanations in the Discussion section to motivate the readers to build on our findings to further advance the theoretical questions of the topic. For this revision, we listed a few topics that our Discussion should cover. Of course, the focus of our Discussion will depend on the findings of the study.

- Dreisbach, G., & Fischer, R. (2015). Conflicts as aversive signals for control adaptation. *Current Directions in Psychological Science*, 24(4), 255-260.
- Jha, A. P., & Kiyonaga, A. (2010). Working-memory-triggered dynamic adjustments in cognitive control. *Journal of Experimental Psychology: Learning, Memory, and Cognition*, 36(4), 1036-1042.
- Scherbaum, S., Dshemuchadse, M., Ruge, H., & Goschke, T. (2012). Dynamic goal states: adjusting cognitive control without conflict monitoring. *Neuroimage*, 63(1), 126-136.

Reviewer: 2

Comment 2

This looks to be a valuable replication of a key paper demonstrating causal relationships between cognitive control engagement processes across syntactic, non-syntactic, verbal, and perceptual domains.

One aspect that I would like to see included (and that I don't think I saw in this version) is some additional description/clarification of the nature of the ambiguities across the multiple languages, as well as some comparison (if at all possible) or their similarity to the ones used in the Kan et al., study.

Reply 2

For the language-based Experiment 1, we will recruit only native speakers of English in Australia, Singapore, and the United Kingdom. The Hungarian subjects will participate only in the non-verbal Necker-cube test.

Reviewer: 3

Comments to the Author(s)

Summary

The manuscript is a stage 1 replication of Kan et al's (2013) paper "To adapt or not to adapt: The question of domain-general cognitive control". The original paper and proposed replication consist of three studies designed to test whether cognitive control adaptations are domain-general. The first study consists of participants performing intermixed trials of a sentence comprehension and Stroop tasks. In both tasks, stimuli could either be unambiguous/congruent or ambiguous/incongruent. The outcome of interest is whether there is an interaction between (current) Stroop trial congruency and (preceding) sentence comprehension trial congruency. Experiments 2 and 3 follow a similar logic, with the sentence comprehension task being replaced by a perceptual task (Necker cube viewing). An unambiguous Necker Cube stimulus has a clear directionality, whereas ambiguous Necker Cube stimulus can be perceived to go in one of two directions, and views may flip between them during passive viewing. Experiment 3 in the original study replaced the ambiguous Necker stimuli with alternating unambiguous stimuli to assess the importance of "internal representational conflict". The original study observed conflict adaptation (reduced Stroop effect following ambiguous preceding trials) in Experiments 1 and 2, but not 3. This

replication study closely follows the design of the original, while targeting larger sample sizes.

Review

Comment 3

My opinion on this manuscript is generally positive, and I think the criteria for a Stage 1 can be met with minor revisions. The extent to which cognitive control processes are domain-general is an ongoing question in the literature. The proposed replication is potentially informative both from a reproducibility perspective, as well in terms of the broader theoretical questions. As a replication, the proposal seems strong. It closely follows the method of the original study, and benefits from an increased sample size and clear stopping rule.

I think that the manuscript would benefit from more detail on a few points, as the rationale/method were not entirely clear without reading the original paper. I also have a question regarding how the numerous critical tests will be interpreted with respect to both the original findings and the overarching theoretical question (point 3 below).

1) The authors describe and reference the statistical approach for the critical tests, but it may be useful to go through the steps in more detail in the manuscript. The authors state on page 5:

“We performed and reported all of our hypothesis tests as either paired or independent t tests for the sake of simplicity.”

The data analyses sections state that the critical analyses are two-way (Exps. 1 & 3) and three-way (Exp 2) interactions. I am not an expert on Dienes’ approach, though my understanding is that these interactions can be reduced to something like a t -test per his 2014 paper. It may not be intuitive to a naïve reader how this is done, though, particularly with the three-way interaction. In order to aid comparisons with the frequentist ANOVAs in the original paper, perhaps the working and Bayes factors could be shown for the original results if sufficient information is given?

Reply 3

When the degrees of freedom of the numerator of an F test is 1, the square root of the F value equals the t value. For a detailed explanation see, for instance, Abelson (1995, p. 63-65). This relationship is true regardless of whether one uses the t value to obtain a p value or the Bayes factor. Should the editor recommend, we can report the F values along with the t values to aid comparability with the original tests.

Abelson, R. A. (1995). *Statistics as principled argument*. Hillsdale, NJ: Erlbaum.

Comment 4

2) The performance cut-offs may warrant consideration. My reading of the original paper is that they report the performance of the participants that were excluded, but I couldn’t see the

threshold that they applied. If the cut-offs used in the original are not known, the authors may want to consider whether those selected are optimal from a data quality perspective. In particular:

-Page 7 of the replication states:

“using the cutting point of the original study, we excluded the data of participants whose performance was below 62% on accurately identifying whether a stimulus change occurred on the current trial.”

Page 646 of the original paper states:

“One subject’s data were removed from analyses because of poor performance on this straightforward task (61%)”

It is possible that the original study applied a more conservative cut-off (e.g.>70%), and no other participants fell below it.

-My understanding is that there are no exclusion criteria biased on Stroop task performance? My reading of the original manuscript is that they did not exclude anyone on this basis, but again this may be because performance did not cross an unstated threshold. As performance cut-offs are applied to other tasks, it may be appropriate to apply them here too.

-The authors state that they will exclude participants who are at chance (50%) in the sentence comprehension task in Study 1. Is <50% the threshold to be used (i.e. will participants be retained if they are at 51%)? The original paper states that the remaining participants scored “at or above 70%” (pg 640).

Reply 4

The original authors reported that they used the 50% threshold, however, we agree with the reviewer and increased our threshold to 70% as 50-70% accuracy scores can be still considered as poor performance. We updated this value in the manuscript.

Comment 5

3) As in the original paper, the analysis plan for the replication consists of multiple experiments, each of which has multiple critical and supporting tests. This complexity makes it difficult to specify a single condition by which to say that the original findings were replicated. However, there is a range of possible outcomes between replicating all or none of the effects observed in the original paper.

In terms of the theoretical question of whether control adaptations are domain-general, I wondered if the authors could be more explicit in advance about what outcomes would be compatible or incompatible with this account. For example, I can imagine some combination of findings that do not have a straightforward interpretation:

- If H1 is supported in experiments 2 & 3, but not experiment 1

- If H1 is supported in the critical tests on reaction time but not in accuracy (or vice versa, which could also be inconsistent across the experiments)
- If H1 is supported in the critical test on (e.g.) reaction time, but the supporting tests show that it comes from an increase in congruent RTs and no change in incongruent RTs.

As above, I appreciate that mapping all the possible combinations of outcomes on to explanations is infeasible. However, it wasn't clear to me then whether the authors were going to consider every critical test individually, or whether a broader conclusion was going to be drawn based on a minimum set of conditions.

Reply 5

This is an important aspect of any replication. Thank you very much for bringing it up. Having three experiments and multiple dependent variables, we plan to use three categories for our possible outcomes: (1) the pattern of our results fully overlap with the original results; (2) none of our results replicated the original results; and (3) partial replication when some, but not all of our results show the same pattern as the original study. In all cases, we would discuss the results in light of the original hypotheses as well, as it is possible that our results support the hypotheses of the original authors without replicating their results (e.g., the original study did not receive statistical support for the effect for all measures). We will say that our results are in line with the original results when we reach Bayesian positive evidence for originally significant results. Now we extended our Method section with this clarification.

Comment 6

4) The authors could give more details regarding the calculation of residual reading time (e.g. if it involves a per-subject regression).

Reply 6

We amended the manuscript (p. 8) with additional details about how we plan to compute the residual reading times. Please see the Data preprocessing subsection for Experiment 1 for the extended analysis plan.

Comment 7

5) The authors give three reasons why they chose to conduct a direct replication of Kan et al. (pg 3). While these seemed like a reasonable justification, I was curious whether any other papers were also evaluated.

Reply 7

Our choice was based on our best knowledge of the literature, but we have also consulted the original author and other experts of the field before our choice.

Comment 8

6) The authors state that they will apply a threshold of 3 (/0.33) as a stopping rule for their Bayes factors. I note that several journals require more conservative thresholds for registered reports (e.g. Cortex & BMJ Open science require >6). I don't object to the

threshold of 3 chosen here, but it is perhaps worth further qualification or commenting that there isn't universal agreement on a 'good enough' threshold.

Reply 8

We acknowledge that the cut-off of good enough evidence is a matter of debate and we chose 3 (and $\frac{1}{3}$) as it is usually what one gets when null-hypothesis significance testing results in a p value of .05 and the obtained effect size is about the same as the one predicted by H1 (Dienes, 2014; Jeffreys, 1961). We added a citation in the manuscript referencing Jeffreys, who argued that the cut-off of substantial (i.e., good enough) evidence should be 3. Nevertheless, we leave the decision to the editor whether we should use a different cut-off, such as 6 (and $\frac{1}{6}$).

Jeffreys, H. (1961). *The theory of probability*. Oxford, UK: Oxford University Press.

Dienes, Z. (2014). Using Bayes to get the most out of non-significant results. *Frontiers in Psychology*, 5, 781.

Comment 9

7) It would be useful to include a diagram of the Necker Cube stimuli.

Reply 9

A diagram of the Necker stimuli is now included in the manuscript on p. 10.

Comment 10

8) A minor point - on page 5 the authors state:

"Hence, our analyses were identical to those of the original paper".

I understand this to mean that the level of aggregation they describe previously is identical, but it may also be interpreted that the original paper used the same Bayesian approach that the authors go on to discuss. Perhaps this discrepancy could be more explicit here.

Reply 10

We amended the text stating it clearly under 'Data analysis' that while we report p values along all of our statistical tests, we also calculate Bayes factors and use that for our conclusions.

Comment 11

9) On page 3:

"All participants performed two tasks: first a Stroop-test, then either a sentence processing task (Experiment

1), or a perceptual processing task (Experiment 2, 3)."

It may be more appropriate to say the sentence/perceptual tasks first, as they preceded the Stroop trials?

Reply 11

Thank you for the suggestions, we have now corrected this sentence.

Comment 12

10) On page 3, I was unsure what the authors meant when they refer to "various NoGo decision criteria" used by Feldman et al. (2015).

Reply 12

A Go/NoGo decision criterion tells when to Go, when to press the button when a sequence of stimuli are presented. For example, press the button only when you see fruits, but not animals. This criterion can change among trials.

Signed,
Craig Hedge

Reviewer: 4

Comments to the Author(s)

Introduction:

Comment 13

Page 1, line 41: The paper will be strengthened if a more thorough discussion of the proposed mechanisms of conflict adaptation. The authors wrote, "It is assumed that in these tasks the detection of conflict enhances cognitive control resulting in improved conflict resolution of subsequent trials. If the presence of conflict is sufficient for this conflict adaptation effect, then conflict adaptation can be expected to occur in cross-domain tasks."

What is it that is getting "passed" between consecutive trials? What is adaptation thought to reflect? Even for studies that do find adaptation, not all agree that conflict is a critical ingredient (see work by Schmidt, Weissman, etc.), even if they do believe that cognitive control is necessary for the effect to emerge. For example, are people biasing attention to task-relevant over task-irrelevant cues in the cross-task paradigm? Such a state in one trial of one task could sustain such that one is ready to bias attention on the next trial too, even if it is another task entirely (one won't know how to bias attention accordingly until one knows what task she's in). Does some central conflict monitor/resolver have to have access to the representations for each task, so it "knows" how to resolve them? Or does this theoretically general mechanism not have to have such access to the representations? If not, how is reactive control achieved?

Reply 13

We agree with the reviewer that the explanation of the effect (if exists) is not a simple question. As we discussed it in Reply 1, we regard it as an important theoretical question. Our motivation with this replication project is to establish the empirical phenomenon to allow further studies to explore the mechanisms behind it. While we built the Introduction on the mainstream interpretation, in the Discussion we will allow more space to the alternative frameworks.

Comment 14

The second sentence in the above quotation is opaque. I think the authors are trying to make contact with the different theories behind what triggers such behavioral adjustments (sometimes referred to as the congruence sequence effect), however, it did not come across as such.

Reply 14

We agree with the reviewer and we changed all ‘conflict adaptation’ label ‘congruency sequence effect’, including the title of the paper.

Comment 15

Page 3, line 13: The authors wrote, “Since the Conflict Monitoring Theory (3), posits that the detection of conflict triggers cognitive control mechanisms, these mechanisms can be studied by comparing performance on the incongruent trials with responses on congruent trials. Greater cognitive control is thought to reflect less observed differences in congruent and incongruent trials.”

It is unclear what is meant by “less observed differences in congruent and incongruent trials”. Again, a more detailed description of the connection to how behavior is regulated in such paradigms should be unpacked a bit more. (Relatedly, it is unclear what specific mechanisms the authors are referring to when they wrote “cognitive control mechanisms”.

Reply 15

As raised in Reply 13, we will elaborate on the alternative frameworks and explanations about the underlying mechanisms of the current effect in the Discussion section.

Comment 16

Page 3, line 44: The authors wrote, “For example, the sentence: “The basketball player accepted the contract would have to be negotiated”, is ambiguous without reinterpreting the sentence after it is read in its entirety.” This description is a bit confusing. It is not ambiguous without reinterpreting (that is not the source of the syntactic ambiguity). The sentence is ambiguous because “the contract” could temporarily be the verb’s direct object or the subject of an ensuing clause. English readers are biased to commit to the direct object interpretation, because “accepted” frequently appears in such syntactic constructions; but the sentence doesn’t turn out that way. In other words, the temporary conflict arose due to such syntactic biases. Readers realize this upon encountering “would have”, which is incompatible with the direct object analysis, which they then have to abandon through some conflict resolution mechanism (theoretically).

Reply 16

We agree with the reviewer. This is the essence of the garden path sentences.

Comment 17

Page 3: The rationale of the present paper would be clearer if the authors had elaborated on how the replication attempt would add to our understanding of the domain-generalty of cognitive control. It will be important and instructive to identify the theoretical contribution of this replication attempt.

Reply 17

We think that the research of this topic has two important challenges. First, we have to establish the existence of the empirical phenomenon. Second, we have to connect it to a theoretical explanation. We believe that it would be futile to start with the theoretical explanation if we are uncertain about the explanandum. The present study is dedicated solely to the first challenge. Nevertheless, the original study is part of a theoretical debate concerning the domain-generalizability of conflict adaptation, therefore, positive or negative findings would equally relate to this account, as we will deliberate about it in the Discussion.

Comment 18

Relatedly, the authors identified limitations of the studies that reported null effects (i.e., absence of evidence is not evidence for absence) and suggested that those prior studies may have lacked sensitivity. But a replication attempt of previous positive findings does not address the issue of sensitivity in prior studies that reported null findings.

Reply 18

If we understand it correctly, the reviewer is asking whether our replication of a study with positive findings will explain the null findings of other studies. Our answer is that our results won't explain why other studies with different designs did not find evidence for the effect. The effect might not exist in any designs, or it might not work in those designs, or those studies had insufficient statistical power. If the question is whether the effect exists then we reason that it makes the most sense to test it on a design which is believed to produce the effect.

Methods & Procedures:

Comment 19

Page 4: College students participated in the original studies. What is the demographic of the participants for this project?

Reply 19

All the participants will be university students. Based on previous studies we estimate that the sample will consist of predominantly female participants (~ 70%) with a median age of 21, and with a few outliers over 30 years old.

Comment 20

Pages 4, 5 & 7 (related to number of subjects and data analysis approach): The authors will be approaching the analyses with a Bayesian perspective, while the original studies utilized the more traditional approach. Given recent interests in the inferences associated with the different analysis approaches, this difference is reasonable. However, the rationale for the authors' choice should be justified.

Reply 20

The Bayes factor allows us to distinguish between evidence supporting the null and insensitive evidence, whereas frequentist statistics cannot substantiate a conclusion about the absence of an effect. This feature of the Bayes factor makes the tool ideal for replication projects and we chose to use it for this reason. We added a sentence in the Manuscript that elaborates this point.

Comment 21

Furthermore, please provide an explanation of the optional stopping rule. The rationale is not obvious, and it would be instructive for the authors to explain why the optional stopping approach is acceptable/unbiased. Relatedly, why 2.5 times the original sample sizes?

Reply 21

Now, we mention in a footnote that using optional stopping is not an issue for Bayesians as the Bayes factor retains its meaning regardless of the applied stopping rule. We also cite two core papers that elaborate this argument (Dienes, 2016; Rouder, 2014). We set the minimum sample size of our study at 2.5 times higher of the original sample size to increase the probability of obtaining evidence at the time of the first data analysis (Simonsohn, 2015). If the Bayes factor has not reached one of the preset cut-off points after the data collection reached the minimum sample size, we are going to continue the data collection until six months after the in-principle acceptance of the Stage 1 procedure unless we reach one of the preset cut-off points earlier. We will analyze the collected data weekly.

Comment 22

Page 5, lines 7 and 8: The authors wrote, “For Experiment 2 and 3, we tested Hungarian native speakers as the stimuli were not language specific.” To clarify, were the color names presented in the Hungarian speakers’ native language? While I agree with the authors that the perceptual task (Necker cube) is less likely to be language-sensitive, the magnitude of the Stroop task is likely to be influenced by the subjects’ language fluency. In sum, if English color names were used, the authors should provide details about subjects’ fluency in English.

Reply 22

Now we made it clear that Experiment 2 and 3 were tested in Hungarian.

Comment 23

Page 5: Please cite Garnsey et al. (1997) when referencing the ambiguous/unambiguous sentence materials, as they were the original source of these sentences.

Garnsey, S. M., Pearlmutter, N. J., Myers, E., & Lotocky, M. A. (1997). The contributions of verb bias and plausibility to the comprehension of temporarily ambiguous sentences. *Journal of Memory and Language*, 37, 58–93. <http://dx.doi.org/10.1006/jmla.1997.2512>.

Reply 23

We added this reference to the article.

Minor methodological differences:

Some minor methodological differences were noted. Although we do not believe these differences to contribute meaningfully to the overall results, their choices should be noted or explained.

Comment 24

Page 4, line 52: If purple was one of the incongruent colors, this differs from the original experiment, where incongruent colors were brown, orange and red.

Page 5, line 9: Participants in the original study received 145 baseline Stroop trials rather than 165.

Reply 24

These details were in the manuscript by mistake, the incongruent colors are brown, orange and red, and there are only 145 baseline Stroop trials in the experiment material. We corrected it in the manuscript.

Comment 25

Page 6, line 3: The original effect size for the interaction between current and preceding Stroop congruency on the RT data was 29 ms. The authors report it as 30 ms.

Reply 25

We based the 30s on the difference between Stroop effect after congruent ($M_{\text{difference}} = 43$ ms) and after incongruent trials ($M_{\text{difference}} = 13$ ms) reported in the 3.2.3. *Stroop reaction time* section of the Kan et al. (2013) paper. However, if we calculate the CSE for Experiment 1. from the mean reaction times reported for each condition (CC, CI, IC, II) separately in Table 1, the raw effect size is 29 ms as the reviewer pointed it out. Thus, we would keep *SD* of the H_1 model for the Bayes factor analysis to 30 ms.

Comment 26

Clarifications needed:

Page 5, line 11: How many practice trials were included?

Reply 26

10 Stroop and 10 filler sentence stimuli. We added it to the manuscript.

Comment 27

Page 6, line 50: Please explain the task requirement for the congruent Necker trials.

Reply 27

We included additional explanation to the congruent Necker trial part.

Comment 28

Page 6, line 60: Please describe how subject group assignment was made. Median split was used in the original study.

Reply 28

We are going to use median-split to assign participants to low and high reversal groups in Experiment 2. We added this information in the manuscript on page 10 under the *Data analysis* subsection.

Comment 29

Page 7, lines 29 - 34: In the original study, all but one subject achieved near ceiling performance (mean accuracy = 96%, SD = 3%) performed at ceiling. The outlier was only 61% correct on the task, and we excluded that subject's data from analyses, with the assumption that the unusually low task performance suggests a lack of engagement. Thus, to describe 62% as a "cut-off" score is not wholly accurate.

Reply 29

We increased our threshold to 70% as 50-70% accuracy scores can be still considered as poor performance. We updated this value in the manuscript.

Comment 30

Minor comments:

Page 3, line 47: "The four conditions were determined by the conflict state of the preceding stimulus..." was not an entirely accurate description. The four conditions were determined by the status of preceding and current stimuli.

Reply 30

We corrected this sentence.

Comment 31

Page 4, line 21: "They read the sentences word-by-word as they stepped to the next word by pressing the keyboard spacebar." The wording is slightly awkward; we suggest "...word-by-word, advancing to the next word..."

Reply 31

Suggestion accepted, we replaced this part with the sentence.

Appendix B

Dear prof. Chambers,

We are happy to submit our replies to the comments we received to our RRR submission.

**Kind regards,
Balazs Aczel, on behalf of all co-authors**

Editor Comments

Three of the four expert reviewers who assessed the first submission have reappraised the revised manuscript. All are now positive and recommend in principle acceptance (IPA). Please attend to the remaining clarifications requested by Reviewer 4. Provided the authors respond thoroughly to this reviewer's comments in a revision, IPA should be forthcoming without requiring further in-depth review.

Reviewer comments to Author:

Reviewer: 4

The authors have addressed my initial comments in a satisfactory manner. Two additional points have arisen in the current manuscript as a result of the revision:

Comment 1

1) At Reviewer 3's request, the authors expanded the section on residual reading times calculation. On page 6, lines 10-12, the authors wrote, "We included only the congruent (N = 21) and the filler sentences (N = 29) in the linear regression, as measures of normal reading time." It is unclear to me why the authors only intend to include the congruent and filler sentences, when the important comparisons are those between incongruent and congruent sentences.

However, a few lines later (lines 25-28), the authors wrote, "We tested the presence of the ambiguity effect by comparing the mean residual reading times of congruent and incongruent sentences in each region separately. We expect to find a difference between the conditions in the temporarily ambiguous and in the disambiguating regions." From these sentences, it seems that the authors will be calculating residual reading times for all sentences.

In sum, I find the description of this analysis to be confusing. Please revise.

Reply 1

Our understanding is that the aim of this analysis is to validate the efficacy of the ambiguity manipulation by examining the residual reading times (of the congruent and incongruent sentences) by subtracting each subject's predicted reading time from their actual reading time. For us, it only makes sense to predict the person's reading time from their normal reading, that is, the reading of congruent and filler sentences. This is why we would not take into account the participants' reading time of the incongruent sentences when estimating their normal reading time. We can only assume that this is how the original analysis has been conducted.

Therefore, our answer is that we would calculate the residual reading times for all sentences, but to estimate the normal reading time, we would not use the data of the incongruent sentences.

To make our procedure more comprehensible, we added the following sentence to this part: "From this calculation, we excluded the reading times of the incongruent sentences as the ambiguity manipulation is expected to make these times longer than the normal reading time."

Comment 2

2) Footnote 4. This is a minor clarification point. The authors wrote, "This is a more conservative threshold than the original authors' 50%, nevertheless, it is based on the original experiment's overall comprehension probe performance." It should be noted that we did not use a strict threshold approach in the original study. As described in section 3.2.1 of the original paper, "One subject's data were removed because of chance-level performance (50%) in the comprehension task. The remaining participants ($n = 39$) scored at or above 70% ($M = .90$, $SD = .09$)." In other words, we discarded one subject's data because he/she performed at chance level. The second sentence was a descriptive of the remaining data.

Reply 2

Thank you for the comment. We changed this footnote to:

"This threshold is based on the overall comprehension probe performance of the original experiment."

Stage 1 Stage 2 Manuscript Comparison

Is there evidence for cross-domain congruency sequence effect?

A replication of Kan et al. (2013)

Balazs Aczel^{a*†}, Marton ~~Kovacs~~^a~~Kovacs~~^{ab}, Miklos ~~Bognar~~^a~~Bognar~~^{ab}, Bence
Palfi^{b,c}~~Palfi~~^c,

Andree Hartanto^d, Sandersan Onie^e, Lucas, E. Tiong^d, Thomas Rhys Evans^f

^a*Institute of Psychology, ELTE, Eotvos Lorand University, Budapest, Hungary*

^b~~School~~^b *Doctoral School of Psychology, ELTE Eotvos Lorand University, Budapest, Hungary*

^c*School of Sussex, Brighton, UK*

^e~~Sackler Centre for Consciousness Science~~^e *Psychology, University of Sussex, Brighton, UK*

^d*Singapore Management University, Singapore*

^e*University of New South Wales, Sydney, Australia*

^f*School of Psychological, Social and Behavioural Sciences, Coventry University, UK*

Keywords: Cognitive control, congruency sequence effect, conflict adaptation, domain-general, replication

*Author for correspondence: aczel.balazs@ppk.elte.hu

†Present address: Institute of Psychology, ELTE, Eotvos Lorand University, Izabella u. 46, 1064, Budapest, Hungary

1. Summary

Exploring the mechanisms of cognitive control is central to understanding how we control our behaviour. These mechanisms can be studied in conflict paradigms, which require the inhibition of irrelevant responses to perform the task. It has been suggested that in these tasks the detection of conflict enhances cognitive control resulting in improved conflict resolution of subsequent trials. If this is the case then this so-called congruency sequence effect can be expected to occur in cross-domain tasks. Previous research on the domain-general,ity of the effect presented inconsistent results. In this study, we provide a multi-site replication of three previous experiments of Kan et al. (2013) which test congruency sequence effect between very different domains: from a syntactic to a non-syntactic domain (Experiment 1), and from a perceptual to a verbal domain (Experiment 2 and 3). Despite all our efforts, we found only partial support for the claims of the original study. With a single exception, we could not replicate the original findings, the data remained inconclusive or went against the theoretical hypothesis. We ~~will also~~ discuss the compatibility of the results with alternative theoretical frameworks.

2. Introduction

In day to day life, we often need to override automatic behaviors or habitual responses in order to reach certain goals. Most of us can relate to having to tear ourselves away from appealing foods at the store in order to make a timely supermarket run or to meet certain nutrition goals. One key factor in overriding pre-potent responses is *cognitive control*, which is broadly defined as the collection of processes which contribute to the generation of a goal-relevant response (1). Cognitive control is typically studied using conflict paradigms, which requires the inhibition of irrelevant stimuli, features, or a habitual response to perform the task. One such paradigm is the Stroop task (2).

In a typical Stroop trial, participants are presented with color words in various print colors, but are only tasked with indicating the print color. *Congruent* trials are those in which the word and ink color are the same (e.g., “BLUE” in the color blue); on *incongruent* trials the word and the ink color are different (e.g., “BLUE” in the color red). Typical findings of the Stroop task are that responses on incongruent trials are slower and less accurate compared to congruent trials due to the conflict between the word and its ink color. Since the *Conflict Monitoring Theory* (3), posits that the detection of conflict triggers cognitive control mechanisms¹, these mechanisms can be studied by comparing performance on the incongruent trials with responses on congruent trials. Greater cognitive control is thought to ~~reflect~~ be reflected in smaller ~~differences~~ differences between congruent and incongruent trials in terms of accuracy and speed.

Previous studies have also found that ~~congruence~~ congruency effects are smaller following incongruent trials rather than congruent trials, called the ~~The~~ congruency sequence effect (also known as conflict adaptation or Gratton effect 4). However, Egner (5) proposed that the recruitment of control occurs only following the detection of the same type of conflict (e.g., Stroop conflict). Therefore, this hypothesis suggests that the overlap of stimulus dimensions between the conflict tasks is a central determinant of the transfer of control. If the congruency sequence effect is domain-general then it could be expected that cognitive control can be sustained across seemingly different tasks. For testing this hypothesis, researchers use cross-task adaptation designs in which one type of conflict task (e.g., a verbal Stroop task) is followed by a different type of conflict task (e.g., a non-verbal Flanker task). In case of domain-general, we should expect the effect on cross-tasks as well. Conversely, if the congruency sequence effect is domain-specific, cross-domain adaptation should not be observable.

Some early studies gave support for domain-general (6,7,8, Condition 1), while others have not found the effect (8, Condition 2,9–11), for a summary see Braem et al. (12). One proposed possibility for these mixed findings was that two different conflict sources (e.g., stimulus location and stimulus feature) were simultaneously present in the designs (13). In these cases, it is possible that the cross-task adaptation effect became masked when incongruity in one feature was presented with congruency in the other feature creating weaker conflict signals compared to the cases that induce incongruity in both of the features.

To address these challenges, Kan et al. (13) used a novel design in which “conflict adaptation” was tested between entirely different domains. In their first experiment, participants were given sentences with or without syntactic conflict, followed by a Stroop trial. Syntactic conflict was manipulated using garden path sentences, which are sentences that lure the reader into one interpretation, while a clause at the end

¹ While there are other approaches in which conflict detection/experience is not the direct trigger of control engagement, we deliberate about them only in the Discussion as they are not central to our investigation.

forces the reader to reanalyze the sentence, revealing its true meaning. For example, the sentence: “The basketball player accepted the contract would have to be negotiated”, is ambiguous without reinterpreting the sentence after it is read in its entirety. To eliminate this conflict, inserting “that” after “accepted” would disambiguate the sentence. The results showed that the detection of syntactic conflict on the sentence trials enhanced conflict resolution on the subsequent, non-syntactic Stroop tasks. In their second and third experiment, the Stroop task was preceded by a nonverbal, purely perceptual task. These designs were meant to test whether cross-task adaptation can be generalized beyond the verbal domain. The Necker Cube (14) was also adopted for this study, as it can create ambiguity by inducing two different perceptual percepts. If this bistable stimulus leads to conflicting experience then the congruency sequence effect can be expected in the subsequent Stroop trial. Compared to Stroop trials after stable, unambiguous versions of the cube, accuracy measures reflected enhanced cognitive control after the experience of conflict, providing support for cross-task adaptation. These results provide a strong argument that the detection of conflict in one domain can enhance control mechanisms in a different domain.

Hsu and Novick (15) found evidence for the domain generality of the congruency sequence effect in reversed setting, when following incongruent Stroop trials the listeners’ ability improved in revising temporarily ambiguous spoken instructions that induce brief misinterpretation. Other research also claimed the cross-task adaptation of conflict and showed that the effect is the largest when the tasks depend on the same cognitive-control mechanism (16).

However, a number of studies seem to support the domain or dimensional specificity of the congruency sequence effect. For example, Feldman, Clark, and Freitas (17) analysed event-related potential in a Go/NoGo task using various NoGo decision criteria (i.e. when (not) to go) across the trials. The results of response time, response accuracy, and event-related potential analyses indicated the presence of the congruency sequence effect only when the same NoGo decision criterion was applied across the consecutive trials. Conflict-specificity (or lack of domain-generality) of cognitive control was further supported in several studies (18–23).

The exploration of the neural background of conflict-control also shows an inconsistent set of results. While some studies supported complete domain-specificity (24–26), other others found evidence for domain generality (27), or a hybrid architecture of the two (28).

In summary, although research on the domain-generality of the congruency sequence effect has the potential to reveal important fundamental aspects of cognitive control mechanisms, the empirical data collected provide mixed evidence for cross-domain adaptation effect. Notably, most arguments against domain-generality are based on the absence of evidence, leaving the possibility open that those designs lacked sensitivity or sample size to detect the effect of interest. For these reasons, it is an advisable strategy to investigate the question by replicating a study which had provided empirical support for domain-generality. We decided to conduct a direct replication Kan et al.’s (13) all three experiments for the following reasons:

- (1) these three experiments tested the congruency sequence effect between very different domains: from a syntactic to a non-syntactic domain (Experiment 1), and from a perceptual to a verbal domain (Experiment 2 and 3);
- (2) they provided strong and influential evidence on cross-task adaptation;
- (3) we judged that the methodological parameters of the design allow for the conduct of a direct replication.

3. Methods

3. Materials ~~and Methods~~

Overview

As a direct replication, this experiment closely followed the methods and procedures of the original study. The few minor deviations from the original protocol are highlighted.

All participants performed two tasks: ~~first parallel in a task switching setting~~; either a sentence processing task (Experiment 1), or a perceptual processing task (Experiment 2, 3), ~~then was intermixed with Stroop test trials~~. All the tasks included congruent (unambiguous) and incongruent (ambiguous) stimuli. ~~The different test sequences were pseudo-randomized to create four different conditions to observe the congruency sequence effect status of stimulus sequences~~. The four conditions were determined by the conflict state of the preceding and current stimulus: congruent-congruent (~~CCcC~~); congruent-incongruent (~~CcI~~); incongruent-congruent (~~IcC~~); and incongruent-incongruent (~~IIiI~~) sequences. All tests ended with a question asking whether the participant experienced any technical problems or whether they have any comment on the experiment. [All the materials are openly available on the projects OSF page: https://osf.io/6bd43/.](https://osf.io/6bd43/)

Participants

~~We According to our registration, we did not run any analysis before reaching a minimum sample size, which was analyse the data until we collected at least 2.5 times the sample size of the original experiment (29). That is, which were 103, 70 and 38 participants for Experiment 1, 2 and 3, respectively. After this, we used optional stopping³ and we ceased collecting data for an experiment when all outcome neutral and crucial tests have provided. Our first analyses did not provide good enough evidence, measured by the Bayes factor, for either for H1H0 or H0 (see the Data analysis sections for the specification of these tests); or six months after the in-principle acceptance of the Stage 1 procedure (whichever comes first). After H1, therefore, we have reached the minimum sample size kept collecting data while regularly conducting the analyses³. Since even with our greatly extended sample we run the Bayes factor analysis weekly to check whether the evidence supporting H1 or H0 exceeded did not reach our preset cut-off point. evidential thresholds for the crucial tests (for more details see the Result section of each experiment), we terminated data collection after reaching the end of our timeframe. The first experiment has been conducted as a multisite project, and each contributing lab have been requested to recruit at least 60 participants and ideally 100 before the end of the deadline of data collection.~~

~~As using the original sentence stimuli is central in the replication of Experiment 1, there, we recruited only. Together, 153 native speakers of English ~~in from~~ Australia, Singapore, and the United Kingdom- participated in the first experiment.~~

In Experiment 2 and 3, we tested Hungarian native speakers as the stimuli were not specific to English language. We collected data from 178 and 94 participants, respectively.

Eligibility for participation in each experiment was based on age (≥ 18 years) ~~and~~, being a native speaker of the language of the test, and being right-handed. We did not collect any identifiable private data during the project. Each lab ~~ascertain~~ ascertained that the local institutional ethical review board

³ Note that the Bayes factor retains its meaning irrespective of the used stopping rule, hence using optional stopping did not bias our conclusion (30, 31).

agrees/agreed with the proposed data collection. This replication was conducted in accordance with the Declaration of Helsinki.

Statistical analyses-

Our data ~~was/were~~ nested (i.e. level of trials embedded into the level of participants), therefore, we calculated aggregate scores within conditions for each participant and conducted all of the analyses on the level of the individuals. Hence, our analyses were identical to those of the original paper. We performed and reported all of our hypothesis tests as ~~either paired or independent t tests for the sake of simplicity~~ ANOVAs or paired sample *t* tests⁴. The steps of the analysis of the second and third experiments ~~were similar to those of the first experiment. Therefore, we do not provide a detailed description of the analyses of the second and third experiments, except where they differ from the first experiment.~~

To test our hypotheses, we ~~used the~~calculated Bayes ~~factor/factors~~ (B), which is a measure of relative evidence provided by the data for one model over another one (in our case, for H1 over H0). We used the Bayes factor instead of frequentist statistics as the former one can distinguish between insensitive evidence and evidence for the null, which makes it more appropriate for the purpose of this replication project. We applied the conventional cut off of 3 and 1/3 of B to differentiate good enough evidence for H1 and H0, respectively (3230). We reported *p* values along the Bs for each test. To calculate the Bs, we applied the R script of Dienes and Mclatchie (33) that models the predictions of H1 in raw units rather than in standardised effect sizes. We modelled the predictions of all H1s with half-normal distributions with a mode of zero as all of the alternative hypotheses have directional predictions and they presume that smaller effects are more probable than large ones (34). Considering comparability of the Bs across experiments, we used the parameters of the first experiments to model the predictions of the H1s of the second and third experiments (i.e., we used identical parameters across all experiments for models testing the same hypothesis). The ~~SD~~*sSDs* of the H1 models are specified and justified in the Data Analysis/Results section of Experiment 1.

We notated the Bs as $B_{-H}B_{H(0, x)}$ in which H implies that the distribution is half-normal, 0 indicates the mode of the distribution and *x* stands for the *SD* of the distribution that can vary among models. We implemented all of our analyses with the R statistical software (35). For the NHST analyses, we used \$\alpha = 0.05\$ across all analyses.

As a sensitivity test, we reported analyses on the arcsin transformed proportions, following the original procedure (Supplementary Materials, Table S2).

General data pre-processing

For the reaction time (RT) analyses, we excluded the erroneous trials. After this exclusion, some participants had no correct trials left in at least one of the four conditions (iI, iC, cC, cI). We excluded these participant's responses from both the RT and the accuracy analyses. After these exclusions, we were left with the data of 152 (Experiment 1), 163 (Experiment 2) and 88 (Experiment 3) participants for further analysis. We replaced those Stroop trial RTs with their cutoff value where the RT was 2.5 SDs slower or faster than the participants' overall mean RT. We conducted the accuracy data analyses on the raw proportions.

4. Experiment 1

⁴ Note that this differs from our Stage 1 registration where we planned to perform only *t* tests. We performed ANOVAs for practical reasons, but it has no effect on our results as 2×2 ANOVAs to test the interaction and the main effects lead to the same results as conducting paired *t* tests on the difference and aggregated scores, respectively (See e.g., (35), pp. 63-65).

Procedure

Materials and procedure

Participants completed a task of an intermixed sequence of Stroop ~~test~~ and sentence ~~processing~~ trials. They read the sentences word-by-word, advancing to the next word by pressing the keyboard spacebar. We used the same sentences as the original research which were designed by Garnsey et al. (32), half of which are congruent and the other half incongruent (available on our OSF project site: <https://osf.io/zkm2a/>). Incongruent sentences temporarily mislead the interpretation during the reading process, and participants have to revise the content as they get to the end of the sentence. Congruent sentences are not misinterpretable and they do not need cognitive-control supported reanalysis.

Example sentences from Kan et al. (13):

- “The basketball player accepted the contract would have to be negotiated.” (Temporally incongruent/ambiguous)
- “The basketball player accepted that the contract would have to be negotiated.” (Congruent/unambiguous)

In the incongruent sentence, the verb “accept” can be interpreted as if its direct object is the noun “contract”, however as the participants encounter the “would have” part they have to reanalyse the meaning of the sentence. This feature is expected to increase reading time due to processing difficulty. According to the original paper, this is the central source of conflict in the sentence task.

By inserting “that” in the sentence we diminished the possibility of temporal misinterpretation.

We used a pseudo-randomized sequence of 162 experimental trials (60 congruent and 60 incongruent Stroop trials, as well as 21 congruent and 21 incongruent sentences). 29 filler sentences were included to avoid expectation-driven strategies, and the easy adaptation to the same type of experimental sentences. All filler sentences were unambiguous. Non-filler trials had 4 conditions that described the conflict state of the preceding and the current trial (CC, CI, IC, II; C, I, i, il).

In Stroop-tests, participants used their dominant hand to submit their answers via button presses. Three adjacent keyboard buttons were marked with blue, green and yellow patches. ~~Color~~Colour names matched with the ink ~~color~~colour in congruent trials, and mismatched in incongruent trials, but mismatching ~~color~~colour names were response-ineligible. We used only ~~color~~colour names that are not in the response set (e.g., brown, orange). Incongruencies such as this generate conflict in the representational level, and not the response level (26). We measured ~~response time~~ (RT) in milliseconds and accuracy of the Stroop-trials. In every experiment, the participants performed a practice sequence with all the possible stimulus types before each trial.

Every trial started with a 500 ms fixation period which was followed by either a Stroop, or a sentence stimulus. Stroop stimuli stayed on screen for 1,000 ms. Sentence stimuli started with a full mask (dashes replacing every letter of the sentence) and words appeared by the participants’ spacebar presses on the keyboard. Only one word was visible, the recent parts of the sentence ~~was were~~ also masked. Word reading duration was recorded. After every trial an empty inter-trial interval was shown for 1,000 ms. After the inter-trial interval of certain filler sentences a comprehension probe appeared and remained on the screen until the participant submitted a true or false answer. After the response, a 1,500 ms blank screen preceded the next trial.

Before the experimental sequence, there were 10 Stroop trials which helped participants learn the keyboard mappings for the color responses. It was followed by a baseline trial with 145 intermixed

congruent and incongruent Stroop stimuli. The sentence task was preceded by a comprehension probed filler sentence to demonstrate the step-by-step nature of sentence stimuli. Subsequently, an extra practice sequence was presented with 10 Stroop and 10 filler sentence stimuli pseudo-randomly intermixed. Practice sequences contained only filler stimuli without any incongruency.

Data analysis

Statistical analysis. Our data ~~was~~were nested (i.e. level of trials embedded into the level of participants), therefore, we calculated aggregate scores within conditions for each participant and conducted all of the analyses on the level of the individuals. Hence, our analyses were identical to those of the original paper. We performed and reported all of our hypothesis tests as ~~either paired or independent t tests for the sake of simplicity~~ANOVAs or paired sample t tests⁵. The steps of the analysis of the second and third experiments were similar to those of the first experiment. Therefore, we do not provide a detailed description of the analyses of the second and third experiments, except where they differ from the first experiment.

To test our hypotheses, we ~~used the~~calculated Bayes ~~factor~~factors (B), which is a measure of relative evidence provided by the data for one model over another one (in our case, for H1 over H0). We used the Bayes factor instead of frequentist statistics as the former one can distinguish between insensitive evidence and evidence for the null, which makes it more appropriate for the purpose of this replication project. We applied the conventional cut off of 3 and $\frac{1}{3}$ of B to differentiate good enough evidence for H1 and H0, respectively (3230). We reported *p* values along the Bs for each test. To calculate the Bs, we applied the R script of Dienes and Mclatchie (33) that models the predictions of H1 in raw units rather than in standardised effect sizes. We modelled the predictions of all H1s with half-normal distributions with a mode of zero as all of the alternative hypotheses have directional predictions and they presume that smaller effects are more probable than large ones (34). Considering comparability of the Bs across experiments, we used the parameters of the first experiments to model the predictions of the H1s of the second and third experiments (i.e., we used identical parameters across all experiments for models testing the same hypothesis). The ~~SDs~~SDs of the H1 models are specified and justified in the Data Analysis~~Results~~ section of Experiment 1.

We notated the Bs as $B_{H(0, x)}$ in which H implies that the distribution is half-normal, 0 indicates the mode of the distribution and x stands for the *SD* of the distribution that can vary among models. We implemented all of our analyses with the R statistical software (35). For the NHST analyses, we used $\alpha = 0.05$ across all analyses.

As a sensitivity test, we reported analyses on the arcsin transformed proportions, following the original procedure (Supplementary Materials, Table S2).

Data pre-processing

The data of those participants who experienced any technical problems during the experiment and of those who performed under 70% level of comprehension probe trials were not included in any of the analyses⁶. Therefore, we excluded 20 participants from Experiment 1. To mitigate the influence of ~~outliers~~outlier RTs, we replaced the raw ~~sentence~~reading times of those trials (words) that are 2.5 SDs above the participant's mean ~~sentence~~word reading time across all the ~~sentence~~types~~sentences~~ with the 2.5 SD cutoff value of the participant. In order to account for the biasing effect of the length of the word on the reading time, we calculated the residual reading time for each word. For every sentence-region (see ~~the~~

⁵ Note that this differs from our Stage 1 registration where we planned to perform only *t* tests. We performed ANOVAs for practical reasons, but it has no effect on our results as 2x2 ANOVAs to test the interaction and the main effects lead to the same results as conducting paired *t* tests on the difference and aggregated scores, respectively (See e.g., (35), pp. 63-65).

⁶ This threshold is based on the overall comprehension probe performance of the original experiment.

supplementary materials for the partition of each sentence into sentence regions) in the OSF repository of the project: <https://osf.io/2v39r/>) of each participant we applied a simple linear regression with the actual reading time as the outcome variable and the length of the region in the number of characters as the predictor variable. We included only the congruent ($N = 21$) and the filler sentences ($N = 29$) in the linear regression, as measures of normal reading time. From this calculation, we excluded the reading times of the incongruent sentences as the ambiguity manipulation is expected to make these times longer than the normal reading time. To compute the residual reading time of the sentence-regions of every sentence for each participant we subtracted the predicted reading time of the given sentence-region from the actual reading time. If a sentence region consisted of more than one word, we averaged the residual reading times of the words. For the RT analyses, we excluded the erroneous trials and conducted the same adjustment on the data as we did for the sentence reading times to reduce the effect of the outliers. We conducted the accuracy data analyses on the raw proportions and as a sensitivity test, in the Supplementary Materials, we also reported analyses on the arcsin-transformed proportions, following the original procedure. For a more detailed description see the Supplementary materials. For both of the RT and accuracy analyses, we only included Stroop trials that were preceded by a sentence reading trial.

Results

Outcome neutral tests. We reported the mean and the SD of

Of the 132 participants who scored above or equal to 70% on the sentence comprehension probe test scores of those people who, the mean comprehension score was 0.85 ($SD = 0.1$). Moreover, 145 participants scored above the chance level on the comprehension probe test with a mean comprehension score of 0.82 ($SD = 0.12$).

We tested the presence of the ambiguity effect by comparing the mean residual reading times of congruent and incongruent sentences in each sentence region separately. We expect to find a difference between For the conditions in comparison, we used a one-tailed paired t test where for the temporarily ambiguous and in alternative hypothesis we expected the disambiguating regions mean residual reading time to be greater for the incongruent than for the congruent sentences. To model the predictions of the H_{1s} , we used (as for all following analyses) a half-normal distribution, centered at 0 with SDs taken from the effect sizes of the original study and use it as the SD of the half normal distributions. These values are 40 ms and 18 ms for the temporarily ambiguous and disambiguating regions, respectively. For the other sentence regions (sentence region 1, 2, 5, 6, and 7), we used the more conservative 18 ms raw effect size as our scale as it is less lenient towards the H_0 . We used the data of 132 participants for each sentence region comparison. There was a difference between the congruent and incongruent sentences in residual reading time in the temporarily ambiguous sentence region, with the incongruent one being longer, $M = 17.26$ ms, $B_{H(0, 40)} = 7.87 * 10^6$, $RR [0.6, 6.13 * 10^3]$ ($t(131) = 6.40, p < 0.01$), and for the disambiguating region $M = 7.56$ ms, $B_{H(0, 18)} = 2.88$, $RR [17.20, 1.73 * 10^2]$ ($t(131) = 2.05, p = 0.02$). We also found good enough evidence for a difference in the mean residual reading time in the fifth sentence region but not in the other sentence regions. Figure S1 and Table S1, shows the difference in the mean residual reading time for all sentence regions.

We tested the presence of the Stroop effect in the RT data by comparing the RTs of congruent and incongruent trials. The SD of the distribution modelling the predictions of H_1 were 28 ms, which was the size of the Stroop effect in the original study. We found a Stroop effect with $M = 38.75$ ms, $B_{H(0, 28)} = 4.61 * 10^{13}$, $RR [0.8, 1.45 * 10^4]$ ($F(1, 131) = 87.03, p < 0.01$).

Crucial tests. The two key analyses (i.e., tests of the presence of congruency sequence effect) are the tests of the interaction between the congruency of the current Stroop trial and the congruency of the preceding sentence reading trial on the accuracy RT and on the RT accuracy data. We applied .03 and 30 ms as the SD for the H_1 models RT analysis and .03 SD for the accuracy and for the RT analyses, respectively analysis to model H_1 . Both of the values are equal to the raw effect sizes found in the original

study. We also reported the extent of the Stroop–The congruency sequence effect for the RT analysis was inconclusive $M = 3.88$ ms, $B_{H(0,30)} = 0.37$, RR [0, 33] ($F(1, 131) = 0.35$, $p = 0.56$), while we found good enough evidence for an effect for the accuracy analysis with $M = 0.05$, $B_{H(0,.03)} = 10.60$, RR [.0095, .28] ($F(1, 131) = 6.50$, $p = 0.01$). Figure 1 shows the mean RT results and Figure 2 shows the mean accuracy results, both broken down by the congruency type of the preceding trial and current Stroop trials.

Figure 1. The figure shows the mean RT broken down by the congruency type of the preceding and current trials for Experiment 1. The X axis shows the congruency type of the current trial, while the Y axis shows the mean reaction times. The legend shows the congruency type of the previous sentence trials.

Figure 2. The figure shows the mean accuracy results broken down by the congruency type of the preceding and current trials for Experiment 1. The X axis shows the congruency type of the current trial, while the Y axis shows the accuracy. The legend shows the congruency type of the previous sentence trials.

Supporting tests of interest. ~~Given that we find~~We found evidence for the presence of the interaction between the congruency of the current and preceding trials,~~we ran in the accuracy analysis, therefore, we conducted~~ further comparisons to test whether congruency sequence effect modulated the performance on congruent, on incongruent trials or on both. To this aim, we tested whether the type of the preceding trial modulates the ~~RTs and~~ accuracy rates of current congruent and incongruent trials. To model the H_{1S} , we utilized the parameters of the ~~teststest~~ of the ~~interactions;interaction. For the comparisons, we used a one-tailed paired-sample t test. In the iI and cI trial comparison, we expected the iI trials to have a higher mean accuracy rate, whereas in the cC and iC trial comparison, we expected the cC trials to have a higher mean accuracy rate. The results show that the mean accuracy of the iI trials was higher than cI trials $M = 0.05$, $B_{H(0, .03)} = 960.63$, $RR [.0036, 14.30]$ ($t(131) = 4.21$, $p < 0.01$), while we found evidence for no difference between the cC and iC trials $M = -0.0045$, $B_{H(0, .03)} = 0.30$, $RR [.027, Inf]$ ($t(131) = -0.37$, $p = 0.64$).~~

~~[We will use three categories for our possible outcomes: (1) the pattern of our results fully overlap with the original results; (2) none of our results replicated the original results; and (3) partial replication when some, but not all of our results show the same pattern as the original study. In all cases, we would discuss the results in light of the original hypotheses as well, as it is possible that our results support the hypotheses of the original authors without replicating their results (e.g., the original study did not receive statistical support for the effect for all measures). We will say that our results are in line with the original results when we reach Bayesian positive evidence for originally significant results.]~~

We analysed the extent of the Stroop effect broken down by the congruency type (congruent, incongruent) of the preceding sentence trial for the RT analysis. To model the H_1 in the Bayes factor analysis, we used the same prior as for the Stroop main effect of the current trial ($SD = 28$). The result of the test was inconclusive with $M = 4.28$ ms, $B_{H(0, 28)} = 0.53$, $RR[0, 44]$ ($F(1, 131) = 1.97$, $p = 0.16$).

5. Experiment 2

As the original paper, we also tested domain general congruency sequence effect with the non-verbal Necker Cube task. In this task, participants passively watched congruent and incongruent Necker Cube stimuli. According to Kan et al. (13) and their references (34–36, 35, 37), the passive observation of the bistable Necker Cube induces several alterations of the cube's perceived direction. These alterations are only experiences of a visual conflict and the participants are not instructed to solve this conflict, only to indicate every alteration of the cube's mental direction. The number of direction reversals varied between individuals, therefore the authors of the original paper hypothesised that the scale of congruency sequence effect is dependent on the size of experienced ambiguity shown by the number of reversals.

Materials and procedure

Participants passively viewed congruent and incongruent Necker stimuli pseudo-randomly intermixed with Stroop stimuli. Participants used their non-dominant hands to respond to Necker trials and their dominant hand to respond to Stroop trials. This allowed them to respond to trials without any movement across the keyboard. On incongruent Necker stimulitrials, participants indicated with button presses how they perceive the direction of the cube (right-downward, or left-upward) and they pressed the according button every time they experienced a direction change. On congruent Necker stimulitrials, participants were asked to indicate the direction of the stimulus as quickquickly as possible. The two corresponding keyskeyboard buttons were labelled with stickers depicting the two directions of the Necker cube.

Before the experimental part 10 Stroop and 7 Necker stimuli were shown to get the participants used to the two tasks. A baseline block was presented with 54 intermixed congruent and incongruent Stroop trials.

In the experimental block, two hundred total trials were included. The above-mentioned four experimental conditions were generated by a pseudo-randomized sequence of 48 congruent Necker stimuli (24 downward, 24 upward), 28 incongruent Necker stimuli (Figure 43), 62 congruent and 62 incongruent Stroop stimuli. The original study used a pseudo-randomized order of Necker and Stroop Stimuli in the experimental block. As we did not know the order of the trials in the original study, we created 6 blocks of pseudo-randomized sequences. Each participant was randomly assigned to one of the sequences. Each trial began with a 500 ms fixation period and it was followed by either a Stroop or a Necker stimulus. Stroop trial timings were trials stayed on the samescreen for 1,000 ms as in experimentExperiment 1. Incongruent Necker stimuli arewere shown for 90 seconds. During this period participants were instructed to press the matching button whenever their directional interpretation changes. Congruent Necker stimuli remained on screen for 1,000 ms and every trial was followed by a 1,000 ms inter-trial interval.

Figure 43. The images of the three Necker stimuli (from left to right: incongruent, congruent upward, and congruent downward).

Data analysispre-processing

The steps of the analysis of the second experiment were similar to those of the first experiment. Considering comparability of the Bs across experiments, we used the parameters of the first experiments to model the predictions of the H₁s of the second experiment (i.e., we had identical SDs across experiments

for models testing the same hypothesis). Differences compared to the first experiments are the following: we assigned participants to high and low reversal groups based on a median-split of the mean number of reversals they experienced on incongruent Necker trials (we. The mean number of reversals for the high reversal group was 18.61 ($SD = 15.92$), while in the low reversal group it was 3.10 ($SD = 2.25$). Based on the number of reversals in the original study, it was expected to see a difference of 16 between low and high reversal groups), whereas we found a difference of 15.51. We conducted the analyses only on those Stroop trials that directly followed Necker trials. On top

Results

Outcome neutral tests.

We found a Stroop effect in the RT analysis with $M = 44.60$ ms, $B_{H(0, 28)} = 8.08 \cdot 10^{18}$, $RR[0.69, 1.70 \cdot 10^4]$ ($F(1, 162) = 122.98, p < 0.01$).

Crucial tests.

The results of the two-way interactions (preceding and current trial type), as part of the crucial tests, were inconclusive with $M = 12.6$ ms, $B_{H(0, 30)} = 1.81$, $RR[0, 180]$ ($F(1, 162) = 2.95, p = 0.09$) for the RT analysis and with $M = -0.0019$, $B_{H(0, .03)} = 0.50$, $RR[0, .05]$ ($F(1, 162) = 0.01, p = 0.92$) for the accuracy analysis. Figure S2 and Figure S3 in the Supplementary materials show the analysis of the two-way interactions for the RT and accuracy, respectively.

We tested three-way interactions including the reversal group variable on the accuracy and RT data. The results of the three-way interaction provides RT analysis showed evidence for H_1 then we will run separate analyses of the two-way interactions for the low and high no effect $M = -10.90$ ms, $B_{H(0, 30)} = 0.28$, $RR[24.3, Inf]$ ($F(1, 161) = 0.55, p = 0.46$), while the accuracy analysis was inconclusive with $M = -0.05$, $B_{H(0, .03)} = 0.45$, $RR[0, 0.05]$ ($F(1, 161) = 1.82, p = 0.18$). Figure 4 shows the mean RT results and Figure 5 shows the mean accuracy results, both broken down by the congruency type of the preceding sentence and current Stroop trials for the high and low reversal groups.

Figure 4. The figure shows the mean RT broken down by the congruency type of the preceding and current trials in the high and the low reversal groups for Experiment 2. The X axis shows the

congruency type of the current trial, whereas the Y axis shows the mean reaction times. The legend shows the congruency type of the previous sentence trials.

Figure 5. The figure shows the mean accuracy results broken down by the congruency type of the preceding and current trials in the high and the low reversal groups for Experiment 2. The X axis shows the congruency type of the current trial, while the Y axis shows the accuracy. The legend shows the congruency type of the previous sentence trials.

Supporting tests of interest.

We tested the extent of the Stroop effect broken down by the congruency type (congruent, incongruent) of the preceding sentence trial for the RT analysis. We found the effect of the congruency of the previous trial with \$M = 46.59\$ ms, \$B_{H(0, 28)} = 1.26 \cdot 10^{11}\$, \$RR[1.08, 1.73 \cdot 10^4]\$ (\$F(1, 162) = 65.84\$, \$p < 0.01\$. RTs were slower when the preceding trial type was incongruent regardless of the current trial congruency type.

Finally, as a supporting analysis, we estimated the extent to which perceived ambiguity (measured as the mean number of experienced reversals) and congruency sequence effect (H—Chi - ci of accuracy rates and RTs) correlated. For these analyses To estimate the effect size, we reported the calculated a Pearson correlation with the 95% Bayesian Credibility Intervals calculated with Interval assuming a uniform prior distributionsdistribution. The plausible effect sizes are small for the RT analysis, \$r = -0.02\$, \$CI_{95\%} [-0.17, 0.13]\$, and for the accuracy analysis as well, \$r = 0.06\$, \$CI_{95\%} [-0.10, 0.21]\$. The correlation between mean experienced reversals and the congruency sequence effect is shown on Figure S4 for the RT analysis and Figure S5 for the accuracy analysis in the Supplementary materials.

6. Experiment 3

Experiment 3 is identical to experimentExperiment 2 regarding the timings and the order of trials; however, the 90 second incongruent Necker stimulus was replaced by 90 secondssecond of periodically switching congruent Necker stimuli (upward and downward). It was an attempt This design aimed to

make the ~~sequence similar to~~ number of reversals experimentally controlled for the ~~behaviour of an incongruent Necker cube's self-reported reversals~~ cube trials. The authors of the original paper used the results of Experiment 2 to define the frequency of congruent Necker stimuli. In the original research, high reversal participants' average frequency ~~of reversals~~ was 27.6 so we showed 28 changes in direction ~~by every~~ for each 90 ~~seconds~~ second period. ~~We divided the 90 second time period into 28 time-intervals with random length (min = 0.289 second; max = 7.445 second) and changed the direction of the unambiguous Necker cube in each time interval. We used the same time intervals throughout the study for all participants. Each participant saw the left facing unambiguous Necker cube first. Participants completed the same practice and the baseline~~ ~~sequence~~ sequences as ~~well, as in~~ ~~experiment~~ Experiment 2.

Data ~~analysis~~ pre-processing

~~The procedure of the analysis closely followed that of the first experiment. Again, to compute the Bs, we informed the models of H1s using the expected effect sizes of the first experiment. A deviation from the first experiment is that here, instead of evaluating sentence reading performance, we~~ We assessed the level of attention to the Necker cube trials and, using the cutting point of the original study, we excluded the data of participants whose performance was below 70% on accurately identifying whether or not a stimulus change occurred on the current trial. ~~Therefore, we excluded 8 participants from further analysis, and we were left with 80 participants after all the exclusions.~~ We included only those Stroop trials ~~to~~ in the analyses that were preceded by a Necker trial.

Results

Outcome neutral tests.

~~As an outcome neutral test, we calculated the Stroop effect of the current trial for the RT analysis. We found an effect with good enough evidence~~ $M = 44.29$, $B_{H(0, 28)} = 7.40 \cdot 10^7$, $RR[1.5, 1.58 \cdot 10^4]$ ($F(1, 79) = 51.74$, $p < 0.01$).

Crucial tests.

~~We conducted two-way interactions to test the congruency sequence effect as part of the crucial tests. Both the RT analysis~~ $M = 1.67$, $B_{H(0, 30)} = 0.36$, $RR[0, 32]$ ($F(1, 79) = 0.03$, $p = 0.87$), and the accuracy analysis $M = 0.03$, $B_{H(0, 0.03)} = 1.63$, $RR[0, 0.28]$ ($F(1, 79) = 1.47$, $p = 0.23$) yielded inconclusive results. Figure 6 shows the mean RT results and Figure 7 shows the mean accuracy results of the test of the congruency sequence effect.

Figure 6. The figure shows the mean RT broken down by the congruency type of the preceding and current trials for Experiment 3. The X axis shows the congruency type of the current trial, while the Y axis shows the mean reaction times. The legend shows the congruency type of the previous sentence trials.

Figure 7. The figure shows the mean accuracy results broken down by the congruency type of the preceding and current trials for Experiment 3. The X axis shows the congruency type of the current trial, while the Y axis shows the accuracy. The legend shows the congruency type of the previous sentence trials.

Supporting tests of interest.

As a supporting test of interest, we tested the main effect of the congruency of the Previous Necker cube trial on the extent of the Stroop effect on RT. We found an effect with $M = 74.93$ ms, $B_{H(0, 28)} = 1.99 \times 10^{12}$, $RR[1.8, 27740]$ ($F(1, 79) = 98.12, p < 0.01$)

7. Discussion

[Here we will discuss our results and their relevance to the different models of control adaptation.

In addition, we plan to relate our findings to alternative accounts to Conflict Monitoring Hypothesis such as working memory triggered dynamic adjustment (e.g., Jha & Kivonage, 2010); the affective signal hypothesis (Dreisbach & Fisher, 2015); or a dynamic goal state explanation (Scherbaum et al., 2012). We conducted the present replication study to investigate the question of domain-generality of the congruency sequence effect. By replicating the design of Kan et al. (13), we expected that the results would reveal the strength of evidence behind a prominent claim of domain-generality.

For Experiment 1, where Stroop task trials followed garden-path sentences, we collected data from three countries with nearly four times the sample size of the original study. The data showed that the participants complied with the instructions, reflected by the presence of the Stroop effect and the longer RTs in the ambiguous regions of the incongruent sentences compared to the same region of the congruent sentences. The crucial tests brought partial support for the cross-task congruency sequence effect. While the RT results were inconclusive, nearly reaching our threshold for evidence for no-effect, the accuracy results were in line with the original finding as the Bayesian analysis showed evidential support for the effect. Follow-up analyses showed that this finding was the result of increased accuracy on incongruent trials when they came after incongruent sentences compared to when they came after congruent ones.

In Experiment 2, where the sentence task was replaced by Necker cubes in the design, we collected data from more than six times the original sample size. Still, the results remained inconclusive for both the RT and the accuracy tests of the congruency sequence effect. Furthermore, we found good enough evidence for no effect in the RT results when we tested whether perceiving more Necker cube reversals led to a greater congruency sequence effect.

Experiment 3, where the orientation of the Necker cube was periodically switched by the testing program, the analyses again brought inconclusive results, the RT and accuracy data did not support the presence of the cross-task incongruency sequence effect. In fact, the RT results nearly passed our preset evidential threshold for no-effect.

Summarizing our findings, first we can conclude that our replication was successful in the sense that we could conduct the experiments according to our plan. Our manipulation checks testified that our participants understood and followed the instructions and they produced the expected default Stroop effect. Second, we found that our crucial tests brought only partial support for the claims of the original study. As Table 1 shows, with a single exception, we could not replicate the original findings, the data remained inconclusive in respect to the competing hypothesis.

Table 1. Summary of Our Findings

Experiment	CSE present for	
	RT	Accuracy
Exp. 1	=	✓
Exp. 2	=	=
Exp. 2 (Reversals)	×	=
Exp. 3	=	=

Note: ✓ = support for the presence of the congruency sequence effect (CSE); × = support against the presence of the CSE; = = inconclusive findings.

The results are surprising from several aspects. Should the original results not be chance findings, one would expect more supportive evidence in this study, given our increased sample size. Bayes factor analyses indicate that the data, with one exception, were far from supporting the alternative hypothesis, if anything, they were leaning towards the null hypothesis. This brings us to our supportive finding, the evidence for congruency sequence effect in the accuracy results of Experiment 1.

The theoretical interest in this supportive finding is whether the observed effect in the garden-path task can indicate the cross-domain nature of the congruency sequence effect. That is, whether readers' detection of syntactic conflict enhanced cognitive control in another domain. As we discussed it in the Introduction, the domain-specificity account assumes that overlap of stimulus dimensions between the conflict tasks is a central determinant of the transfer of control (5, 8, 12). Should we regard the sentence-processing task and the Stroop task to differ in their domain (syntactic vs. non-syntactic) then our results of Experiment 1 provide partial evidence that control adaptation can work across domains. The interpretation is weaker if we regard the two more similar in nature. The original authors themselves pointed out that despite the difference in stimulus characteristics, both tasks are verbal. In fact, this motivated them to test the effect between perceptual and verbal domains in Experiment 2 and 3. If the effect is domain-general, then it remains a question regarding why the incongruent Necker cube did not decrease the congruency effect. Either way, we think that the positive result of Experiment 1 is noteworthy for the theory and it, together with the lack of support for the null in other analyses, prevents us from completely rejecting the domain-general hypothesis of the congruency sequence effect.

The results brought up some new questions for further investigations. First of all, it remains unexplained why RT data of Experiment 1 were in sharp contrast with the original findings, being very far from detecting the congruency sequence effect between the two tasks. One could speculate that enhanced control after incongruent sentence trials manifested in participants' higher motivation for accuracy which could increase the time needed to spend on the incongruent Stroop trials, counteracting any speeding effect. However, our pattern of results does not support this speculation, as after the incongruent trials the mean RTs are not slower than after the congruent trials. Alternatively, if the effect is domain-specific, then the positive accuracy results seek an explanation of a different scope.

It is worth noting that investigating the evidence for the cross-task congruency sequence effect is interesting not just for the domain-generality debate, but it is also relevant to the research on the mechanisms of cognitive control. According to the *Conflict monitoring Theory* (3) the detection of conflict is what triggered subsequent cognitive control. In contrast, the *Affective Signaling Hypothesis* (38) suggests that it is not conflict per se that is responsible for control adaptation, but the negative affect that this conflict triggers. In this view, the experience of conflict evokes a negative affective reaction and this affect is what elicits the performance-monitoring system to upgrade attentional resources leading to the attenuation of further conflict. From this alternative framework, it is plausible that reading incongruent garden-path sentences causes phasic negative affect, so we would expect control adaptation on subsequent trials. It is less clear, however, whether watching incongruent Necker cubes is a source of negative affect. Perhaps, the lack of elicited negative affect in that design is behind the complete lack of evidence for the congruency sequence effect in the Necker cube experiments. Future research should consider applying cross-task designs to test the predictions of the Affective Signaling Hypothesis.

In sum, this study provides a registered replication of three experiments testing the congruency sequence effect between two tasks. We could not replicate the original positive results, except in one measure in Experiment 1. In our interpretation, this pattern of findings weakens, but does not fully reject the hypothesis that the effect can occur across domains. We highlight that failure to detect control adaptation after the Necker cube might reflect that the two designs differ not just in the domain of the trigger trial but also in other characteristics, such as the affect the trigger trials induce, drawing attention to alternative frameworks of control adaptation.

Ethical Statement

Each lab ascertains that the local institutional ethical review board agrees with the proposed data collection. This replication was conducted in accordance with the Declaration of Helsinki. We did not collect any identifiable private data during the project.

Funding Statement

Balazs Aczel was supported by the János Bolyai Research Fellowship from the Hungarian Academy of Sciences. ~~Bence Palfi is grateful to the Dr Mortimer and Theresa Sackler Foundation which supports the Sackler Centre for Consciousness Science.~~

Data Accessibility

All of our analyses will be publicly preregistered on the OSF site after Stage 1 “in principle” acceptance. Collected raw and processed data will be publicly shared on the OSF page of the project. Code for data management and statistical analyses will be written in R and will be made open access. All materials of the three experiments are available through OSF: <https://osf.io/6bd43/>.

Competing Interests

We have no competing interests.

Authors' Contributions

~~[A listing of the specific contribution of each author will be added at the completion of the study.]~~

Conceptualization: Balazs Aczel, Marton Kovacs, and Bence Palfi.

Data Curation: Marton Kovacs and Miklos Bogнар.

Formal Analysis: Marton Kovacs and Bence Palfi.

Investigation: Balazs Aczel, Marton Kovacs, Miklos Bogнар, Andree Hartanto, Sandersan Onie, Lucas E. Tiong, and Thomas R. Evans.

Methodology: Balazs Aczel, Marton Kovacs, Bence Palfi, and Miklos Bogнар.

Project Administration: Balazs Aczel and Miklos Bogнар.

Resources: Balazs Aczel, Marton Kovacs, and Miklos Bogнар.

Software: Marton Kovacs and Miklos Bogнар.

Supervision: Balazs Aczel.

Validation: Marton Kovacs and Bence Palfi.

Visualization: Marton Kovacs.

Writing - Original Draft Preparation: Balazs Aczel, Marton Kovacs, Bence Palfi, and Miklos Bogнар.

Writing - Review & Editing: Balazs Aczel, Marton Kovacs, Bence Palfi, Miklos Bogнар, Andree Hartanto, Sandersan Onie, Lucas E. Tiong, and Thomas R. Evans.

References

- Gratton G, Cooper P, Fabiani M, Carter CS, Karayanidis F. Dynamics of cognitive control: Theoretical bases, paradigms, and a view for the future. *Psychophysiology*. 2018;55(3):e13016.
- Stroop JR. Studies of interference in serial verbal reactions. *J Exp Psychol*. 1935;18(6):643.
- Botvinick MM, Braver TS, Barch DM, Carter CS, Cohen JD. Conflict monitoring and cognitive control. *Psychol Rev*. 2001;108(3):624–52.
- Gratton G, Coles MG, Donchin E. Optimizing the use of information: Strategic control of activation of responses. *J Exp Psychol Gen*. 1992;121(4):480–506.
- Egner T. Multiple conflict-driven control mechanisms in the human brain. *Trends Cogn Sci*. 2008;12(10):374–80.
- Freitas AL, Bahar M, Yang S, Banai R. Contextual adjustments in cognitive control across tasks. *Psychol Sci*. 2007;18(12):1040–3.
- Kunde W, Wühr P. Sequential modulations of correspondence effects across spatial dimensions and tasks. *Mem Cognit*. 2006;34(2):356–67.
- Notebaert W, Verguts T. Cognitive control acts locally. *Cognition*. 2008;106(2):1071–80.
- Akçay Ç, Hazeltine E. Domain-specific conflict adaptation without feature repetitions. *Psychon Bull Rev*. 2011;18(3):505–11.
- Funes MJ, Lupiáñez J, Humphreys G. Analyzing the generality of conflict adaptation effects. *J Exp Psychol Hum Percept Perform*. 2010;36(1):147–61.
- Kiesel A, Kunde W, Hoffmann J. Evidence for task-specific resolution of response conflict. *Psychon Bull Rev*. 2006;13(5):800–6.
- Braem S, Abrahamse EL, Duthoo W, Notebaert W. What determines the specificity of conflict adaptation? A review, critical analysis, and proposed synthesis. *Front Psychol*. 2014;5.
- Kan IP, Teubner-Rhodes S, Drummey AB, Nutile L, Krupa L, Novick JM. To adapt or not to adapt: The question of domain-general cognitive control. *Cognition*. 2013;129(3):637–51.
- Necker LA. LXI. Observations on some remarkable optical phenomena seen in Switzerland; and on an

optical phenomenon which occurs on viewing a figure of a crystal or geometrical solid. *Lond Edinb Dublin Philos Mag J Sci*. 1832;1(5):329–37.

15. Hsu NS, Novick JM. Dynamic engagement of cognitive control modulates recovery from misinterpretation during real-time language processing. *Psychol Sci*. 2016;27(4):572–82.

16. Freitas AL, Clark SL. Generality and specificity in cognitive control: conflict adaptation within and across selective-attention tasks but not across selective-attention and Simon tasks. *Psychol Res*. 2015;79(1):143–62.

17. Feldman JL, Clark SL, Freitas AL. Conflict adaptation within but not across NoGo decision criteria: Event-related-potential evidence of specificity in the contextual modulation of cognitive control. *Biol Psychol*. 2015;109:132–40.

18. Braem S, Hickey C, Duthoo W, Notebaert W. Reward determines the context-sensitivity of cognitive control. *J Exp Psychol Hum Percept Perform*. 2014;40(5):1769–78.

19. Forster SE, Cho RY. Context specificity of post-error and post-conflict cognitive control adjustments. *PLoS One*. 2014;9(3):e90281.

20. Kunde W, Augst S, Kleinsorge T. Adaptation to (non) valent task disturbance. *Cogn Affect Behav Neurosci*. 2012;12(4):644–60.

21. Lee J, Cho YS. Congruency sequence effect in cross-task context: evidence for dimension-specific modulation. *Acta Psychol (Amst)*. 2013;144(3):617–27.

22. Scherbaum S, Frisch S, Holfert A-M, O’Hora D, Dshemuchadse M. No evidence for common processes of cognitive control and self-control. *Acta Psychol (Amst)*. 2018;182:194–9.

23. Schlaghecken F, Refaat M, Maylor EA. Multiple systems for cognitive control: Evidence from a hybrid prime-Simon task. *J Exp Psychol Hum Percept Perform*. 2011;37(5):1542–53.

24. Egner T, Delano M, Hirsch J. Separate conflict-specific cognitive control mechanisms in the human brain. *neuroimage*. 2007;35(2):940–8.

25. Liston C, Matalon S, Hare TA, Davidson MC, Casey BJ. Anterior cingulate and posterior parietal cortices are sensitive to dissociable forms of conflict in a task-switching paradigm. *Neuron*. 2006;50(4):643–53.

26. Van Veen V, Carter CS. Separating semantic conflict and response conflict in the Stroop task: a

functional MRI study. *Neuroimage*. 2005;27(3):497–504.

27. Peterson BS, Kane MJ, Alexander GM, Lacadie C, Skudlarski P, Leung H-C, et al. An event-related functional MRI study comparing interference effects in the Simon and Stroop tasks. *Cogn Brain Res*. 2002;13(3):427–40.

28. Jiang J, Egner T. Using neural pattern classifiers to quantify the modularity of conflict-control mechanisms in the human brain. *Cereb Cortex*. 2013;24(7):1793–805.

29. Simonsohn U. Small telescopes: Detectability and the evaluation of replication results. *Psychol Sci*. 2015;26(5):559–69.

30. Dienes Z. How Bayes factors change scientific practice. *J Math Psychol*. 2016;72:78–89.

31. Rouder JN. Optional stopping: No problem for Bayesians. *Psychon Bull Rev*. 2014;21(2):301–308.

32. Garnsey SM, Pearlmuter NJ, Myers E, Lotocky MA. The contributions of verb bias and plausibility to the comprehension of temporarily ambiguous sentences. *J Mem Lang*. 1997;37(1):58–93.

33. Dienes Z, Mclatchie N. Four reasons to prefer Bayesian analyses over significance testing. *Psychon Bull Rev*. :1–12.

34. Toppino TC, Long GM. Top-down and bottom-up processes in the perception of reversible figures: Toward a hybrid model. In: *Dynamic cognitive processes*. Springer; 2005. p. 37–58.

35. Liebert RM, Burk B. Voluntary control of reversible figures. *Percept Mot Skills*. 1985;61(3_suppl):1307–10.

36. Abelson RP. *Statistics as principled argument*. Hillsdale, NJ: L.

37. Porter ELH. Factors in the fluctuation of fifteen ambiguous phenomena. *Psychol Rec*. 1938;2(8):231–53.

38. Dreisbach G, Fischer R. Conflicts as aversive signals for control adaptation. *Current Directions in Psychological Science*. 2015 Aug;24(4):255-60.

Appendix D

Review of “Is there evidence for cross-domain control adaptation? A replication of Kan et al. (2013)”

Stage 2 Primary Criterion #1: Whether the introduction and methods are the same as the approved Stage 1 submission

Strictly speaking, the introduction differed from the approved Stage 1 submission (based on the Track Changes version provided by the authors). However, the changes were minor wording changes that did not impact the meaning of the text.

There were methodological differences between the approved Stage 1 submission and the Stage 2 submission.

- First, additional data cleaning procedures were introduced in the current submission. With that said, the data cleaning procedures are in line with common practice in the field.
- Second, the authors noted three additional deviations in the Supplementary Materials. 1) The classic Stroop effect was moved from a “crucial” to “supporting” test. This change is appropriate because the Stroop effect was not part of the critical hypotheses of the original paper. 2) The authors did not collect participants’ gender or age so as to allow anonymous data collection (perhaps due to IRB and/or practical data collection concerns). This is unfortunate, because it precludes comparison of the demographics of the original sample to the replication sample. The authors should mention this as a limitation in the Discussion. 3) The authors conducted ANOVAs and follow-up t-tests rather than t-tests only. This change is appropriate because it brings the analyses in line with the original paper and the conflict congruency sequence effect literature, which typically examines the interaction of preceding and current trial type.
- Third, an examination of the raw data file on OSF’s website reveals something unusual and unexpected in the baseline block of Stroop trials. (Note: I only looked at Experiment 1, so I’m not sure if this also applies to Experiments 2 & 3.) If I understand the file structure correctly, it appears that many of the trials within the baseline block did not use color words (e.g., tent, house, hungry). The inclusion of these trials was not mentioned in the manuscript. Given the extensive literature documenting the influence of incongruent trials proportion on Stroop performance, this design choice needs to be acknowledged and justified. Furthermore, how the inclusion of these neutral stimuli may have affected subjects’ expectations of the experimental task and the results should also be addressed.

Stage 2 Primary Criterion #2: Whether the authors’ conclusions are justified given the data.

Several major issues were identified in the authors’ conclusions.

- Broadly speaking, the authors did not provide adequate discussion of the discrepancies in findings between the two studies. For example, the authors only provided a cursory explanation of the contrasting patterns in RT data in Experiment 1. Another possibility for the difference may lie in the differences in the samples. In the current study, the participants are sampled from 3 different dialects of English, whereas the original study tested only Mainstream American English speakers (which is not represented in the

current sample). This may or may not matter, but since the RT effects did not replicate, one wonders whether different dialects have distributions of the same verbs in different grammatical environments, generating various biases that speakers of the different dialects use to guide comprehension. If so, it is possible that these differences would produce different ambiguity effects that may interact with the need for cognitive control in different ways (e.g., speakers of dialect X have a larger garden-path effect than speakers of dialects Y and Z). Perhaps the self-paced reading time is not sensitive enough to detect these kinds of differences, but it would be worth discussing such possibilities. Additionally, the authors ignored large differences between the samples in overall accuracy on the Stroop task in Experiment 1. The original study observed 97%, 94%, 97%, and 97% accuracy in CC, CI, IC, and II conditions, respectively. The replication reported (approximately) 87%, 82%, 87%, and 87% accuracy in the same conditions. Therefore, overall accuracy was about 10% lower in the current study. This could reflect differences in inhibitory control ability between the 2 participant groups (the original sample consisted exclusively of college students at a private university in the U.S.) or different strategies used during the Stroop task (i.e., placing the emphasis on accuracy versus speed). These accuracy differences could have affected the RT results—if participants in the present study were responding too quickly to maintain high accuracy, then engagement of cognitive control may have slowed responding to improve accuracy (akin to a post-error slowing effect).

- Furthermore, the current replication is insufficiently contextualized. The introduction is unusually light on literature reviews (which I also noted in the Stage 1 review). Apart from describing the Kan study, unfortunately, only one follow-up study was summarized (Hsu & Novick, 2016), but there have been several other cross-task replications and extensions since the original Kan article, particularly in the area of sentence processing (e.g., Thothathiri et al., 2018 in *Cognition*; Hsu, Kuchinsky, & Novick, 2020 in *Language, Cognition, and Neuroscience*; Adler, Valdes Kroff, & Novick, 2020 in *JEP:LMC*; Navarro Torres et al., 2018 in *Brain Research*). All of this other work details careful theoretical ideas and predictions about shared processing demands across tasks, which this attempted replication does not. Perhaps it does not need to since it is a registered report, but minimally the other work should be considered with respect to their specific accounts and how they contribute to the notions that the current authors test. Two particularly relevant studies are Adler et al., 2020 and Hsu et al., 2020. In Adler, they observe that the flanker-conflict effect is reduced following a code-switched sentence as opposed to a monolingual sentence, suggesting that the cross-linguistic conflict detected during processing/integration of a code-switch engages cognitive control. The direction of the effect is the same as that in Kan et al., which the current authors try to replicate. It also uses Flanker, which is non-verbal and weighs-in on the domain-general issue the current authors address. Similarly, Hsu et al., 2020 show that both perceptual conflict and non-verbal conflict engage control mechanisms that facilitate syntactic revision using fine-grained eye-movement measures. This paper addresses both the domain-general and perceptual issues the current authors address. Thus, it is notable that none of this is reviewed or discussed in terms of the current replication attempt.

- In Kan et al.'s Experiment 2, the key metric used to index conflict experienced by the participants is the number of reversals, with the key finding being a positive relationship between number of reversals experienced and magnitude of conflict adaptation effect. Furthermore, in conjunction with findings of Experiment 3, Kan and colleagues suggested that conflict experienced is crucial to the observation of conflict adaptation effects. Thus, it seems that one of the most apparent explanations for the current replication failure likely lies in the substantially lower number of reversals experienced by the current sample. However, this issue was neither noted nor discussed. Indeed, the number of reversals experienced by the current "high reversal" group ($M = 18.6$, $SD = 15.9$) is similar to the number of reversals experienced by the "low reversal" group in the Kan et al. paper ($M = 16.3$, $SD = 3.9$). A rough comparison based on an independent samples t-test revealed that mean number of reversals experienced by these two groups failed to reach significance, $t [93] = 0.538$, $p = .592$. (Note: the actual number of subjects in this group in the current sample was not reported; the t-test was conducted assuming 81 subjects). In sum, since "low reversal" subjects in Kan et al. did not experience conflict adaptation, it is unsurprising that the current "high reversal" sample also did not display a conflict adaptation effect. After all, if the participants did not experience sufficient conflict (as Kan et al. hypothesized), they would not be expected to engage cognitive control that would result in conflict adaptation.
 - Among the high reversal subjects, participants in the Kan et al. study experienced significantly more reversals ($M = 31.8$, $SD = 8.3$) than subjects in the current sample ($M = 18.6$, $SD = 15.9$), $t [93] = 3.03$, $p = .003$. Similarly, the low reversal participants in the Kan et al. study also experienced a higher number of reversals ($M = 16.3$, $SD = 3.9$) than the current sample ($M = 3.1$, $SD = 2.3$), $t [93] = 17.91$, $p < .001$.
- The presentation of rationale and results for Experiment 3 was misleading. As stated in Kan et al., the goal of Experiment 3 was to examine an alternative effort-related explanation for the results of Experiment 2. That is, "in Experiment 2, one could argue that, compared to congruent Necker trials, incongruent Necker trials were more effortful and demanded more attention because they lasted for a longer duration and required multiple responses throughout a trial. We ... [hypothesized] that greater effort alone, in the absence of conflict, should not trigger cognitive control functions that operate across trials. In other words, the experience of internal conflict should be critical to showing adaptation effects; without it, behavioral adjustments are not expected to occur." Thus, the expectation is that conflict adaptation will not be observed in Experiment 3, where internal conflict is not induced. And the findings were in line with these expectations. However, this rationale was not clearly explained in Experiment 3. Without this context, a reader unfamiliar with the Kan et al. paper may misinterpret this pattern as a replication failure rather than a pattern of findings that in fact aligned with the hypothesis from Kan et al.

In sum, given the major concerns described above, the current manuscript did not meet this review criterion.

Stage 2 Secondary Criterion #1: *Whether the data are able to test the authors' proposed hypotheses by passing the approved outcome-neutral criteria (such as absence of floor and ceiling effects or success of positive controls or other quality checks). Failure to pass these conditions will not lead to manuscript rejection but could require authors to explicitly acknowledge such limitations in their discussion, and in severe cases could also lead to the publication of an accompanying editorial cautionary note.*

Please see other comments.

Other major concerns:

The paper uses “attention control” and “cognitive control” interchangeably, and it is not clear why when other work draws distinctions (for a lengthy discussion, including empirical data on the difference, see abovementioned Hsu et al., 2020).

There should also be discussion of work by those in the conflict adaptation arena that illustrates the conditions under which the effect is found and the conditions under which it is not (e.g., when subjects perceive the two tasks as separate, creating a “task set”). See work by Daniel Weissman's lab for inspiration (Grant et al., 2020). Discussions of these important issues will further contextualize the current findings and will strengthen this work's potential contribution to the field.

The authors state that the demarcation of sentence regions is included in the OSF repository (p. 6, ln 58). However, I could not locate this information. This is relevant for concerns regarding the computation of residual reading times (below).

The authors provided additional methodological detail about the computation of residual reading times in the Supplementary Materials. This raised concerns about the computation, which deviated substantially from common practice and the original work. 1) The authors included congruent and filler sentences and excluded incongruent sentences in the linear regression for residual reading times, whereas the original work included congruent and incongruent sentences but excluded filler sentences. The filler sentences have a different syntactic structure and are not comparable to the congruent/incongruent sentences, and so should not be included in this computation. 2) It is not clear that the authors compare the correct regions of the sentences. They state, *“As the sentences have different lengths, some sentence region's mean residual reading times were calculated from more data points than others. As the congruent sentences have more sentence regions than the incongruent sentences the last two sentence regions for congruent sentences did not have an incongruent pair to do the comparison, therefore, we excluded these sentence regions from further analysis”* (Supp. Mat., p. 3). The congruent and incongruent sentences in the original study were designed to have the same number of words (11, after excluding ‘that’ from congruent sentences). Therefore, sentence length should not have affected the number of contributing data points and congruent sentences should not have had more regions than the incongruent sentences. This difference may have been due to the inclusion of filler sentences in the calculation. Additionally, authors state, *“we compared the fourth sentence region of the incongruent sentences to the third sentence region of the congruent sentences”* (Supp. Mat., p. 3). This should be reversed, if the numbers are assigned based on word position with the additional word “that” in congruent sentences. Additionally, more than just these two

regions were compared (ambiguous region, disambiguating region, post-disambiguating region, etc.). Although this analysis is “outcome-neutral,” it is nevertheless important because residual reading times are used to index the ambiguity effect, which is expected to relate to the magnitude of congruency sequence effects (i.e., greater experience of conflict results in larger adaptation). The authors report smaller ambiguity effects than the original study (17.26 ms and 7.56 ms vs 40 ms and 18 ms), which could have resulted in smaller CSEs. However, this apparent discrepancy could also be due to their unconventional computation of residual reading times and/or different assignment of sentence regions.

Appendix E

Dear Prof. Chambers,

We are happy to submit a revised version of our Stage 2 RRR manuscript to the Royal Society Open Science.

We would like to thank you and the reviewers for their comments and suggestions. Below, in bold, you can find the detailed responses to all comments.

Kind regards,

Balazs Aczel, on behalf of all co-authors

Editor's Comments

The reviewers and Subject Editor have recommended publication, but also suggest some minor revisions to your manuscript. Therefore, I invite you to respond to the comments and revise your manuscript.

All four reviewers from Stage 1 very kindly returned to assess the Stage 2 manuscript. Reviewers 1, 2 and 3 are broadly positive while noting areas that would benefit from some improvement, including the balance of the conclusions, general presentation and usability of the OSF repository.

Reviewer 4 is more critical, in fact recommending rejection, largely on the basis of various deviations from the approved protocol and disagreement about the interpretation of results (linking to Stage 2 primary criterion #2 -- whether the conclusions are based on the evidence). The reviewer's assessment is very detailed and helpful and I would like to provide some guidance on certain points to address in revision. First, concerning deviations from the Stage 1 protocol, it is vital that these are as transparent as possible and clearly justified. Second, please do not make any changes to the rationale or Introduction to address additional literature. Any points raised by the reviewer in this regard (e.g. the reviewer's comment about insufficient contextualisation) should instead be addressed in the Discussion. Third, like Reviewer 4, the reviewer also finds the OSF archive difficult to navigate (with potentially missing data) so please address this carefully. And of course, please address all other remaining points raised by this reviewer and the other reviewers in a point-by-point response and revision.

For this revision, we

- **tried to make the manuscript more clear and transparent by extending our descriptions, as detailed below;**
- **we increased the contextualisation of our research in the Discussion;**
- **we ensured that our OSF archive is well organised;**
- **addressed all the comments of the reviewers.**

Reviewers' comments to Author:

Reviewer: 1

Comment 1

I have no issues with the study itself (methods, data, analyses, or conclusions). However, the presentation of both the report and the raw data are in need of improvement. Specifically,

1. The text is difficult to read due to grammatical issues, e.g.,
 - a. Misplaced commas (e.g., "was followed by either a Stroop, or a sentence stimulus")
 - b. Misplaced ellipses ("on congruent, on incongruent trials or on both")
 - c. Missing words ("As the original paper, we also", "if we regard the two more similar in nature")
 - d. Unusual phrases ("To this aim", "We found the effect of the congruency")

None of these instances is particularly dramatic, however, the cumulative effect is that the text is rather hard-going, to an extent that I suspect will negatively affect the study's impact.

Reply 1

Thank you very much for spotting these mistakes in the text. For this revision, we carefully revised the language of the manuscript.

Comment 2

2. Some information is incorrect or missing:

- a. In the figure captions, "The X axis shows the congruency type of the current trial," this should probably read "previous trial" (if the figure legends and axes titles are to be trusted)

Reply 2

We corrected all figure captions.

Comment 3

- b. I could not find information on what the error bars represent

Reply 3

We added one sentence of the following explanation under each figure with an error bar: The error bars represent the 95% confidence interval.

Comment 4

c. In the "Supporting tests of interest", the authors seem report large difference in *Stroop effects* ("tested the extent of the Stroop effect") as a function of the preceding trial's congruency, which stands in clear contrast to the preceding analyses and the data as shown in the figures. I suspect they mean differences in mean RTs (not in Stroop effects) as a function of the preceding trial's congruency.

Reply 4

We updated the text to make it clear that the supporting test of interest tests the main effect of the previous trials congruency rather than the interaction effect.

Comment 5

3. The figures are generally of low quality. In addition, however, I would strongly recommend to use the same type of chart as in the Kan et al paper (i.e., bar graphs instead of line graphs) in order to facilitate comparison between the two papers

Reply 5

We think that for reporting congruency sequence effect the best practice is to use line graphs, as they can easily visualise the presence of interactions very well. In this regard, we follow the examples of previous pivotal publications in the field, such as Botvinick, et al. (2001) and Egnér (2007). We were unsure which attributes of the presented figures conveyed low quality, nevertheless, we increased the font size on the axes to aid readability.

Comment 6

4. The raw data on OSF is of such low quality as to be almost useless to anyone other than the authors themselves:

- a. Lacking documentation – each data file should have a corresponding file / worksheet explaining the data in each column
- b. Variable names are not sufficiently informative
- c. Large proportions of each data file appear to be merely tags inserted by the program that hold no relevant information (e.g., dozens of columns filled with “N/A”), making the data files slow to download and difficult to handle.

Signed

Friederike Schlaghecken

Reply 6

Before our last submission, we uploaded the raw and the processed data files to our OSF page. We understand that the raw datafile contains several non-relevant variables. However, the raw data are the anonymised output of the data collection software (OpenSesame), hence the high amount of redundant variables. We included the data formatting script that creates the processed data for further analyses.

Formatting script -> “Kan_Raw_Processed.Rmd”

Processed data -> eg. “Kan_Processed_Exp3_Main_Rt_data.tsv”

To assist in better comprehension of the datafiles, we have now included a corresponding metadata file for each processed datafile in the projects Github repository (https://github.com/marton-balazs-kovacs/kan_rep_analysis/tree/master). This repository is linked to our OSF page in the Analysis component (<https://osf.io/kyz8b/>). The metadata files contain a comprehensive explanation of the variables of the processed data files. Each metadata file has the same name as the processed data file.

Reviewer: 2**Comment 7**

I think this is an interesting paper that will be of value to the scientific community. As the authors themselves indicate in the Introduction, the evidence for conflict adaptation across domains is somewhat mixed, and one of the biggest strengths of the paper is the large sample sizes by which they tested the cross-task cognitive control manipulations. The methods and statistical analyses seem sound.

My one general comment is for the authors to consider somewhat tempering the conclusions of the paper. Yes, it is true that the authors were not able to replicate the perceptual/verbal experimental manipulations with more subjects than in the original Kan et al. study. But at the same time, it is also true that with 108 subjects, the authors did find support for syntactic/non-syntactic cross-task adaptation as measured by accuracy. While this is a "single exception" I wonder if the authors could speculate as to why they found the support in this manipulation but not the perceptual/verbal manipulations.

Reply 7

Thank you for your suggestion, we extended our Discussion with our speculation on this matter.

Comment 8

Relatedly, the concept of affect impacting conflict adaptation processes is interesting, and the paper could benefit from further speculation from the authors on how they might include that within a cognitive control framework, as well as intersect with different cognitive control domains.

The authors might also be interested in recent findings demonstrating that experiencing perceptual or attentional conflict can result in adaptation processes that impact sentence processing:

<https://www.tandfonline.com/doi/full/10.1080/23273798.2020.1836379>

Reply 8

We thank the reviewer for this suggestion. Now, we refer to this study in our Discussion.

Reviewer: 3**Comment 9**

My opinion of the manuscript is positive and I have no suggestions for improvement. I thought that the authors handled the somewhat difficult pattern of results well in the discussion.

I responded 'no' to Stage 2 Primary Criterion #1 ("Are the introduction and methods the same as the approved Stage 1 submission?"), as the authors note some minor deviations in their supplementary material. However, I thought that these deviations were reasonable, and do not compromise the interpretation of the main results.

Signed,
Craig Hedge

Reply 9

Thank you.

Reviewer: 4**Comment 10**

Review of "Is there evidence for cross-domain control adaptation? A replication of Kan et

al. (2013)” Stage 2 Primary Criterion #1: Whether the introduction and methods are the same as the approved Stage 1 submission Strictly speaking, the introduction differed from the approved Stage 1 submission (based on the Track Changes version provided by the authors). However, the changes were minor wording changes that did not impact the meaning of the text. There were methodological differences between the approved Stage 1 submission and the Stage 2 submission.

- First, additional data cleaning procedures were introduced in the current submission. With that said, the data cleaning procedures are in line with common practice in the field.
- Second, the authors noted three additional deviations in the Supplementary Materials. 1) The classic Stroop effect was moved from a “crucial” to “supporting” test. This change is appropriate because the Stroop effect was not part of the critical hypotheses of the original paper. 2) The authors did not collect participants’ gender or age so as to allow anonymous data collection (perhaps due to IRB and/or practical data collection concerns). This is unfortunate, because it precludes comparison of the demographics of the original sample to the replication sample. The authors should mention this as a limitation in the Discussion.

Reply 10

We now have this limitation mentioned in the Discussion.

Comment 11

3) The authors conducted ANOVAs and follow-up t-tests rather than t-tests only. This change is appropriate because it brings the analyses in line with the original paper and the conflict congruency sequence effect literature, which typically examines the interaction of preceding and current trial type.

- Third, an examination of the raw data file on OSF’s website reveals something unusual and unexpected in the baseline block of Stroop trials. (Note: I only looked at Experiment 1, so I’m not sure if this also applies to Experiments 2 & 3.) If I understand the file structure correctly, it appears that many of the trials within the baseline block did not use color words (e.g., tent, house, hungry). The inclusion of these trials was not mentioned in the manuscript. Given the extensive literature documenting the influence of incongruent trials proportion on Stroop performance, this design choice needs to be acknowledged and justified. Furthermore, how the inclusion of these neutral stimuli may have affected subjects’ expectations of the experimental task and the results should also be addressed.

Reply 11

We used the unmodified baseline test sequence that was provided by the original authors through personal communications. As we did not run any analyses on the baseline data the proportion of the congruency types was not relevant to our study. The instruction of the baseline trials was the same as in the experimental trials: participants needed to signal the displayed colour of the appearing words. Now, we made this explicit in the description of our methods (page no. 6).

Comment 12

Stage 2 Primary Criterion #2: Whether the authors’ conclusions are justified given the data. Several major issues were identified in the authors’ conclusions.

- Broadly speaking, the authors did not provide adequate discussion of the discrepancies in findings between the two studies. For example, the authors only provided a cursory

explanation of the contrasting patterns in RT data in Experiment 1. Another possibility for the difference may lie in the differences in the samples. In the current study, the participants are sampled from 3 different dialects of English, whereas the original study tested only Mainstream American English speakers (which is not represented in the current sample). This may or may not matter, but since the RT effects did not replicate, one wonders whether different dialects have distributions of the same verbs in different grammatical environments, generating various biases that speakers of the different dialects use to guide comprehension. If so, it is possible that these differences would produce different ambiguity effects that may interact with the need for cognitive control in different ways (e.g., speakers of dialect X have a larger garden-path effect than speakers of dialects Y and Z). Perhaps the self-paced reading time is not sensitive enough to detect these kinds of differences, but it would be worth discussing such possibilities.

Reply 12

All samples of Exp 1 consisted of exclusively native English speaker university students. Based on the fact that the original paper did not mention any sample attributes as preconditions to observe the effect of interest, we could not anticipate the resulting difference in this regard. Different dialects of English may or may not moderate the extent of the congruency sequence effect. However, this is not central to the main question tested by this registered report, namely, the cross-domain generality of conflict adaptation. What is relevant is that the outcome-neutral test of Experiment 1 found evidence for the presence of the ambiguity effect deeming the quality of the data sufficient to test the crucial hypothesis. Therefore, we judge that no further post-hoc test is necessary here.

Comment 13

Additionally, the authors ignored large differences between the samples in overall accuracy on the Stroop task in Experiment 1. The original study observed 97%, 94%, 97%, and 97% accuracy in CC, CI, IC, and II conditions, respectively. The replication reported (approximately) 87%, 82%, 87%, and 87% accuracy in the same conditions. Therefore, overall accuracy was about 10% lower in the current study. This could reflect differences in inhibitory control ability between the 2 participant groups (the original sample consisted exclusively of college students at a private university in the U.S.) or different strategies used during the Stroop task (i.e., placing the emphasis on accuracy versus speed). These accuracy differences could have affected the RT results—if participants in the present study were responding too quickly to maintain high accuracy, then engagement of cognitive control may have slowed responding to improve accuracy (akin to a post-error slowing effect).

Reply 13

The participants of the replication study received the same instructions under similar testing circumstances as the participants of the original study. Moreover, in both cases the sample consisted of native English speaking university students. Based on these similarities and on the fact that the original paper did not mention any sample attributes as preconditions to observe the effect of interest, we could not anticipate the resulting difference in the accuracy. We have now included a sentence in the Discussion about the apparent difference in the accuracy scores, however, we believe that this should not affect the conclusions drawn .

Comment 14

• Furthermore, the current replication is insufficiently contextualized. The introduction is unusually light on literature reviews (which I also noted in the Stage 1 review). Apart from describing the Kan study, unfortunately, only one follow-up study was summarized (Hsu & Novick, 2016), but there have been several other cross-task replications and extensions since the original Kan article, particularly in the area of sentence processing (e.g., Thothathiri et al., 2018 in *Cognition*; Hsu, Kuchinsky, & Novick, 2020 in *Language, Cognition, and Neuroscience*; Adler, Valdes Kroff, & Novick, 2020 in *JEP:LMC*; Navarro Torres et al., 2018 in *Brain Research*). All of this other work details careful theoretical ideas and predictions about shared processing demands across tasks, which this attempted replication does not. Perhaps it does not need to since it is a registered report, but minimally the other work should be considered with respect to their specific accounts and how they contribute to the notions that the current authors test. Two particularly relevant studies are Adler et al., 2020 and Hsu et al., 2020. In Adler, they observe that the flanker-conflict effect is reduced following a code-switched sentence as opposed to a monolingual sentence, suggesting that the cross-linguistic conflict detected during processing/integration of a code-switch engages cognitive control. The direction of the effect is the same as that in Kan et al., which the current authors try to replicate. It also uses Flanker, which is non-verbal and weighs-in on the domain-general issue the current authors address. Similarly, Hsu et al., 2020 show that both perceptual conflict and non-verbal conflict engage control mechanisms that facilitate syntactic revision using fine-grained eye-movement measures. This paper addresses both the domain-general and perceptual issues the current authors address. Thus, it is notable that none of this is reviewed or discussed in terms of the current replication attempt.

Reply 14

Thank you very much for your suggestions. Some of these studies were published only after our original submission, but now we refer to them in our Discussion.

Comment 15

• In Kan et al.'s Experiment 2, the key metric used to index conflict experienced by the participants is the number of reversals, with the key finding being a positive relationship between number of reversals experienced and magnitude of conflict adaptation effect. Furthermore, in conjunction with findings of Experiment 3, Kan and colleagues suggested that conflict experienced is crucial to the observation of conflict adaptation effects. Thus, it seems that one of the most apparent explanations for the current replication failure likely lies in the substantially lower number of reversals experienced by the current sample. However, this issue was neither noted nor discussed. Indeed, the number of reversals experienced by the current "high reversal" group ($M = 18.6$, $SD = 15.9$) is similar to the number of reversals experienced by the "low reversal" group in the Kan et al. paper ($M = 16.3$, $SD = 3.9$). A rough comparison based on an independent samples t-test revealed that mean number of reversals experienced by these two groups failed to reach significance, $t [93] = 0.538$, $p = .592$. (Note: the actual number of subjects in this group in the current sample was not reported; the t-test was conducted assuming 81 subjects). In sum, since "low reversal" subjects in Kan et al. did not experience conflict adaptation, it is unsurprising that the current "high reversal" sample also did not display a conflict adaptation effect. After all, if the participants did not experience

sufficient conflict (as Kan et al. hypothesized), they would not be expected to engage cognitive control that would result in conflict adaptation.

o Among the high reversal subjects, participants in the Kan et al. study experienced significantly more reversals ($M = 31.8$, $SD = 8.3$) than subjects in the current sample ($M = 18.6$, $SD = 15.9$), $t [93] = 3.03$, $p = .003$. Similarly, the low reversal participants in the Kan et al. study also experienced a higher number of reversals ($M = 16.3$, $SD = 3.9$) than the current sample ($M = 3.1$, $SD = 2.3$), $t [93] = 17.91$, $p < .001$.

Reply 15

We respectfully disagree with this line of reasoning as we think that it is neither (1) appropriate nor (2) possible to conduct this analysis.

(1) We don't see that the original work would claim that low and high reversals are compared to some absolute values, instead they gain meaning only within the sample. Therefore, we think it has no meaning to compare one group of one study to another group of another study. In fact, the mean difference between our low and high reversal groups was comparable to the one in the original study (15.51 in our study; 16 in the original study).

(2) As it is a registered report, we are bound to follow the procedure accepted in the first round of the study.

The number of participants in each reversal group is now added to the manuscript.

Comment 16

- The presentation of rationale and results for Experiment 3 was misleading. As stated in Kan et al., the goal of Experiment 3 was to examine an alternative effort-related explanation for the results of Experiment 2. That is, "in Experiment 2, one could argue that, compared to congruent Necker trials, incongruent Necker trials were more effortful and demanded more attention because they lasted for a longer duration and required multiple responses throughout a trial. We ... [hypothesized] that greater effort alone, in the absence of conflict, should not trigger cognitive control functions that operate across trials. In other words, the experience of internal conflict should be critical to showing adaptation effects; without it, behavioral adjustments are not expected to occur." Thus, the expectation is that conflict adaptation will not be observed in Experiment 3, where internal conflict is not induced. And the findings were in line with these expectations. However, this rationale was not clearly explained in Experiment 3. Without this context, a reader unfamiliar with the Kan et al. paper may misinterpret this pattern as a replication failure rather than a pattern of findings that in fact aligned with the hypothesis from Kan et al.

In sum, given the major concerns described above, the current manuscript did not meet this review criterion.

Reply 16

Thank you for highlighting this aspect of the study. We have now added several lines to ensure that this point is interpreted correctly by the reader.

Comment 17

Other major concerns:

The paper uses "attention control" and "cognitive control" interchangeably, and it is not clear

why when other work draws distinctions (for a lengthy discussion, including empirical data on the difference, see abovementioned Hsu et al., 2020).

There should also be discussion of work by those in the conflict adaptation arena that illustrates the conditions under which the effect is found and the conditions under which it is not (e.g., when subjects perceive the two tasks as separate, creating a “task set”). See work by Daniel Weissman’s lab for inspiration (Grant et al., 2020). Discussions of these important issues will further contextualize the current findings and will strengthen this work’s potential contribution to the field.

Reply 17

We could not find the use of “attention control” in our manuscript. We used “attentional resources” at one point. Now, we changed it to “control resources”.

Comment 18

The authors state that the demarcation of sentence regions is included in the OSF repository (p.6, ln 58). However, I could not locate this information. This is relevant for concerns regarding the computation of residual reading times (below).

Reply 18

We made sure that the OSF repository as well as the github repository that contains the file with the demarcation of each sentence region is publicly available on the internet: <https://osf.io/2v39r/> The link is included in the manuscript and we added an additional reference in the Supplementary materials.

Comment 19

The authors provided additional methodological detail about the computation of residual reading times in the Supplementary Materials. This raised concerns about the computation, which deviated substantially from common practice and the original work. 1) The authors included congruent and filler sentences and excluded incongruent sentences in the linear regression for residual reading times, whereas the original work included congruent and incongruent sentences but excluded filler sentences. The filler sentences have a different syntactic structure and are not comparable to the congruent/incongruent sentences, and so should not be included in this computation.

Reply 19

While it is true that additional methodological details were provided in the Stage 2 submission regarding the computation of the residual reading times, the information that only the filler and congruent sentences were included in the linear regression models has already been stated in the Stage 1 manuscript. However, as a sensitivity analysis, we have now added the analysis method suggested by the reviewer in the Supplementary materials. Its results show a similar pattern with the only minor difference in the disambiguating region. The registered method showed inconclusive results in that region with $B = 2.88$ but with the analysis suggested by the reviewer we found a good enough evidence for the difference in the same direction for the residual reading time between the congruent and incongruent sentences with $B = 3.33$.

Comment 20

2) It is not clear that the authors compare the correct regions of the sentences. They state, “As the sentences have different lengths, some sentence region’s mean residual reading times were calculated from more data points than others. As the congruent sentences have more sentence regions than the incongruent sentences the last two sentence regions for congruent sentences did not have an incongruent pair to do the comparison, therefore, we excluded these sentence regions from further analysis” (Supp. Mat., p. 3). The congruent and incongruent sentences in the original study were designed to have the same number of words (11, after excluding ‘that’ from congruent sentences). Therefore, sentence length should not have affected the number of contributing data points and congruent sentences should not have had more regions than the incongruent sentences. This difference may have been due to the inclusion of filler sentences in the calculation.

Reply 20

The demarcation of all the sentences into sentence regions was not provided in the original paper but only for the example sentence used throughout the paper. We followed this example throughout the demarcation of all the sentences. The result of the demarcation process was shared with the reviewers on OSF after the Stage 1 manuscript was submitted. The demarcation of the sentences led to a different number of sentence regions (and not words, as a sentence region can contain more than one word) for certain sentences. However, this difference only affected the eight and ninth sentence regions which are not crucial for the test of the ambiguity effect as the temporarily ambiguous and disambiguating regions are the third and fourth respectively.

Comment 21

Additionally, authors state, “we compared the fourth sentence region of the incongruent sentences to the third sentence region of the congruent sentences” (Supp. Mat., p. 3). This should be reversed, if the numbers are assigned based on word position with the additional word “that” in congruent sentences.

Reply 21

We thank the reviewer for pointing out this mistake in our wording. We made the appropriate changes in the text: we compared the fourth sentence region of the congruent sentences to the third sentence region of the incongruent sentences” (Supp. Mat., p. 3).

Comment 22

Additionally, more than just these two regions were compared (ambiguous region, disambiguating region, post-disambiguating region, etc.).

Reply 22

Although special emphasis is given to the main regions of interest (ambiguous region and disambiguating region) in the manuscript, the result of the comparisons (in addition with the results of the analysis suggested by the reviewer) for all the regions can be found in the Supplementary materials.

Comment 23

Although this analysis is “outcome-neutral,” it is nevertheless important because residual reading times are used to index the ambiguity effect, which is expected to relate to the magnitude of congruency sequence effects (i.e., greater experience of conflict results in larger adaptation).

Reply 23

Analyses labelled as “outcome-neutral” are meant to indicate that positive finding on the given test is a prerequisite for the crucial test, i.e. without finding an ambiguity effect the test of the congruency sequence effect is not viable. This labelling practice is in line with the guideline of the journal for Registered Reports: <https://royalsocietypublishing.org/rsos/registered-reports#Stage1>.

Comment 24

The authors report smaller ambiguity effects than the original study (17.26 ms and 7.56 ms vs 40 ms and 18 ms), which could have resulted in smaller CSEs. However, this apparent discrepancy could also be due to their unconventional computation of residual reading times and/or different assignment of sentence regions.

Reply 24

We registered an outcome-neutral test, as a prerequisite of our crucial test, to assess whether there is an ambiguity effect caused by the garden-path design. Based on the results of the outcome-neutral test and our registered decision criteria, we concluded that the manipulation was successful and it is possible to test the congruency sequence effect. Therefore, we respectfully disagree with the reviewer on whether this discrepancy between the size of the ambiguity effects between the original study and the replication attempt should be taken into account. We believe that a similar argument could be applied here as in Reply 15 regarding the difference in the number of reversals in each reversal group.

References

Egner, T. (2007). Congruency sequence effects and cognitive control. *Cognitive, Affective, & Behavioral Neuroscience*, 7(4), 380-390.

Botvinick, M. M., Braver, T. S., Barch, D. M., Carter, C. S., & Cohen, J. D. (2001). Conflict monitoring and cognitive control. *Psychological review*, 108(3), 624-652.